# Understanding Edge-of-Stability Training Dynamics with a Minimalist Example

**Xingyu Zhu**[*1]**, Zixuan Wang**[*2]**, Xiang Wang**[1]**, Mo Zhou**[1]**, Rong Ge**[1]
[1]Duke University, [2]Tsinghua University
{xingyu.zhu,zw270}@duke.edu,{xwang,mozhou,rongge}@cs.duke.edu

## Abstract

Recently, researchers observed that gradient descent for deep neural networks operates in an "edge-of-stability" (EoS) regime: the sharpness (maximum eigenvalue of the Hessian) is often larger than stability threshold $2/\eta$ (where $\eta$ is the step size). Despite this, the loss oscillates and converges in the long run, and the sharpness at the end is just slightly below $2/\eta$. While many other well-understood nonconvex objectives such as matrix factorization or two-layer networks can also converge despite large sharpness, there is often a larger gap between sharpness of the endpoint and $2/\eta$. In this paper, we study EoS phenomenon by constructing a simple function that has the same behavior. We give rigorous analysis for its training dynamics in a large local region and explain why the final converging point has sharpness close to $2/\eta$. Globally we observe that the training dynamics for our example have an interesting bifurcating behavior, which was also observed in the training of neural nets.

## 1 Introduction

Many works tried to understand how simple gradient-based methods can optimize complicated neural network objectives. However, recently some empirical observations show that optimization for deep neural networks may operate in a more surprising regime. In particular, Cohen et al. (2021) observed that when running gradient descent on neural networks with a fixed step-size $\eta$, the sharpness (largest eigenvalue of the Hessian) of the training trajectory often oscillates around the stability threshold of $2/\eta$[1], while the loss still continues to decrease in the long run. This phenomenon is called "edge-of-stability" and has received a lot of attention (see Section 1.2 for related works).

While many works try to understand why (variants of) gradient descent can still converge despite that the sharpness is larger than $2/\eta$, empirically gradient descent for deep neural networks has even stronger properties. As shown in Fig. 1a, for a fixed initialization, if one changes the step size $\eta$, the final converging point has sharpness very close to the corresponding $2/\eta$. We call this phenomenon "sharpness adaptivity". Another perspective on the same phenomenon is that for a wide range of initializations, for a fixed step-size $\eta$, their final converging points all have sharpness very close to $2/\eta$. We call this phenomenon "sharpness concentration".

Surprisingly, both sharpness adaptivity and sharpness concentration happen on deeper networks, while for shallower models of non-convex optimization such as matrix factorization or 2-layer neural networks, the gap between sharpness and $2/\eta$ is often much larger (see Fig. 1b). This suggests that these phenomena are related to network depth. What is the mechanism for sharpness adaptivity and concentration, and how does that relate to the number of layers? To answer these questions, in this paper we consider a minimalist example of edge-of-stability.

More specifically, we construct an objective function (4-layer scalar network with coupling entries), such that gradient descent on this objective has similar empirical behavior as deeper networks. We give a rigorous analysis for the training dynamics of this objective function in a large local region, which proves that the dynamics satisfy both sharpness adaptivity and sharpness concentration. The global training dynamics for our objective exhibit a complicated fractal behavior (which is also why our rigorous results are local), and such behavior has been observed in training of neural networks.

---

[*]Equal Contribution.
[1]The value $2/\eta$ is called the stability threshold, because if the objective has a fixed Hessian, the gradient descent trajectory will become unstable if the largest eigenvalue of the Hessian is larger than $2/\eta$.

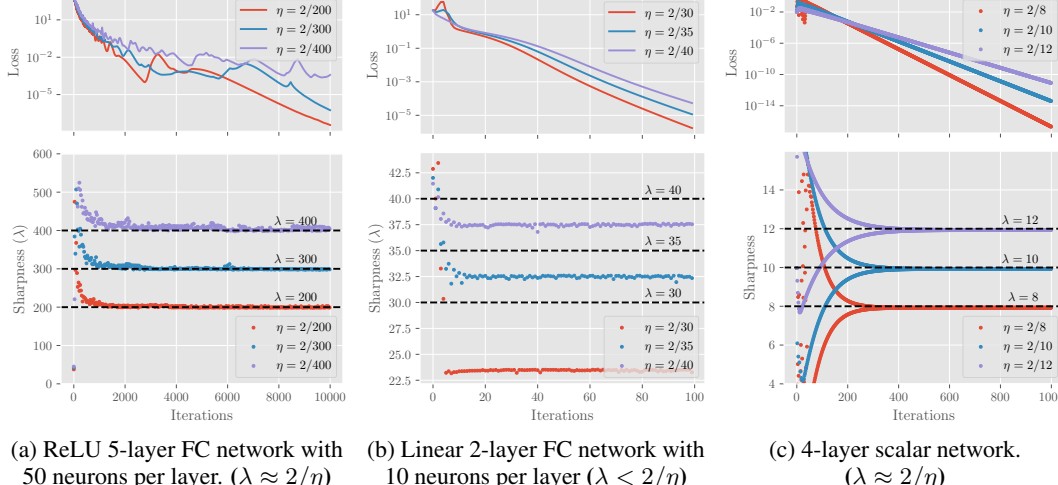

(a) ReLU 5-layer FC network with 50 neurons per layer. ($\lambda \approx 2/\eta$)

(b) Linear 2-layer FC network with 10 neurons per layer ($\lambda < 2/\eta$)

(c) 4-layer scalar network. ($\lambda \approx 2/\eta$)

Figure 1: **EoS Phenomena in NN Training.** We consider three models including a 5-layer ReLU activated fully connected network, a 2-layer fully connected linear network with asymmetric initialization factor $(4, 0.1)$ (see Appendix A.1 for explanation), and a 4-layer scalar network equivalent to $\min_{x,y} \frac{1}{4}(1 - x^2 y^2)^2$. For each model we run gradient descent from the same initialization using different learning rates. For (a) and (c), the sharpness converges very close to $2/\eta$ with loss continuing to decrease. For (b), the sharpness decreases to be significantly lower than $2/\eta$.

## 1.1 OUR RESULTS

The objective function we consider is very simple: $\mathcal{L}(x, y, z, w) \triangleq \frac{1}{2}(1 - xyzw)^2$. One can view this as a 4-layer scalar network (each layer has a single neuron). We even couple the initialization so that $x = z, y = w$ so effectively it becomes an objective on two variables $\mathcal{L}(x, y) \triangleq \frac{1}{4}(1 - x^2 y^2)^2$. For this objective function we prove its convergence and sharpness concentration properties:

**Theorem 1.1** (Sharpness Concentration, Informal). *For any learning rate $\eta$ smaller than some constant, there is a constant size region $\mathbb{S}_\eta$ such that the GD trajectory with step size $\eta$ from all initializations in $\mathbb{S}_\eta$ converge to a global minimum with sharpness within $(2/\eta - \frac{20}{3}\eta, 2/\eta)$.*

As a direct corollary, we can also prove that it has the sharpness adaptivity property.

**Corollary 1.1** (Sharpness Adaptivity, Informal). *There exists a constant size region $\mathbb{S}$ and a corresponding range of step sizes $\mathbb{K}$ that for all $\eta \in \mathbb{K}$, the GD trajectory with step size $\eta$ from any initialization in $\mathbb{S}$ converges to a global minimum with sharpness within $(2/\eta - \frac{20}{3}\eta, 2/\eta)$.*

The training dynamics are illustrated in Fig. 2. To analyze the training dynamics, we reparametrize the objective function and show that the 2-step dynamics of gradient descent roughly follow a parabola trajectory. The extreme point of this parabola is the final converging point which has sharpness very close to $2/\eta$. Intuitively, the parabola trajectory comes from a cubic term in the approximation of the training dynamics (see Section 3.1 for detailed discussions). We can also extend our result to a setting where $x, y$ are replaced by vectors, see Theorem 3.2 in Section 3.3.

In Section 4 we explain the difference between the dynamics of our degree-4 model with degree-2 models (which are more similar to matrix factorizations or 2-layer neural networks). We show that the dynamics for degree-2 models do not have the higher order terms, and their trajectories form an ellipse instead of a parabola.

In Section 5 we show why it is difficult to extend Theorem 3.1 to global convergence – the training trajectory exhibits fractal behavior globally. Such behaviors can be qualitatively approximated by simple low-degree nonlinear dynamics standard in chaos theory, but are still very difficult to analyze.

Finally, in Section 6 we present the similarity between our minimalist model and the GD trajectory of some over-parameterized deep neural networks trained on a real-world dataset. Toward the end of convergence, the trajectory of the deep networks mostly lies on a 2-dimensional subspace and can be well characterized by a parabola as in the scalar case.

## 1.2 RELATED WORKS

The phenomenon of gradient descent on the Edge of Stability (EoS) was first formalized and empirically demonstrated in Cohen et al. (2021). They show that the loss can non-monotonically decrease even when the *sharpness* $\lambda > 2/\eta$. The non-monotone property of the loss has also been observed in many other settings (Jastrzebski et al., 2020; Xing et al., 2018; Lewkowycz et al., 2020; Wang et al., 2022; Arora et al., 2018; Li et al., 2022a).

Recently several works try to understand the mechanism behind EoS with different loss functions under various assumptions (Ahn et al., 2022; Ma et al., 2022; Arora et al., 2022; Lyu et al., 2022; Li et al., 2022b). Ahn et al. (2022) studied the non-monotonic decreasing behavior of gradient descent (which they call *unstable convergence*) and discussed the possible causes of this phenomenon. From a landscape perspective, Ma et al. (2022) defined a special *subquadratic* property of the loss function, and proved that EoS occurs based on this assumption. Despite the simplicity, their model displayed the EoS phenomenon without sharpness adaptivity. Instead, our model focuses on a minimalist scalar network and proves the convergence results together with the sharpness adaptive phenomenon.

Arora et al. (2022) and Lyu et al. (2022) studied the implicit bias on the sharpness of gradient descent in some general loss function. Both works focus on the regime where the parameter is close to the manifold of minimum loss. Arora et al. (2022) proved that with a modified loss $\sqrt{L}$ or using normalized GD, gradient descent enters the EoS regime and has a sharpness reduction effect around the manifold of minima. Lyu et al. (2022) provably showed how GD enters EoS regime and keeps reducing *spherical sharpness* on a scale-invariant objective. In both works, the effective step-size $\eta$ changes throughout the training process, so sharpness adaptivity and concentration do not apply. Our results start from a simpler example without normalization, whereas the above works focus on general functions with normalized gradient or scale-invariance property.

Another line of works (Lewkowycz et al., 2020; Wang et al., 2022) focuses on the implicit bias introduced by a large learning rate. Lewkowycz et al. (2020) first proposed "catapult phase", a regime similar to the EoS, where loss does not diverge even if sharpness is larger than $2/\eta$. Wang et al. (2022) provided a convergence analysis on the matrix factorization problem for large learning rate beyond $2/\lambda$ where $\lambda$ is the sharpness. Their results include two stages: in the first phase, the loss may oscillate but never diverge; the sharpness decreases to enter the second phase, where the loss decreases monotonically. Recently Li et al. (2022b) provided a theoretical analysis on sharpness along the GD trajectory in a two-layer linear network setting under some assumptions during the training process. These works mostly focus on the degree-2 setting which does not have the sharpness adaptivity and sharpness concentration properties.

## 2 PRELIMINARIES AND NOTATIONS

In this section, we introduce the minimalist model which exhibits both sharpness adaptivity and sharpness concentration.

### 2.1 GRADIENT DESCENT ON PRODUCT OF 4 SCALARS

We focus on the simple objective $\mathcal{L}(x, y, z, w) \triangleq \frac{1}{2}(1 - xyzw)^2$. Let the learnable parameters $x, y, z, w \in \mathbb{R}$ to be trained using gradient descent with a fixed step size $\eta \in \mathbb{R}^+$ that

$$(x_{t+1}, y_{t+1}, z_{t+1}, w_{t+1}) = (x_t, y_t, z_t, w_t) - \eta \nabla \mathcal{L}(x_t, y_t, z_t, w_t). \quad (1)$$

Here $x_t$ denotes the value of parameter $x$ after the $t$-th update. To further simplify the problem, we consider the symmetric initialization of $z_0 = x_0$, $w_0 = y_0$. Note that due to symmetry of objective, the identical entries will remain identical throughout the training process, so the training dynamics reduces to two dimensional and the 1-step update of $x$ and $y$ follows

$$x_{t+1} = x_t - x_t y_t^2 \eta (x_t^2 y_t^2 - 1), \quad y_{t+1} = y_t - x_t^2 y_t \eta (x_t^2 y_t^2 - 1). \quad (2)$$

It's easy to show that the set of global minima for this function form the hyperbola $xy = 1$. Without loss of generality we focus on the case when $x, y > 0$, and in most of the analysis we also focus on the side where $x > y$. As shown in Fig. 2, with GD running on such a minimal model, we observe convergence on EoS for a wide range of initializations. Eventually all such trajectories converge to minima that are just slightly flatter than the "EoS minima" (the minima whose sharpness is exactly $2/\eta$, see Definition 1).

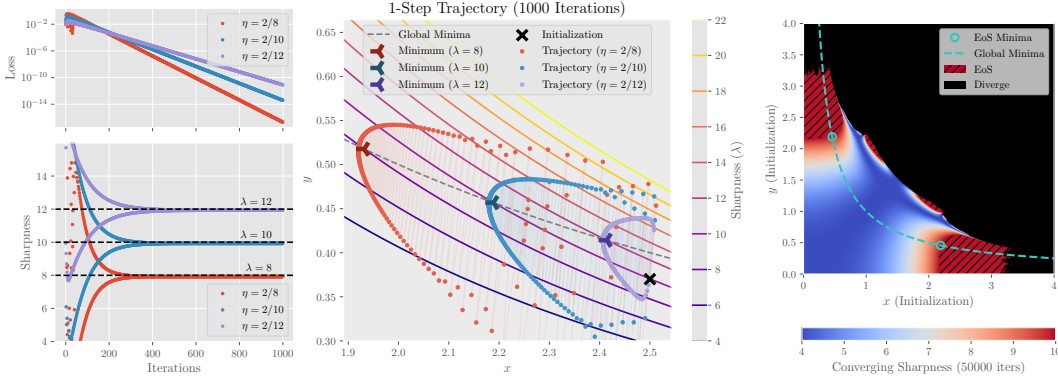

(a) Evolution of training loss, sharpness, and trajectory of GD on the 4 scalar example from the same initialization with different learning rates.

(b) Initializations converging close to $\lambda = 2/\eta$ ($\eta = 0.2$)

Figure 2: **EoS phenomenon on degree-4 model.** In **(a)** we demonstrate sharpness adaptivity by running GD with learning rate $\eta = \frac{2}{8}, \frac{2}{10}, \frac{2}{12}$ from the same initialization. The sharpness of all trajectories converges to around their corresponding stability threshold $2/\eta$ while the loss decreases exponentially. In the 2D trajectory, the 2-step movement quickly converges to some smooth curves ending very close to the EoS minimum. In **(b)** we demonstrate sharpness concentration by running GD with constant learning rate $\eta = 0.2$ for 50000 iterations from a dense grid of initializations and plot the sharpness of their converging minima. Initializations in the red shaded area all converge to a minima with sharpness in $(2/\eta - 0.1, 2/\eta)$.

## 2.2 EoS Minima and Reparameterization

Given that a wide range of initializations all converge very close to the "EoS minima" with sharpness $2/\eta$, we want to concretely characterize those points. The complete calculations are deferred to Appendix B.1. Denote $\gamma = xy$, the Hessian of the objective $\mathcal{L}$ at $(x, x, y, y)$ admits eigenvalues

$$\lambda_1 = \tfrac{1}{2}\left((x^2 + y^2)(3\gamma^2 - 1) + \sqrt{(x^2 + y^2)^2(1 - 3\gamma^2)^2 + 4\gamma^2(3 - 10\gamma^2 + 7\gamma^4)}\right),$$
$$\lambda_2 = \tfrac{1}{2}\left((x^2 + y^2)(3\gamma^2 - 1) - \sqrt{(x^2 + y^2)^2(1 - 3\gamma^2)^2 + 4\gamma^2(3 - 10\gamma^2 + 7\gamma^4)}\right). \tag{3}$$

and $\lambda_3 = x^2(1 - \gamma), \lambda_4 = y^2(1 - \gamma)$. When $(x, y)$ converges to any minimum, $\gamma = xy = 1$, so $\lambda_2, \lambda_3, \lambda_4$ all vanishes. Therefore it is $\lambda_1$ that corresponds to the EoS phenomenon people observe. When $\eta < \frac{1}{2}$, solving $\lambda_1 = 2/\eta$ with $x^2 y^2 = 1$ gives $x = \pm\frac{1}{\sqrt{2}}((-4 + \eta^{-2})^{\frac{1}{2}} + \eta^{-1})^{\frac{1}{2}}$, $y = \pm\sqrt{2}((-4 + \eta^{-2})^{\frac{1}{2}} + \eta^{-1})^{-\frac{1}{2}}$ and their multiplicative inverses. These solutions correspond to the minima with sharpness exactly equal to the EoS threshold of $2/\eta$. Since they are all symmetric with each other, without loss of generality we pick the minimum of interest as follows.

**Definition 1** ($\eta$-EoS Minimum). For any step size $\eta \in (0, \frac{1}{2})$, the $\eta$-EoS minimum under the $(x, y)$-parameterization is

$$(\check{x}, \check{y}) \triangleq \left(\frac{1}{\sqrt{2}}\left((-4 + \eta^{-2})^{\frac{1}{2}} + \eta^{-1}\right)^{\frac{1}{2}}, \sqrt{2}\left((-4 + \eta^{-2})^{\frac{1}{2}} + \eta^{-1}\right)^{-\frac{1}{2}}\right). \tag{4}$$

Though we are able to obtain a closed-form expression for the EoS minimum, its $x$-$y$ coordinate could still be tricky to analyze. Thus we consider the following reparameterization: For any $(x, y) \in \{(x, y) \in \mathbb{R}^+ \times \mathbb{R}^+ : x > y\}$, define $c \triangleq (x^2 - y^2)^{\frac{1}{2}}$ and $d \triangleq xy$. This gives a bijective continuous mapping between $\{(x, y) \in \mathbb{R}^+ \times \mathbb{R}^+ : x > y\}$ and $\{(c, d) \in \mathbb{R}^+ \times \mathbb{R}^+\}$. This is a natural reparameterization since intuitively the basis in the new coordinate system are the two orthogonal family of hyperbolas $xy = C$ and $x^2 - y^2 = C$. The former captures the movement orthogonal to the manifold of minima $xy = 1$ while the latter captures the movement along the manifold of minima. Note that a similar separation of dynamics was also used in Arora et al. (2022).

With $c, d$ as defined, the $\eta$-EoS minimum simplifies to $(\check{c}, \check{d}) \triangleq ((\eta^{-2} - 4)^{\frac{1}{4}}, 1)$. To expand the dynamics near the $\eta$-EoS minimum, we let $a \triangleq c - (\eta^{-2} - 4)^{\frac{1}{4}}$ and $b \triangleq d - 1$ to be the offset from $(\check{c}, \check{d})$. Our analysis will primarily be using the $(a, b)$-parameterization.

**Definition 2** ($\eta$-EoS Reparameterization). For any step size $\eta > 0$, for any $(x, y) \in \mathbb{R}^+ \times \mathbb{R}^+$ such that $x > y$, the $(a, b)$ reparameterization of $(x, y)$ are respectively given by

$$(a, b) \triangleq \left( \left( x^2 - y^2 \right)^{\frac{1}{2}} - \left( \eta^{-2} - 4 \right)^{\frac{1}{4}}, xy - 1 \right). \tag{5}$$

Let $\kappa \triangleq \sqrt{\eta}$, following Eq. (2), the 1-step update under the reparameterization becomes

$$a_{t+1} = (\kappa^{-4} - 4)^{\frac{1}{4}} + \left( a_t + (\kappa^{-4} - 4)^{\frac{1}{4}} \right) \left( 1 - \left( (1 + b_t)^3 - (1 + b_t) \right)^2 \kappa^4 \right)^{\frac{1}{2}},$$

$$b_{t+1} = b_t + ((1 + b_t)^3 - 2(1 + b_t)^5 + (1 + b_t)^7)\kappa^4 \tag{6}$$

$$+ \left( (1 + b_t) - (1 + b_t)^3 \right) \left( 4(1 + b_t)^2 \kappa^4 + (a_t \kappa + (1 - 4\kappa^4)^{\frac{1}{4}})^4 \right)^{\frac{1}{2}}.$$

Now we can proceed to analyze the dynamics of this simple example.

# 3 DYNAMICS OF GRADIENT DESCENT ON DEGREE-4 MODEL

In this section, we will rigorously analyze the training dynamics characterized by Eq. (6). First we will introduce the approximation of one and two-step update and build up intuition on the dynamics. Then we will present our main theoretical results that the degree-4 model exhibits both characterizations of EoS training.

## 3.1 APPROXIMATING 1-STEP AND 2-STEP UPDATES

Here we introduce the informal approximation on Eq. (6) and the corresponding two-step updates. For cleanness of presentation we will use $\approx$ to hide all dominated terms. The rigorous statements of the approximations and corresponding proofs are deferred to Appendix B.3. When we are only describing the one/two-step dynamics, we use $a, a', a''$ to denote $a_t, a_{t+1}, a_{t+2}$ and $b, b', b''$ to denote $b_t, b_{t+1}, b_{t+2}$. Denoting $\kappa \triangleq \sqrt{\eta}$, when $\kappa, |a|, |b|$ are all not too large (see precise ranges in condition B.1), we have

$$a' \approx a - 2b^2\kappa^3, \qquad b' \approx -b - 4ab\kappa - 3b^2 - b^3;$$
$$a'' \approx a - 4b^2\kappa^3, \qquad b'' \approx b + 8ab\kappa - 16b^3. \tag{7}$$

In the approximation, $a$ is monotonically decreasing at a steady rate of $2b^3\kappa^3$ per step. The one step update of $b$ is flipping signs and contains second and third order terms of $b$. For the two-step approximation however, the oscillation behavior and the even-order terms of $b$ all cancels. This is consistent with the analysis in (Arora et al., 2022) that the two step dynamics travels along a sharpness reducing flow.

Before proceeding to analyze the discrete GD movement, we first get intuition by approximating the two-step dynamics with a simple ODE

$$\frac{db}{da} = \frac{b'' - b}{a'' - a} = \frac{16b^3 - 8ab\kappa}{4b^2\kappa^3}. \tag{8}$$

This would be the limit when $\kappa$ is going to 0 and the movement of two-step dynamics become very small. The general solution for Eq. (8) is given by

$$b^2 = \tfrac{1}{2}a\kappa + \tfrac{1}{16}\kappa^4 + C \exp(8a\kappa^{-3}) \tag{9}$$

for some constant $C \in \mathbb{R}$. As $a$ decreases following Eq. (7), the trajectory converges toward the parabola $b^2 = \tfrac{1}{2}a\kappa + \tfrac{1}{16}\kappa^4$. Note that the convergence to the parabola is exponential with respect to $a$, so if $a$ is initialized positive and not too small, it will converge to a minima that is very close to $a = -\tfrac{1}{8}\kappa^3$ as shown in Fig. 3. This is a minimum that is just slightly flatter than the $\kappa^2$-EoS minimum.

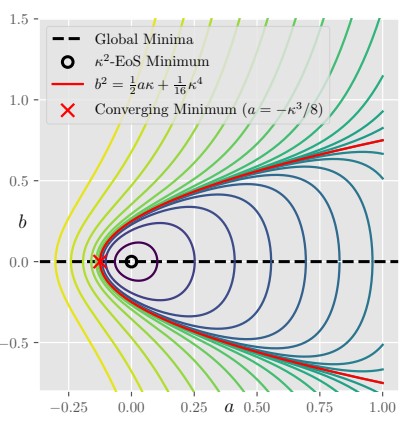

Figure 3: Solutions of Eq. (8) ($\kappa = 1$)

## 3.2 Convergence on EoS for the Degree-4 Model

Now we state our convergence result on the 4 scalar objective under $(a, b)$-parameterization.

**Theorem 3.1** (Sharpness Concentration). *For a large enough absolute constant $K$, suppose $\kappa < \frac{1}{2000\sqrt{2}}K^{-1}$, and the initialization $(a_0, b_0)$ satisfies $a_0 \in (12\kappa^{\frac{5}{2}}, \frac{1}{4}K^{-2}\kappa^{-1})$ and $b_0 \in (-K^{-1}, K^{-1})\backslash\{0\}$. Consider the GD trajectory characterized in Eq. (6) with fixed step size $\kappa^2$ from $(a_0, b_0)$, for any $\epsilon > 0$ there exists $T = \mathcal{O}(K^{-2}\kappa^{-\frac{15}{2}} + \log(\epsilon^{-1}) + \log(|b_0|^{-1})\kappa^{-\frac{7}{2}})$ such that for all $t > T$, $|b_t| < \epsilon$ and $a_t \in (-\frac{5}{3}\kappa^3, -\frac{1}{10}\kappa^3)$.*

Under the context of $x, y$ coordinate and sharpness, Theorem 3.1 gives the following corollary:

**Corollary 3.1** (Sharpness Concentration under $(x, y)$-Parameterization). *For a large enough absolute constant $K$, suppose $\eta < \frac{1}{8000000}K^{-2}$, and the initialization $(x_0, y_0)$ satisfies $x_0 \in (\check{x} + 13\eta^{\frac{5}{4}}, \check{x} + \frac{1}{5}K^{-2}\eta^{-\frac{1}{2}})$ and $|x_0 y_0 - 1| \in (0, K^{-1})$ where $(\check{x}, \check{y})$ is the $\eta$-EoS minima defined in Definition 1. The GD trajectory characterized in Eq. (2) with fixed step size $\eta$ from $(x_0, y_0)$ will converge to a global minimum with sharpness $\lambda \in (\frac{2}{\eta} - \frac{20}{3}\eta, \frac{2}{\eta})$.*

Note that when the step size $\eta$ (and hence $\kappa$) is relatively small, the final sharpness is very close to $2/\eta$. The range of initialization that satisfies the requirement is quite large: in the original $(x, y)$-parameterization it contains a box of width $\Theta(K^{-2}\eta^{-\frac{1}{2}})$ and height $\Theta(K^{-1}\eta^{\frac{1}{2}})$. Many of the initial points can be far from the EoS-minimum.

The complete proofs are deferred to Appendix B.6. Here we discuss the proof sketch of Theorem 3.1. Our convergence analysis focuses on the 2-step update. It contains two phases:

**Phase 1. (Convergence to near parabola)**
We consider initializations in region I, II, and III.
- In I, $b'' - b$ is dominated by $-b^3$ and $(a, b)$ follows an exponential trajectory. We show that $|b|$ decreases exponentially with respect to $a$ and enters region II (Lemma 8).
- In III, $b'' - b$ is dominated by $ab\kappa$ and $(a, b)$ follows an elliptic trajectory centered at $(0, 0)$. We show that $|b|$ increases at superlinearly with respect to $a$ and enters II (Lemma 9).
- We also show that once $(a, b)$ enters II, it will stay in II until it exits from the left and enters IV (Lemma 11). Thus after Phase 1, all initializations will be in IV.

**Phase 2. (Convergence along parabola)**
- After $(a, b)$ enters IV, we show that it will further converge to the parabola that $|b^2 - \frac{1}{2}a\kappa - \frac{1}{16}\kappa^4| < \frac{1}{200}\kappa^4$ will be satisfied before $a$ decreases to $\kappa^{\frac{5}{2}}$ and enters V (Lemma 13).
- Then we show that the inequality will be preserved in V while it moves left until it enters VI (Lemma 14).
- In VI, the dynamics is again similar to III, but with $a$ being negative. We conclude our proof by showing $|b|$ will converge to 0 superlinearly with respect to $a$ (Lemma 15).

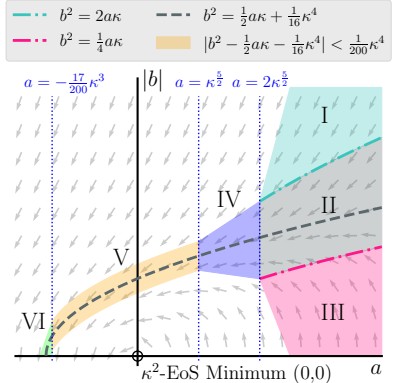

Figure 4: Convergence Diagram for GD on the degree-4 example. The quiver arrows indicate the directions of local 2-step movement. This diagram is only for demonstration purpose and ratios are not exact.

Following Theorem 3.1, we can also formally characterize the sharpness adaptive phenomenon for a local region using the following corollary. The proof is deferred to Appendix B.6.2.

**Corollary 3.2** (Sharpness Adaptivity). *For a large enough constant $K$, fix any $\alpha < \frac{1}{2000\sqrt{2}}K^{-1}$. For all initialization $(x_0, y_0)$ in the region characterized by*

$$x_0 \in (\alpha^{-1} + \frac{1}{15}K^{-2}\alpha^{-1}, \alpha^{-1} + \frac{1}{6}K^{-2}\alpha^{-1}) \quad (10)$$

*and $|x_0 y_0 - 1| \in (0, K^{-1})$, the GD trajectory from $(x_0, y_0)$ characterized by Eq. (2) with any step size $\eta \in (\alpha^2 - \frac{1}{10}K^{-2}\alpha^2, \alpha^2)$ will converge to a minima with sharpness $\lambda \in (\frac{2}{\eta} - \frac{20}{3}\eta, \frac{2}{\eta})$.*

## 3.3 Convergence on EoS for Rank-1 Factorization of Isotropic Matrix

Inspired by the scalar factorization problem, we extend it to a rank-1 factorization of an isotropic matrix. In particular, we consider the following optimization problem:

$$\min_{\boldsymbol{x},\boldsymbol{y}\in\mathbb{R}^d} \frac{1}{4} \left\| \boldsymbol{I}_{d\times d} - \boldsymbol{x}\boldsymbol{y}^\top \boldsymbol{x}\boldsymbol{y}^\top \right\|_F^2 \tag{11}$$

Similar to the under-parameterized case in Wang et al. (2022), this problem also guarantees the alignment between $\boldsymbol{x}$ and $\boldsymbol{y}$ if $(\boldsymbol{x}, \boldsymbol{y})$ is a global minimum, i.e., $\boldsymbol{x} = c\boldsymbol{y}$ for some $c \in \mathbb{R}$. To prove the convergence for Eq. (11) at the edge of stability, we first prove the alignment can be soon achieved. After the alignment, we prove the equivalence between this problem and the degree-4 scalar model, and prove the convergence of this problem.

We directly give the final theorem and the proof is deferred to Appendix C. The experiments demonstrate similar EoS phenomenon (See Appendix A.5).

**Theorem 3.2.** *For a large enough absolute constant $K$, with all the initialization $(\boldsymbol{x}_0, \boldsymbol{y}_0)$ satisfying $\boldsymbol{x}_0 \sim \delta_x Unif(\mathbb{S}^{d-1})$, $\boldsymbol{y}_0 \sim \delta_y Unif(\mathbb{S}^{d-1})^2$, $\delta_x\delta_y = \frac{1}{2}$, $\delta_x \in (\breve{x} + \frac{1}{80}K^{-2}\eta^{-\frac{1}{2}}, \breve{x} + \frac{1}{8}K^{-2}\eta^{-\frac{1}{2}})$, if step size $\eta < \min\{\frac{K^{-4}}{8000000}, \frac{K^{-2}}{20000+2000(\log(d)-\log(\delta_0))}\}$, and a multiplicative perturbation $\boldsymbol{y}'_t = \boldsymbol{y}_t(1 + 2K^{-1})$ is performed at time $t = t_p$ for some $t_p > \mathcal{O}(-\log(\eta) + \log(d) - \log(\delta_0) + K^3)$, then for any $\epsilon > 0$, with probability $p > 1 - 2\delta_0 - 2\exp\{-\Omega(d)\}$ there exists $T = \mathcal{O}(K^{-2}\kappa^{-\frac{15}{2}} - \log(\epsilon) - \log(\delta_0))$ such that for all $t > T$, $\mathcal{L}(x,y) < \epsilon$ and $\|x_t\|^2 + \|y_t\|^2 \in (\frac{1}{\eta} - \frac{10}{3}\eta, \frac{1}{\eta})$.*

Note that we require an additional perturbation because we need to guarantee that the trajectory does not converge to an unstable point (where sharpness $\lambda > 2/\eta$). This was proved without perturbation for the scalar case but is more challenging in higher dimensions. The objective will still converge to a minimum very close to an $\eta$-EoS minimum. The experiment results are available in Appendix A.5.

## 4 DIFFERENCES IN DEGREE-2 AND HIGHER DEGREE MODELS

In this section, we will look at some similar models of lower degree, and explain why for degree-2 models the sharpness of final converging point is often farther from $2/\eta$ compared to higher degree models. We will use similar methods as in Section 2 and Section 3 to gain intuition for the dynamics.

Previous works including (Chen & Bruna, 2022) and (Wang et al., 2022) have studied the dynamics of beyond EoS training on the problem of factorizing a single scalar or an isotropic matrix into two components. The objectives studied includes $\min_{\boldsymbol{x},\boldsymbol{y}\in\mathbb{R}^d}(\mu - \boldsymbol{x}^\top\boldsymbol{y})^2$, $\min_{\boldsymbol{x},\boldsymbol{y}\in\mathbb{R}^d}\|\mu\boldsymbol{I}_d - \boldsymbol{x}\boldsymbol{y}^\top\|_F^2$, and the corresponding scalar case $\min_{x,y\in\mathbb{R}}(\mu-xy)^2$. They were able to show that for initializations with sharpness greater than $2/\eta$, GD with constant learning rate $\eta$ provably converges to a global minimum with sharpness less or equal to $2/\eta$. Empirically, the sharpness reduction process on these 2-component objectives will usually "overshoot" the EoS threshold and converge to a minima that is significantly flatter than the EoS minimum, and one does not observe the oscillation of sharpness around the EoS threshold (see Appendix A.3).

In this section we consider the scalar objective $\min_{x,y\in\mathbb{R}}(1 - xy)^2$ since it is able to captures the major dynamical properties of those more complex objectives as discussed in Wang et al. (2022). As shown in Fig. 5, initializations with sharpness exceeding the EoS threshold will converge to a minima that is distinguishably flatter than the EoS minimum, and globally there is not a region of initialization that gives EoS convergence. Unlike the parabola for the degree-4 case, the 2-step update travels in a roughly circular trajectory centered at the $\kappa^2$-EoS minimum as shown in Fig. 5a (right). Therefore locally we observe that sharper initializations tend to converge to flatter minima.

The difference between the degree-2 and degree-4 case can be easily explained by a local expansion. Using the same $(c, d)$-reparameterization and setting $(a, b)$ to be the offset of $(c, d)$ from the EoS minimum, the two step update of $(a, b)$ under learning rate $\kappa^2$ can be approximated by

$$a'' \approx a - \sqrt{2}b^2\kappa^3, \quad b'' \approx b + 4\sqrt{2}ab\kappa. \tag{12}$$

This is very similar to Eq. (7) except that we no longer have the $-b^3$ term for the 2-step update on $b$ which was attracting $b$ close to 0. In this case, the ODE approximation $\mathrm{d}b/\mathrm{d}a = 4a/b\kappa^2$ gives the general solution $b^2 = 4(C - a^2)/\kappa^2$ for $C \in \mathbb{R}^+$, which corresponds to the family of ellipses centered at $(0,0)$ and matches the two step trajectory in Fig. 5a.

---

[2]$\delta_0 Unif(\mathbb{S}^{d-1})$ denote the uniform distribution over $(d-1)$-dimensional sphere with radius $\delta_0$.

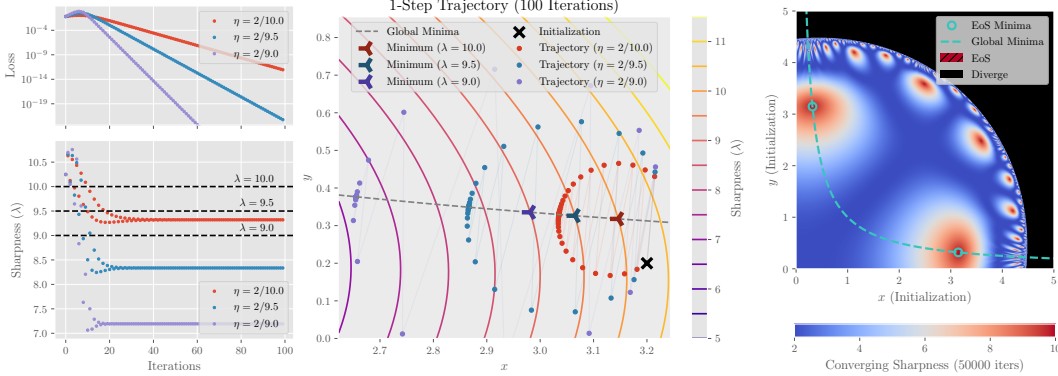

(a) Evolution of training loss, sharpness, and trajectory of GD on the 2 scalar example from the same initialization with different learning rates.

(b) Converging sharpness of initializations ($\eta = 0.2$)

Figure 5: **Beyond EoS training on product of two scalars.** We run the same experiment as in Fig. 2 except for using objective $\min_{x,y \in \mathbb{R}} (1 - xy)^2$. Note that in this case the two-step trajectories form circular curves and converge to points that are farther from EoS minima.

In Appendix A.2.3, we discuss a degree-3 model exhibiting mixed behavior around different EoS minima, which further verifies our explanation above. We also empirically note that the coupling of entries will naturally arise when training general scalar networks from non-coupling initializations. Thus it is not an artifact we have to impose on the model to observe EoS (see Appendix A.4).

## 5   GLOBAL TRAJECTORY AND CHAOS

There exists very limited global convergence analysis for constant step size gradient descent training beyond EoS on complicated non-convex objectives. Even for the product of 4 scalars, the boundary separating converging and diverging initializations (Fig. 6a) exhibits complicated fractal structures.

Moreover, we observe that for initializations close to such boundary, their GD training trajectories usually begin with a phase of chaotic oscillation which eventually "de-bifurcates" and converges to the parabolic two-step trajectory as discussed in Section 3. Similar oscillation phenomenon has also been empirically observed by Ruiz-Garcia et al. (2021) in neural networks when they increase the learning rate and destabilize the network from a local trajectory.

So what is causing the bifurcation? Previously, Ruiz-Garcia et al. (2021) attributed the phenomenon to the cascading effect of oscillation along multiple large eigendirections of the network. Yet this explanation is quite unsatisfying for our simple model as there is only one oscillating direction.

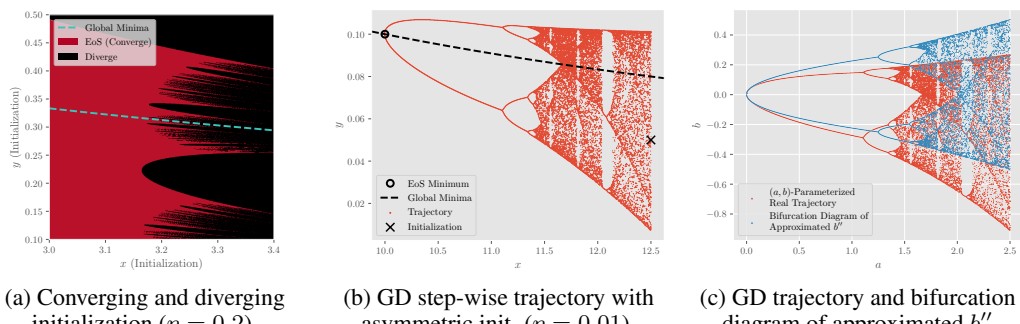

(a) Converging and diverging initialization ($\eta = 0.2$).

(b) GD step-wise trajectory with asymmetric init. ($\eta = 0.01$).

(c) GD trajectory and bifurcation diagram of approximated $b''$

Figure 6: **Bifurcation behavior of GD on the degree-4 model.** In **(a)** we show the zoomed in version of the lower right part of Fig. 2b, the fractal boundary can be clearly observed. In **(b)**, we run GD with $\eta = 0.01$ starting from the asymmetric initialization $(x_0, y_0) = (12.5, 0.05)$ close to the boundary of divergence until it converge close to the EoS minimum at around $(10, 0.1)$. In **(c)**, we plot the trajectory with bifurcation under $(a, b)$-reparameterization and compare it with the bifurcation diagram of the approximated dynamical system characterized by $b'' = b(1 + 8a\kappa - 16b^2)$.

Looking closely to the trajectory (Fig. 6b), one will find it very similar to the bifurcation diagram of self-recurrent polynomial maps (such as the famous logistic map $x_{t+1} = rx_t(1 - x_t)$ parameterized by $r$). In the degree-4 model, the existence of such self-recurrent map is explicit since following Eq. (7), the approximate 2-step update of $b$ can be rewritten as $b'' = b(1 + 8a\kappa - 16b^2)$.

If we consider $a$ to be relatively stationary, the trajectory of $b$ will be locally characterized by the self-recurrent 1D nonlinear dynamical system $b_{t+1} = b_t(1 + 8a\kappa - 16b_t^2)$ parameterized by $a$. In Fig. 6c, we compute the bifurcation diagram for the recurrent map numerically and see that they are qualitatively similar. Following this analogy, one may instantly relate the first bifurcating point with the EoS minima that the trajectory eventually converges to, and the non-bifurcating regime for the polynomial maps with the "sub-EoS regime" on the left (in Fig. 6b) of the EoS minima.

## 6 CONNECTION TO REAL-WORLD MODELS

In this section we show how the degree-4 model analyzed above resembles the converging dynamics of *over-parameterized* regression models trained on real-world dataset. We train a 5-layer ELU-activated fully connected network on a 2-class small subset of CIFAR-10 (Krizhevsky et al., 2009) with GD. The loss converges to 0 and the sharpness converges to just slightly below $2/\eta$.

We visualize the dynamics by projecting the trajectory onto the subspace spanned by the top eigenvector of minimum (oscillation direction) and the movement direction of parameters orthogonal to oscillation (see Definition 4 in Appendix A.6.1 for exact characterization). As shown in Fig. 7 (mid), after some initial bifurcation-like oscillation, the 2-step trajectory stabilizes and moves along some smooth curves toward the minimum. Near the minimum (Fig. 7, right), the trajectory in fact lies mostly in this 2-dimensional subspace (see Fig. 23c in Appendix) and can be very well-captured by a parabola, which is very similar to our minimalist example. More experimental results on real-world models are available in Appendix A.6.

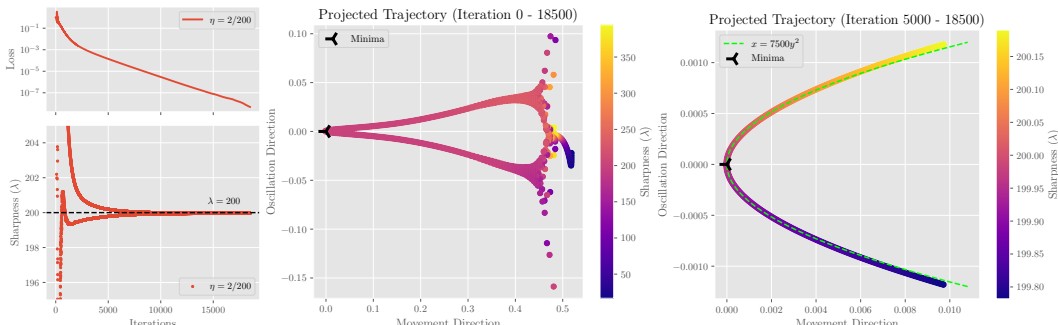

Figure 7: **Training trajectory of 5-layer ELU-activated FC Network**. We train the model using with $\eta = 0.01$ for 18500 iterations. The sharpness converges to 199.97 while $2/\eta = 200$. The local trajectory (right) can be very well approximated by the parabola $x = 7500y^2$.

## 7 DISCUSSION AND CONCLUSION

In this paper we proposed a simple degree-4 model that captures the sharpness adaptivity and sharpness concentration phenomena that happen in gradient descent training of deep neural networks. The simplicity of the model allowed us to perform rigorous analysis on the training dynamics for a large local region. The analysis gives new insights on why the training dynamics of the degree-4 model is inherently different from the training dynamics of degree-2 models. Finally we show that the over-paramterized deep networks trained on real data exhibits a similar parabolic converging trajectory as the scalar example. We hope many of these observations can be generalized to highlight the difference between training dynamics of deeper networks and the shallower models.

There are still many open problems. Can we identify the hidden dynamics of the real world model that yields the parabolic converging trajectory? Can we theoretically understand the automatic coupling of small entries as discussed in Appendix A.4? Is there a way to understand and leverage the fractal/bifurcation behavior in Section 5 toward global dynamics analysis?

## 8 ACKNOWLEDGMENTS

This work is supported by NSF Award DMS-2031849, CCF-1845171 (CAREER), CCF-1934964 (Tripods) and a Sloan Research Fellowship.

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

# Supplementary Materials for Understanding Edge-of-Stability
# Training Dynamics with a Minimalist Example

## A    ADDITIONAL EXPERIMENTS

In this section, we provide more empirical evidences supporting the main text.

In Appendix A.1, we will first introduce our experiment setup including the structure and initialization for the neural network models as well as the generative model for the synthetic datasets.

In Appendix A.2, we will present some additional figures demonstrating the training dynamics near the EoS minima for the degree-2 and degree-4 examples discussed in Section 3 and Section 4. We will also discuss a degree-3 example exhibiting different behavior around different EoS minima. We will explain the phenomenon using our understanding of the 2 and degree-4 models.

In Appendix A.3, we will provide additional empirical evidence that shallow neural networks usually does not converge to the exact EoS threshold.

In Appendix A.4, we will present some results on the training dynamics of scalar networks without the coupling initialization. We will empirically show that the coupling of entries will arise along the training process.

In Appendix A.5, we will demonstrate the EoS phenomenon on the rank-1 factorization.

In Appendix A.6, we will introduce the experiment on learning real-world images (as presented in Section 6) in more detail. We will also present additional experiment results on networks with different activations and local trajectory with perturbation.

In Appendix A.7, we will present some experiments on the edge of stability phenomenon when the model is optimized with stochastic gradient descent.

### A.1    EXPERIMENT SETTINGS

#### A.1.1    CALCULATION OF NUMERICAL SHARPNESS

For the scalar network examples the closed-form Hessian is simple. We compute the exact parameter Hessian and use numerical packages to compute its top eigenvalue.

For neural networks, we use the `PyHessian` package by (Yao et al., 2020), which compute the top eigenvector eigenvalue pair by inferencing the Hessian vector product and do power iteration. For all numerical sharpness computed for neural networks, we set `tol=1e-6` and `max_iter=10000`.

#### A.1.2    SYNTHETIC EXPERIMENTS

For all experiments involving neural networks on synthetic datasets (Fig. 1a, Fig. 1b, Fig. 15), we use fully connected networks with the same dimension for input, output, and all hidden layers. The bias of all layers are fixed to 0. Formally, a $L$-layer width $d$ network can be modeled by $f : \mathbb{R}^d \to \mathbb{R}^d$ such that for input vector $\boldsymbol{x} \in \mathbb{R}^d$,

$$f(\boldsymbol{x}) = \boldsymbol{W}_L \sigma(\boldsymbol{W}_{L-1} \ldots \sigma(\boldsymbol{W}_2 \sigma(\boldsymbol{W}_1 \boldsymbol{x})) \ldots ) \tag{13}$$

where $\sigma : \mathbb{R}^d \to \mathbb{R}^d$ is some entry-wise activation function and $\boldsymbol{W}_l \in \mathbb{R}^{d \times d}$ for all $l \in [L]$. For this paper we only considered $\sigma$ being the ReLU activation $\sigma(x) = x\mathbf{1}_{x \geq 0}$ or the identity $\sigma(x) = x$.

**Initialization of Neural Networks**

We use Xavier initialization (Glorot & Bengio, 2010) with gain of 1 to initialize the all weight matrices. For shallow two-layer networks that will not enter the EoS regime if using completely random initialization, we will asymmetrically re-scale the layers after random initialization by multiplying a constant to all entries of the same layer. When we present results for the re-scaled experiments, we will state the re-scale factor.

**Synthetic Dataset and Loss Function**

For the experiments involving neural networks, we use synthetic datasets very similar to the linear network experiment in section L.3 of Cohen et al. (2021). For a neural network as described above with dimension $d$, we consider the problem of mapping $n$ inputs $\boldsymbol{x}_1, \ldots, \boldsymbol{x}_n \in \mathbb{R}^d$ to $n$ outputs $\boldsymbol{y}_1, \ldots, \boldsymbol{y}_n \in \mathbb{R}^d$. Let $\boldsymbol{X} \in \mathbb{R}^{d \times d}$ and $\boldsymbol{Y} \in \mathbb{R}^{d \times d}$ denote the vertically stack inputs and outputs respectively (Here $n = d$). We generate $\boldsymbol{X}$ as a whitened matrix such that $\boldsymbol{X}\boldsymbol{X}^T = d\boldsymbol{I}_d$ and generate $Y$ by $\boldsymbol{Y} = \boldsymbol{X}\boldsymbol{A}$ where $\boldsymbol{A} = \mathrm{diag}(1, \frac{d-1}{d}, \ldots, \frac{2}{d}, \frac{1}{d})$.

For all experiments with neural networks on synthetic datasets, we consider the simple squared loss

$$\frac{1}{n} \sum_{i=1}^{n} \|f(\boldsymbol{x}_i) - \boldsymbol{y}_i\|_2^2,$$ (14)

which matches our analysis on scalar networks.

### A.1.3 REAL-WORLD DATA EXPERIMENTS

Here we provide the detailed setting for the experiment results shown in Section 6 in the main text as well as Appendix A.6. We consider a binary classification problem on a subset of CIFAR-10 image classification dataset (Krizhevsky et al., 2009).

**Dataset**

To study the training process in an over-parameterized setting (in which the loss can converge close to 0), we take a binary 50-sample subset from CIFAR-10 containing the first 25 samples of class 0 (airplane) and class 1 (automobile). Then we label samples from class 0 by -1 and samples from class 1 by +1.

Here we are consider a binary classification problem since for networks with output dimension larger than 2, there is typically not a strong eigengap between the first eigenvalue and the other eigenvalues (Sagun et al., 2016; Papyan, 2018; Wu et al., 2020). The dynamics with multiple eigenvalues around the stability threshold may exhibits cascading oscillation along different eigendirections (Ruiz-Garcia et al., 2021), and could be complicated to analyze.

**Network Structure**

We conduct the experiment on fully-connected neural networks with four hidden layers of width 200. We consider tanh and ELU as activations. In Table 1 we provide the structure of a fully-connected ELU-activated architecture. This architecture follows the experiments in Li et al. (2022b).

Table 1: Structure of fully-connected network

| # | Name | Module | In Shape | Out Shape |
|---|------|--------|----------|-----------|
| 1 | | `Flatten()` | (32,32,3) | 3072 |
| 2 | fc1 | `nn.Linear(3072, 200, bias=False)` | 3072 | 200 |
| 3 | | `nn.ELU()` | 200 | 200 |
| 4 | fc2 | `nn.Linear(200, 200, bias=False)` | 200 | 200 |
| 5 | | `nn.ELU()` | 200 | 200 |
| 6 | fc3 | `nn.Linear(200, 200, bias=False)` | 200 | 200 |
| 7 | | `nn.ELU()` | 200 | 200 |
| 8 | fc4 | `nn.Linear(200, 200, bias=False)` | 200 | 200 |
| 9 | | `nn.ELU()` | 200 | 200 |
| 10 | fc5 | `nn.Linear(200, 1, bias=False)` | 200 | 1 |

**Loss Function** For the CIFAR-10 subset experiment $\{(\boldsymbol{x}_1, y_i)\}_{i=1}^{n}$ where $n = 50$, $\boldsymbol{x}_i \in \mathbb{R}^{3072}$, and $y_i \in \{1, -1\}$, we consider the mean squared loss

$$\frac{1}{n} \sum_{i=1}^{n} \|f(\boldsymbol{x}_i) - y_i\|_2^2,$$ (15)

which matches our theoretical analysis on the scalar example.

## A.2 Additional Experiments for Scalar Network Examples

In this section we show some additional figures demonstrating the local training dynamics and convergence boundary for the two cases we analyzed in Section 3 and Section 4.

### A.2.1 4-Layer Scalar Network

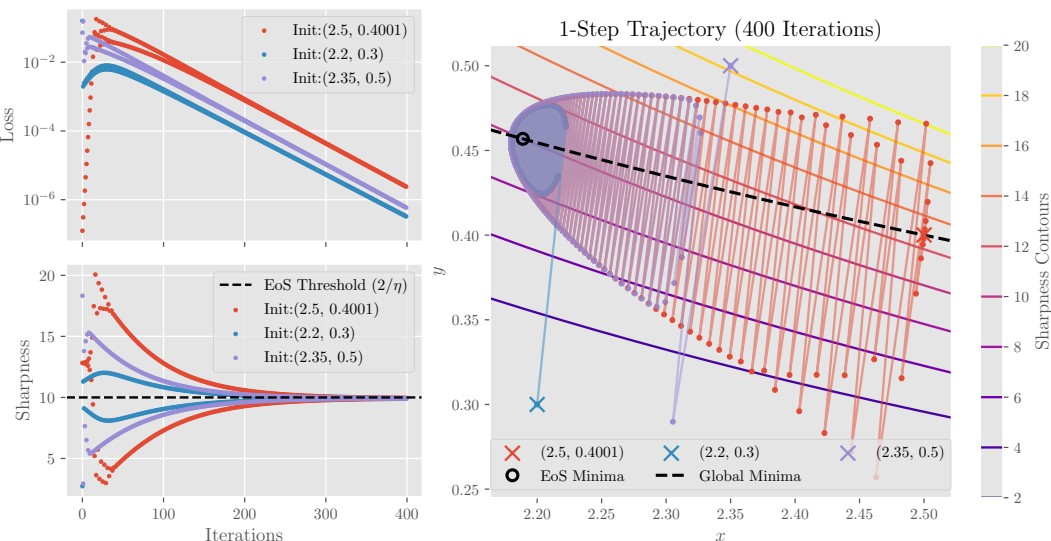

Figure 8: **Sharpness concentration for the degree-4 example.** We run GD with $\eta = 0.2$ from 3 initializations that are above, below, and very close to the line of global minima. The sharpness of all trajectories converges to around $2/\eta$ while the loss decreases exponentially. In the 2D trajectory, the 2-step movement quickly converges to the parabolic curves ending very close to the EoS minimum.

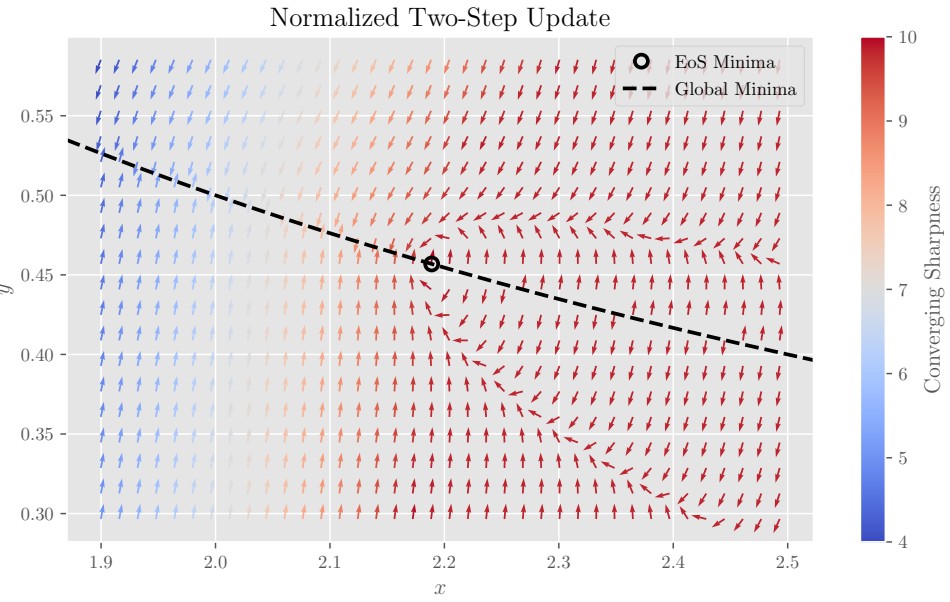

Figure 9: **Local two-step movement for the degree-4 example.** We record the two-step movement with $\eta = 0.2$ from a grid of initializations near the EoS minima. At each point, the arrow points toward the direction of two-step movement from that point and the color of arrow indicates the converging sharpness of trajectories passing that point. We can see the parabolic trajectory and how initialization to the right of the EoS minima all tend to converge to it.

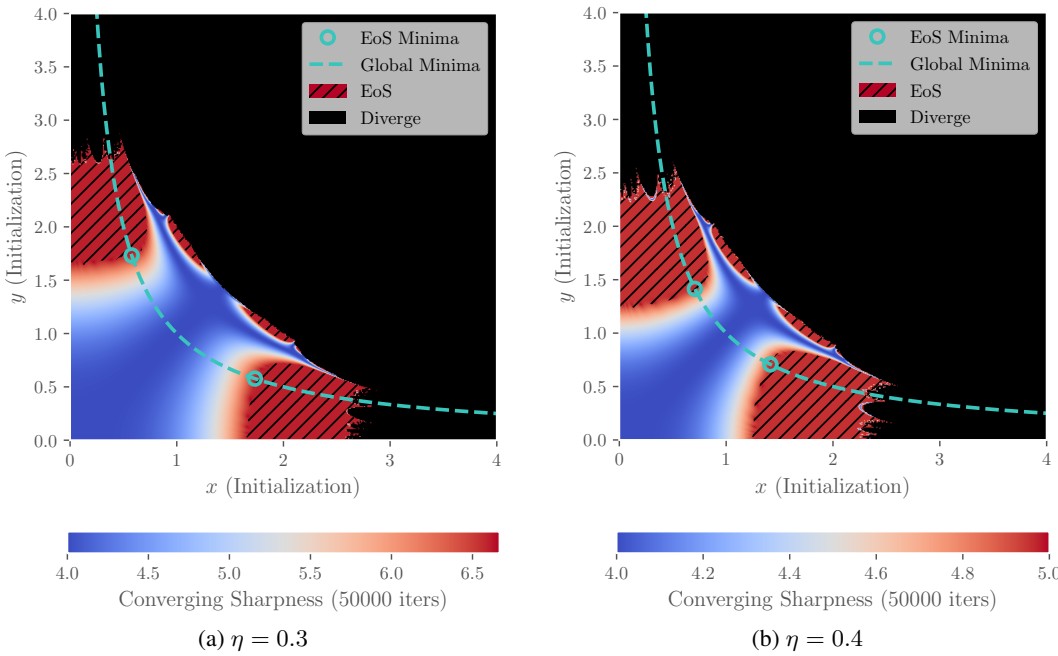

(a) $\eta = 0.3$                    (b) $\eta = 0.4$

Figure 10: **Converging Sharpness of Initializations Under Different Step Sizes**. Please see the caption of Fig. 2b for detailed description.

### A.2.2  2-LAYER SCALAR NETWORK

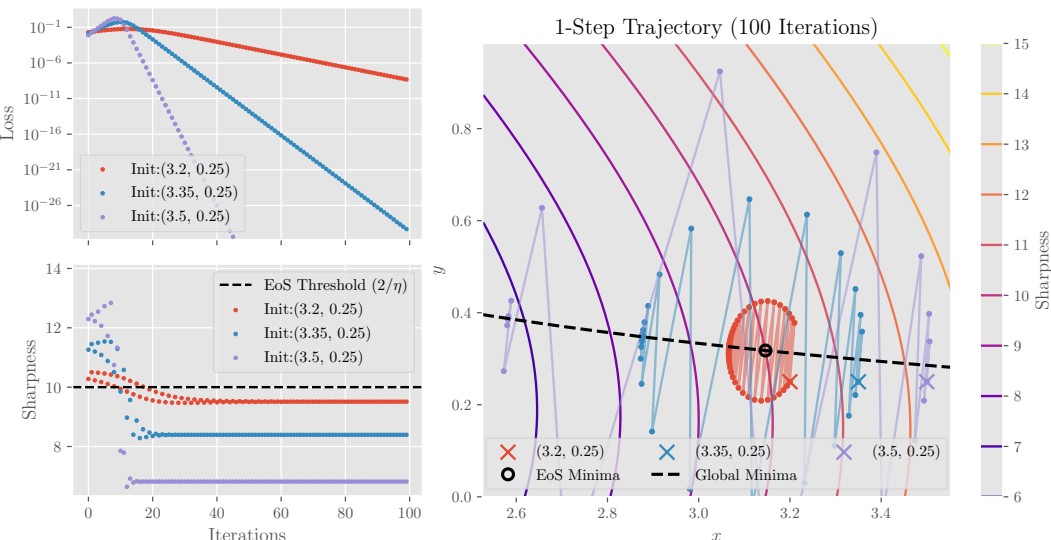

Figure 11: **Beyond EoS training on product of two scalars.** We run the identical experiment as in Fig. 8 except for the objective $\min_{x,y \in \mathbb{R}}(1 - xy)^2$. Note that in this case we do not observe both sharpness concentration and sharpness adaptivity. The two-step trajectory follows an elliptical trajectory centered at the EoS minima. In this context, a sharper initialization (e.g. the purple curve) will eventually converge to a flatter minima (as shown in the left figure).

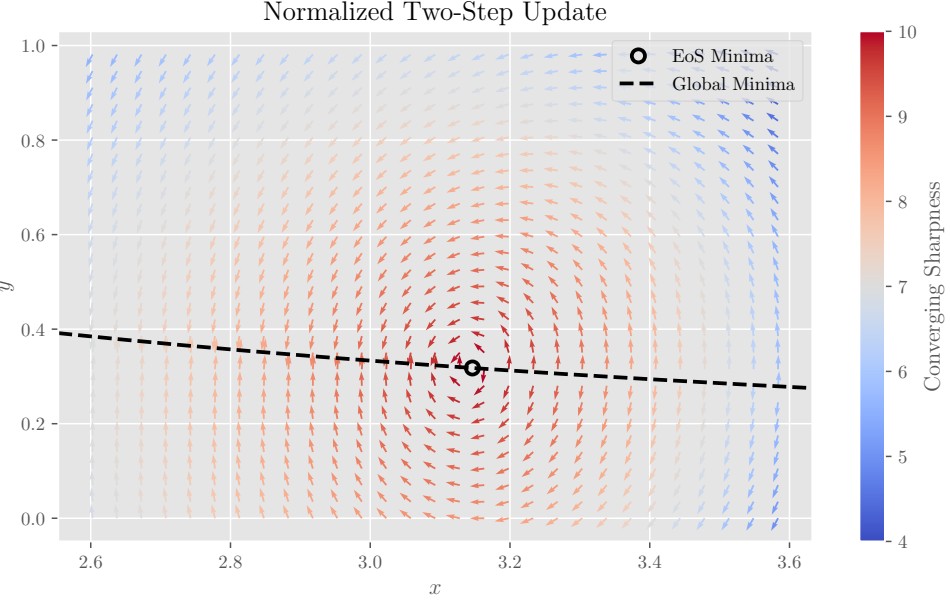

Figure 12: **Local two-step movement for the degree-2 example.** This is the same figure as Fig. 9 except for the degree-2 example. We use the same step size $\eta = 0.2$. There is no longer the concentration behavior as we see for the degree-4 case. Locally, only the initialization very close to the EoS minimum will converge to a sharpness near the stability threshold (which is 10 with $\eta = 0.2$).

### A.2.3    3-Layer Scalar Network

Now we look into an interesting example with different behaviors around different EoS minima.

We consider a 3-layer scalar network with objective

$$\min_{x,y,z\in\mathbb{R}} \frac{1}{2}(1-xyz)^2. \tag{16}$$

To make the dynamics two dimensional, we consider the initialization with $z = y$. The equality of the last two entries will be preserved through training so the dynamics is two dimensional in terms of $x$ and $y$. In the positive quadrant, the global minima is $\sqrt{x}y = 1$ and there are two EoS minima.

In Fig. 13, we plot the converging sharpness from different initializations in comparison with Fig. 2b and Fig. 5b in the main text. Around the EoS minimum that the single entry $x$ is small and the duplicated entries $y$ are large (upper left of Fig. 13), the behavior is similar to the 2 scalar case (Fig. 5b) with no sharpness concentration. Around the EoS minima with large single entry and small duplicating entries (lower right of Fig. 13), we have a region of initialization (the red shaded area) with sharpness concentration similar to the 4 scalar case (Fig. 2b).

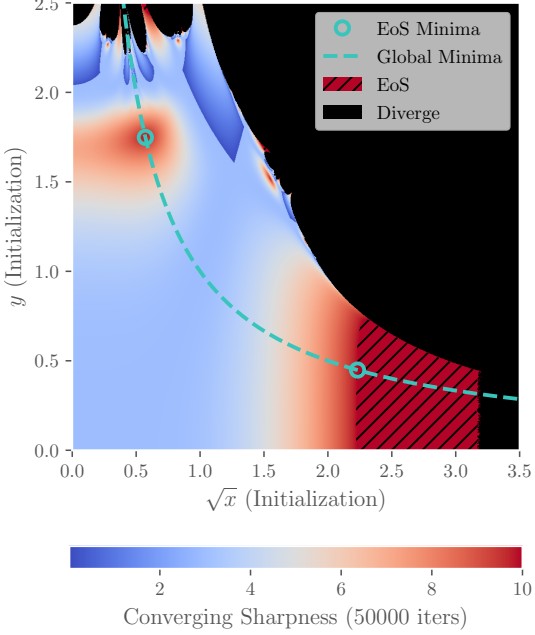

Figure 13: Converging sharpness of $(x, y, y)$ parameterized initializations ($\eta = 0.2$).

A heuristic explanation to this difference lies in the difference in the degree of the small entries. Around the EoS minima that the single entry is small, the local two-step approximation is similar to Eq. (12) and gives us elliptical two-step trajectories. Around the minima with small duplicating entries, the two-step approximation would contain the cubic term as in Eq. (7), which gives us both sharpness concentration and adaptivity.

Such heuristics can be further verified by visualizing the local dynamics around the minima. As shown in Fig. 14, the local dynamics around the minima with small duplicating entries is similar to the case of the degree-4 example with convergence toward a parabolic trajectory. On the other hand, the local dynamics around the minima with only one small entry is similar to the case of the degree-2 example where parameters follow an locally elliptic trajectory centered at the EoS minima.

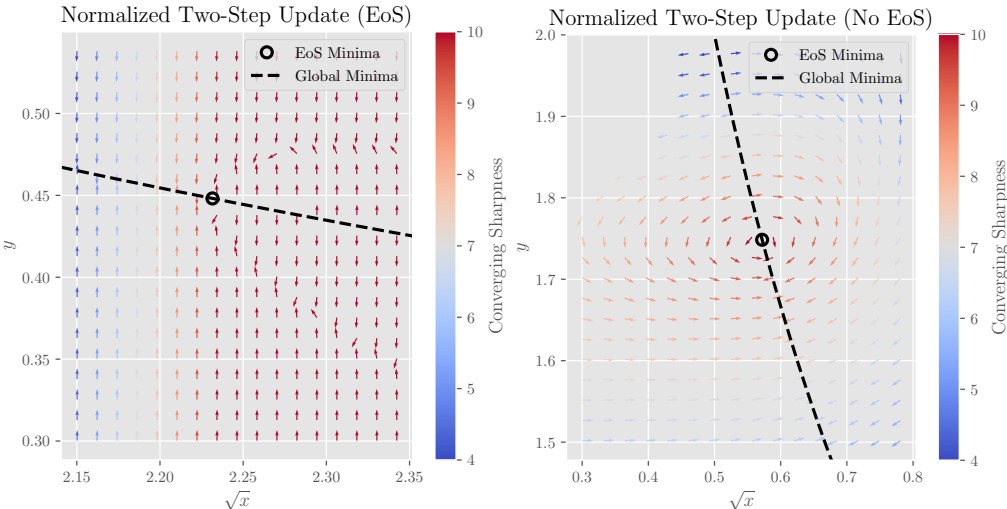

Figure 14: **Local two-step movement for the degree-3 example.** In this figure, we show the local dynamics near the two EoS minima in the positive quadrant of the degree-3 example. The **left** figure corresponds to the EoS minima at the lower right of Fig. 13. At this EoS minima, the duplicated entry $y$ is small, and the local behavior is very similar to the case of degree-4 example (Fig. 9) for which we have provable sharpness concentration. The **right** figure corresponds to the EoS minima at the top left of Fig. 13. The local behavior is very similar to the case of degree-2 example (Fig. 12), for which we do not have EoS behaviors.

## A.3 ADDITIONAL EXPERIMENTS FOR 2-COMPONENT SCALAR FACTORIZATION

In this section we present the experiment results for 2-component scalar factorization deferred from Section 4. The dynamics as shown in Fig. 15 is very similar to the degree-2 example in Fig. 11.

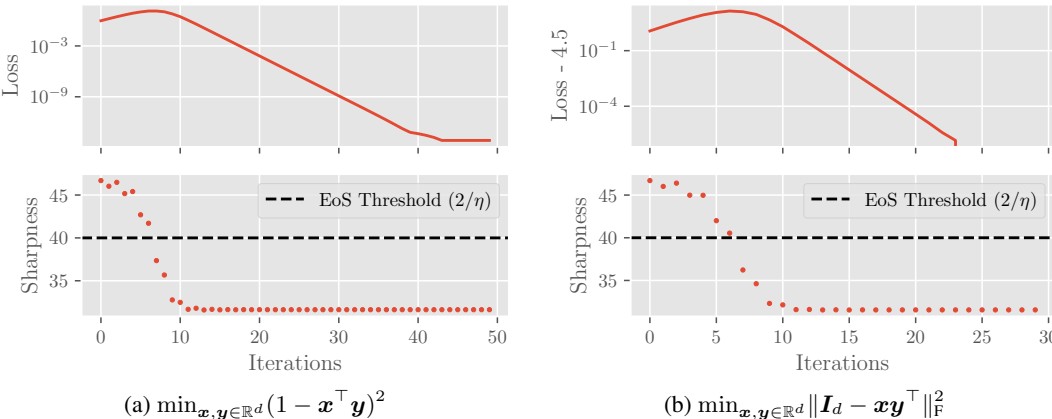

(a) $\min_{\boldsymbol{x},\boldsymbol{y}\in\mathbb{R}^d}(1-\boldsymbol{x}^\top\boldsymbol{y})^2$

(b) $\min_{\boldsymbol{x},\boldsymbol{y}\in\mathbb{R}^d}\|\boldsymbol{I}_d-\boldsymbol{x}\boldsymbol{y}^\top\|_{\mathrm{F}}^2$

Figure 15: **GD on 2-component scalar factorization problems.** We run gradient descent with $\eta=0.05$ for two different objectives. Both models are asymmetrically initialized with factor $(5,0.1)$ so that they have an initial sharpness larger than $2/\eta$. For both cases, the converging sharpness is distinguishably smaller than the stability threshold.

### A.4 Experiments for General Scalar Networks

In this section, we will present some empirical observations on training dynamics of more general scalar networks related to the sharpness concentration and adaptation phenomena. A $n$-layer scalar network is defined to be the model parameterized by $n$ entries $x_1, \ldots, x_n \in \mathbb{R}$ with objective

$$\mathcal{L}(x_1, \ldots, x_n) \triangleq \frac{1}{2}\left(1 - \prod_{i=1}^{n} x_i\right). \tag{17}$$

#### A.4.1 Initialization without Duplicated Entries

We first consider a variant of the degree-3 example as discussed in Appendix A.2.3. In particular, we initialize the two small entries differently and record their values throughout the training trajectory.

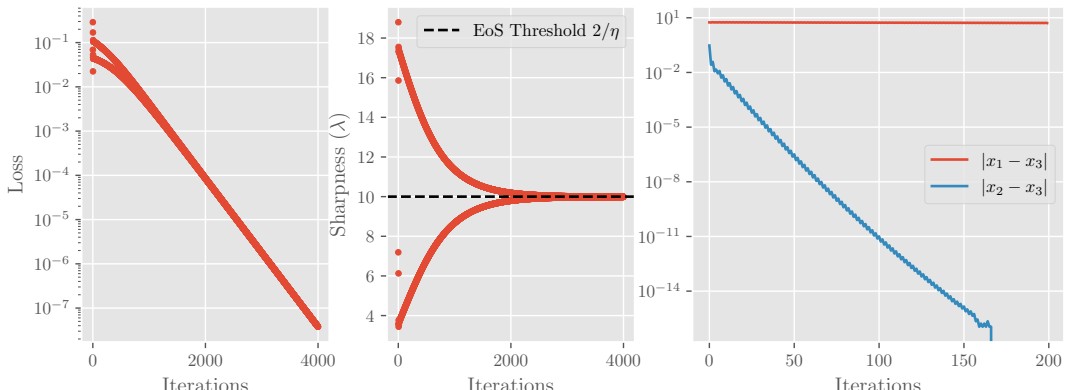

Figure 16: This figure records the training loss (**left**) and sharpness (**middle**) of a degree-3 scalar network with initialization $(6, 0.1, 0.4)$ optimized by gradient descent with fixed step size $\eta = 0.2$. In (**right**) we plot the distance of the last entry $x_3$ to other entries.

As we can see in Fig. 16, at the very beginning of the training, the second entry $x_2$ converges to $x_3$ geometrically, then the dynamics is reduced to the case of duplicated entries, which we know that the sharpness concentration behavior would happen for sufficiently asymmetric initialization. In Fig. 17 we consider a 7-layer scalar network with 3 different large entries and 4 different small entries. We observe similar behavior as $x_4, x_5, x_6$ all converges to $x_7$ geometrically, and we observe concentration of sharpness with 4 duplicated small entries.

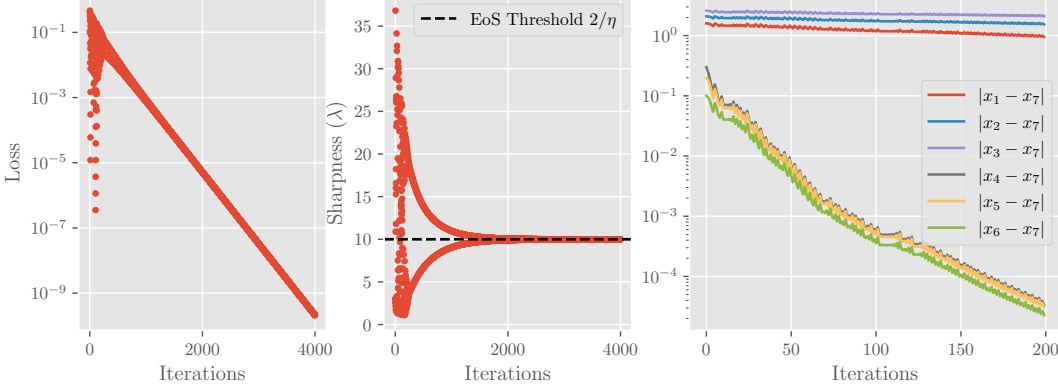

Figure 17: This figure records the training loss (**left**) and sharpness (**middle**) of a 7-layer scalar network with initialization $(2, 2.5, 3, 0.1, 0.2, 0.3, 0.4)$ optimized by gradient descent with fixed step size $\eta = 0.2$. In (**right**) we plot the distance of the last entry $x_7$ to other entries.

To probe into the detailed training dynamics of general scalar networks, we plot the pairwise dynamics of the entries as shown in Fig. 18.

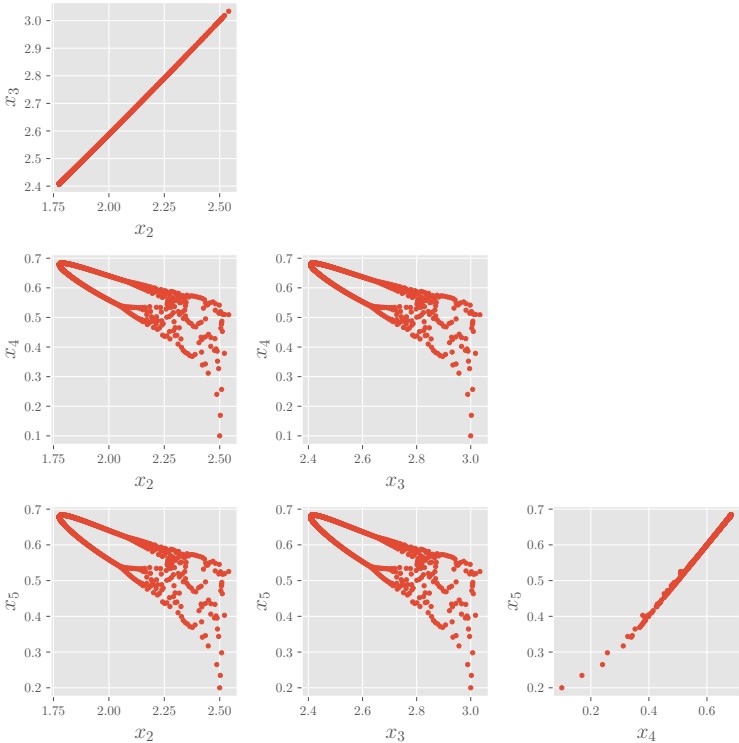

Figure 18: **Pairwise Training Dynamics.** This set of figures record the pairwise training dynamics for entries $x_2, x_3, x_4, x_5$ in the same experiment as Fig. 17. $x_2$ and $x_3$ are initialized large while $x_4$ and $x_5$ are initialized small. We see that within the small entries and the large entries, the pairwise dynamics are all approximately linear while the cross comparisons across the small entries and large entries gives the parabolic two-step trajectory (and also some bifurcation behavior).

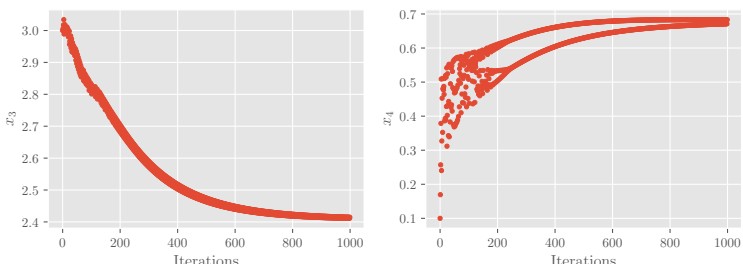

Figure 19: **Single Entry Movement.** We plot the value of $x_3$ (initialized to 3) and $x_3$ (initialized to 0.1) along the training process. The larger entry decreases in an approximately monotone manner and the small entry increase while oscillating.

### A.4.2    ON "LARGE" AND "SMALL" INITIALIZATIONS

In the experiments shown above, we have seen that there are mainly two classes of behaviors for the entries: the entries that were initialized to be large moves slowly with little oscillation while the entries that were initialized to be small has significant oscillation along the trajectory. Intuitively, it is the decreasing large entry that decreases the sharpness and stabilizes the oscillating small entries and result in the final convergence close to the EoS minimum. A natural question to ask is whether there exists a clear boundary separating the "small" and "large" entries.

We consider a 4-layer scalar network with initialization $(6, 0.7, 0.3, 0.2)$ optimized with GD with step size $\eta = 0.2$. In Fig. 20 and Fig. 21 we visualize the training loss, sharpness, and the pairwise dynamics. The mixed behaviors suggests a clear boundary between the "large" and "small" entries does not exists, and the complexity of this problem is beyond this simple heuristics.

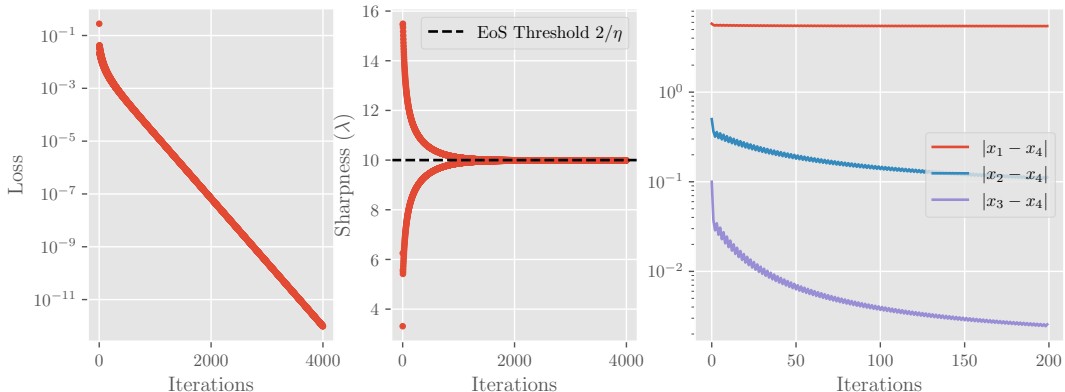

Figure 20: This figure records the training loss (**left**) and sharpness (**middle**) of a 4-layer scalar network with initialization $(6, 0.7, 0.3, 0.2)$ optimized by gradient descent with fixed step size $\eta = 0.2$. In (**right**) we plot the distance of the last entry $x_4$ to other entries. In this example, the small entries did not converge to be exactly the same value, yet the loss still decreased geometrically and the sharpness concentration phenomenon still occurred.

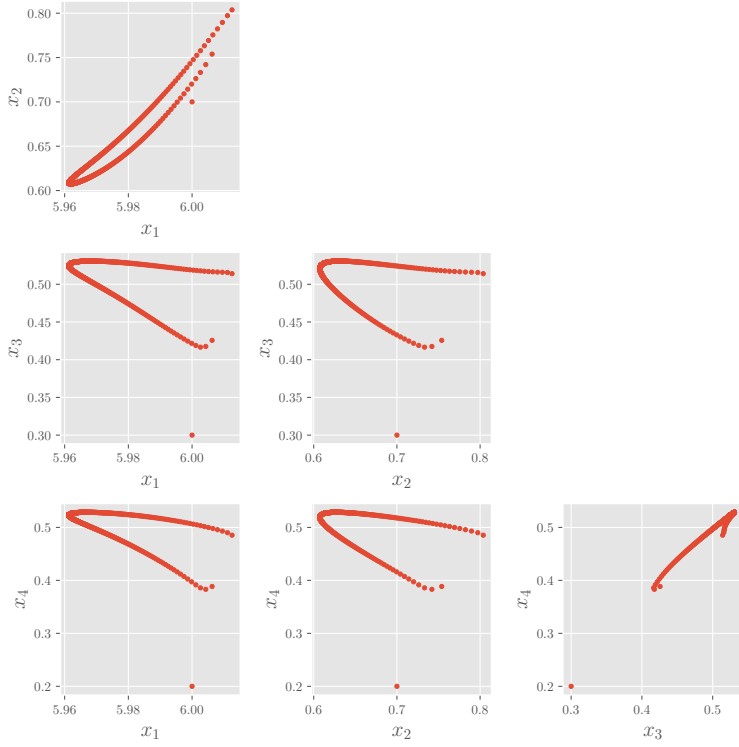

Figure 21: **Pairwise Training Dynamics.** This set of figures record the pairwise training dynamics for entries $x_1, x_2, x_3, x_4$ in the same experiment as Fig. 20. $x_1$ was initialized large (at 6), $x_3, x_4$ were initialized small (at 0.2, 0.3), and $x_2$ was initialized moderately small (at 0.7). We see that the two-step pairwise dynamics between $x_2$ and $x_3$ roughly follows a parabolic trajectory, yet the pairwise dynamics between $x_1$ and $x_2$ also exhibits similar features while still following a roughly linear relation.

## A.5 Additional Experiments for Rank-1 Factorization of Isotropic Matrix

In this section, we will demonstrate the EoS phenomenon on the rank-1 factorization of isotropic matrix in Section 3.3.

We first show that the loss, the sharpness and the trajectory of GD is very similar to the degree-4 scalar network case. For each different learning rate, the sharpness concentrates to a tiny interval close to the stability threshold $2/\eta$ when trained with gradient descent.

Also, we consider the 2D trajectory of the two vectors $\boldsymbol{x}, \boldsymbol{y}$. We plot the the trajectory in the norm of each vector, i.e. $\|\boldsymbol{x}\|\|\boldsymbol{y}\|$, and get a similar figure to the scalar case. Actually, we can prove that the dynamics of this training objective will eventually be reduced to our degree-4 scalar network. That is because all the global minima of this optimization problem requires that $\boldsymbol{x}$ is aligned with $\boldsymbol{y}$, i.e. $\boldsymbol{x} = c\boldsymbol{y}$. After the two vectors are aligned, the training dynamics of $\|x\|, \|y\|$ will be exactly equivalent to those of the scalar network.

The following figure shows how GD enters EoS on the rank-1 factorization problem, and how fast the alignment of the two vectors is achieved. Here we consider an alignment indicator $\|\boldsymbol{x}\|^2\|\boldsymbol{y}\|^2 - (\boldsymbol{x}^\top\boldsymbol{y})^2$ showing how the two vectors are aligned. If $\boldsymbol{x}$ is parallel to $\boldsymbol{y}$, i.e. $\boldsymbol{x} = c\boldsymbol{y}$, then the variable becomes 0. Detailed analysis for this problem is deferred to Appendix C.

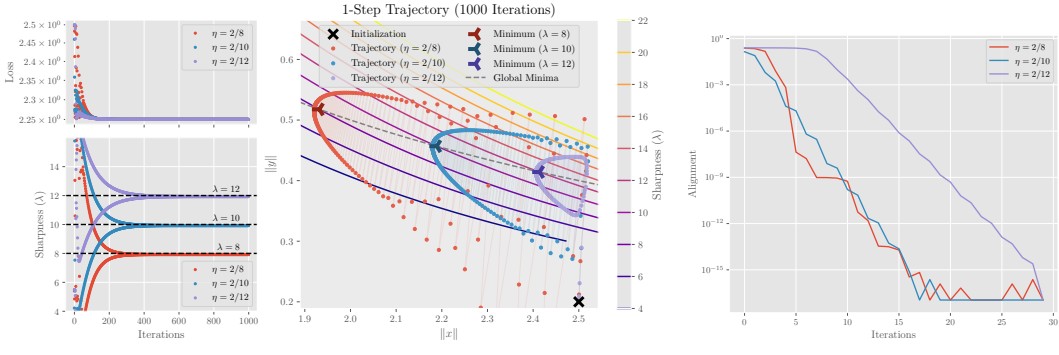

(a) Evolution of training loss, sharpness and the trajectory of GD on the rank-1 factorization of isotropic matrix.

(b) Evolution of the alignment indicator $\|\boldsymbol{x}\|^2\|\boldsymbol{y}\|^2 - (\boldsymbol{x}^\top\boldsymbol{y})^2$.

Figure 22: **EoS phenomenon on the rank-1 factorization of isotropic matrix.** In **(a)**, similar to the degree-4 scalar case, we demonstrate sharpness adaptivity by running GD with learning rate $\eta = \frac{2}{8}, \frac{2}{10}, \frac{2}{12}$ from the same initialization ($\|\boldsymbol{x}_0\|\|\boldsymbol{y}_0\| = 1/2$). All the sharpness of each trajectory converges to around their corresponding stability threshold $2/\eta$ while the loss decreases exponentially. In the 2D trajectory, GD quickly converges along some smooth curves ending near the EoS minimum. In **(b)**, we see that in the first 30 iterations, the alignment indicator ($\|\boldsymbol{x}\|^2\|\boldsymbol{y}\|^2 - (\boldsymbol{x}^\top\boldsymbol{y})^2$) decreases geometrically, and stabilizes at its numerical minimal value. Theoretically, we characterizes the geometric decay for the alignment variable by Lemma 20 in Appendix C.

## A.6 EoS Convergence for Deep Neural Networks Trained on Real Data

In this section, we present a more comprehensive description and additional results for the experiment of learning a 2-class small subset of CIFAR-10 with 50 images in an *over-parameterized* regression setting. The details for the network structures and dataset construction are available in Appendix A.1.3.

In this experiment, we train two 5-layer fully connected (fc) networks of width 200 with ELU and tanh activation using (full-batch) gradient descent on the binary dataset with mean squared loss. We chose these two activation functions following the empirical experiments in Cohen et al. (2021). ReLU is not being used since its training dynamics as the loss converges to 0 is very unstable.

We record the training loss and sharpness of the two training processes. To better visualize the training trajectory, we consider the following projection mechanism.

A.6.1 TRAJECTORY PROJECTION

Inspired by the observation on the scalar example, we note that the dynamics toward the end of the convergence has two distinctive directions: an "oscillation direction" which is aligned with the first eigenvector of the Hessian, and an "movement direction" which the 2-step average of the model moves along and converges to the final minimum.

In the context of our $(a, b)$-reparameterization for theoretical analysis (Definition 2), the oscillation direction corresponds to $b$ and the movement direction corresponds to $a$. In a local region around the converging minima, $(a, b)$ constitutes a parabolic trajectory that can be well captured by the solution of the ODE in Eq. (8). In a high-dimensional setting, the oscillation direction is still naturally the top eigenvector at minimum, but we have to manually pick a movement direction to project onto.

To be concrete, consider a trajectory of the parameters $\{\theta_1, \theta_2, \ldots, \theta_{T-1}, \theta_T\}$, where $\theta_t \in \mathbb{R}^d$ is the parameter vector for the model after the $t$-th iteration. We define the oscillation direction $v_{\text{osc}}$ as the first eigenvector of the parameter Hessian $\boldsymbol{H}(\theta_T)$ and the movement direction $v_{\text{move}}(\hat{t})$ as follows:

**Definition 3** (Movement direction). For some iteration $\hat{t} \in [T]$, define $v_{\text{move}}(\hat{t})$ as

$$v_{\text{move}}(\hat{t}) \triangleq \tfrac{1}{2} \left( \theta_{\hat{t}-1} + \theta_{\hat{t}} \right) - \tfrac{1}{2} \left( \theta_{T-1} + \theta_T \right). \tag{18}$$

Fix some iteration $\hat{t}$, $v_{\text{move}}(\hat{t})$ captures the non-oscillatory movement of the parameters from step $\hat{t}$ to step $T$. We orthonormalize the basis by projecting $v_{\text{osc}}$ off from $v_{\text{move}}(\hat{t})$ and get

$$\begin{aligned}
\tilde{v}_{\text{move}}(\hat{t}) &\triangleq v_{\text{move}}(\hat{t}) - \text{proj}_{v_{\text{move}}(\hat{t})}(v_{\text{osc}}), \\
\bar{v}_{\text{move}}(\hat{t}) &\triangleq \tilde{v}_{\text{move}}(\hat{t}) / \left\| \tilde{v}_{\text{move}}(\hat{t}) \right\|, \\
\bar{v}_{\text{osc}} &\triangleq v_{\text{osc}} / \left\| v_{\text{osc}} \right\|.
\end{aligned} \tag{19}$$

Now with the orthonormal basis, we define the movement-oscillation projection of $\theta_t$ to be the projection of its offset from the minima (which we approximate by the mean of the last two steps in the trajectory) onto $\bar{v}_{\text{move}}(\hat{t})$ and $\bar{v}_{\text{osc}}$.

**Definition 4** (Movement-Oscillation Projection). Fix an iteration $\hat{t}$ for determining the movement direction, the movement-oscillation projection of $\theta_t$ is

$$\left( \bar{v}_{\text{move}}(\hat{t})^\top \left( \theta_t - \tfrac{1}{2} \left( \theta_{T-1} + \theta_T \right) \right), \bar{v}_{\text{osc}}^\top \left( \theta_t - \tfrac{1}{2} \left( \theta_{T-1} + \theta_T \right) \right) \right) \tag{20}$$

When doing the projection in practice (as in Fig. 24 and Fig. 26), we fix $\hat{t} = 5000$, which is when the 2-step trajectory becomes relatively stable. We also record the norm of the component of the offset $\theta_t - \tfrac{1}{2} \left( \theta_{T-1} + \theta_T \right)$ that is orthogonal to the subspace spanned by $\bar{v}_{\text{move}}(\hat{t})$ and $\bar{v}_{\text{osc}}$. These results are shown in Fig. 23c and Fig. 25c.

A.6.2 ELU-ACTIVATED FULLY CONNECTED NETWORK

Here we present the experiment results for training a 5-layer ELU-activated FC network on the binary subset of CIFAR-10. In Fig. 23, we show the evolution of loss and sharpness along the training process. The sharpness eventually converge to just slightly below the $2/\eta$ threshold. We also observe that the dynamics toward the end of the converging process is mainly happening in the 2-dimensional subspace spanned by the oscillation and movement directions (Fig. 23c).

In Fig. 24 we plot the projected trajectory of the training process. Toward the end of the training process, the trajectory can be very accurately characterized by a parabola and the converging sharpness is just slightly below the stability threshold. This is identical to what we observe (and proved) for the scalar network case.

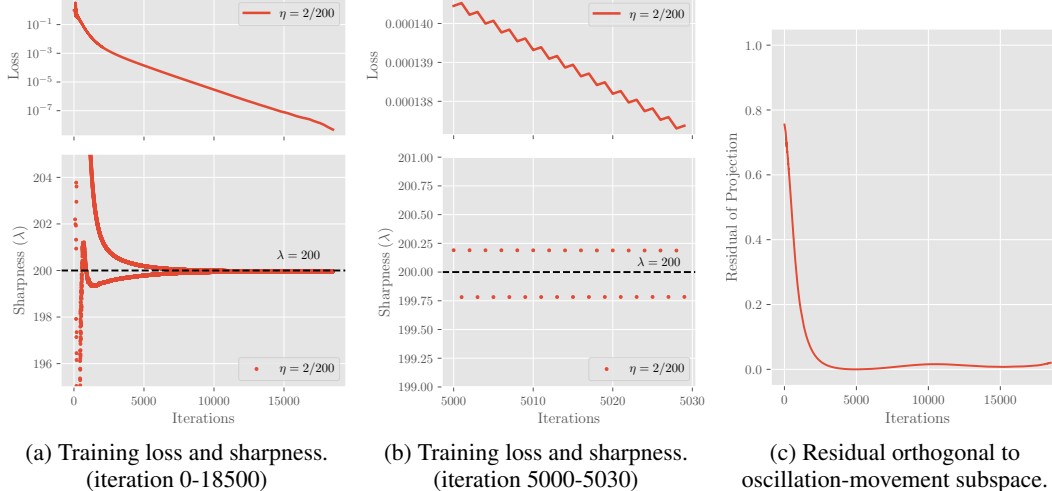

(a) Training loss and sharpness.
(iteration 0-18500)

(b) Training loss and sharpness.
(iteration 5000-5030)

(c) Residual orthogonal to
oscillation-movement subspace.

Figure 23: **Training Statistics for ELU-activated 5-layer FC network.** ($\eta = 0.01$, $2/\eta = 200$)
**(a)** is identical to Fig. 7 (left) in the main text. We see that the model is capable of memorizing all data as the loss decreases exponentially to 0. Toward convergence, the sharpness oscillates very close to the stability threshold and eventually converges to 199.97. In **(b)** we show a section of (a) between iteration 5000 and 5030. We can clearly observe two distinctive features of the EoS regime: the loss decreases non-monotonically and the sharpness oscillates around $2/\eta$. In **(c)** we plot the norm of the offset from minima that is orthogonal to the movement-oscillation projection. After 3000 iterations the residual becomes very small, suggesting that dynamics is mainly happening in the 2 dimensional subspace and hence the projection captures the dynamics quite well.

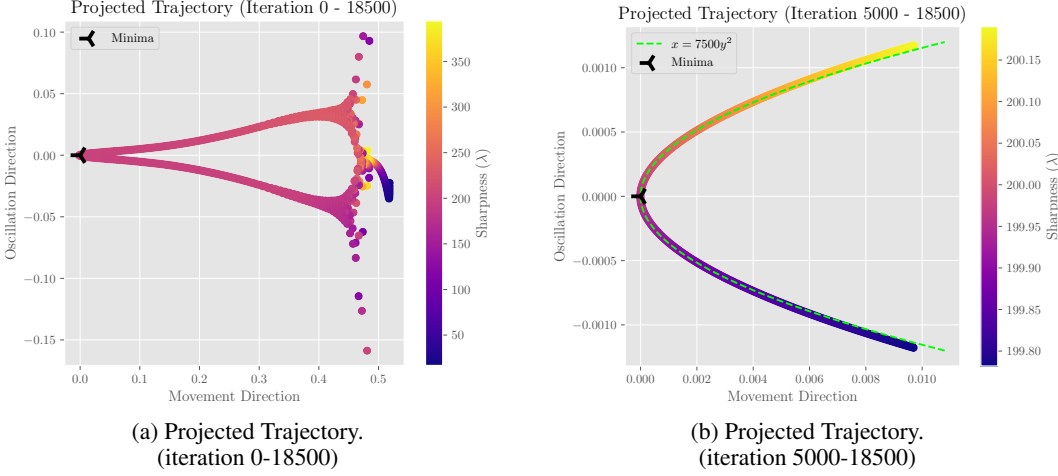

(a) Projected Trajectory.
(iteration 0-18500)

(b) Projected Trajectory.
(iteration 5000-18500)

Figure 24: **Projected Trajectory of ELU-activated 5-layer FC network.** ($\eta = 0.01$, $2/\eta = 200$)
In **(a)**, we plot the projected trajectory for the entire training process. After some large bifurcation like oscillation, the 2-step trajectory quickly stabilizes and moves toward the minimum along the movement direction. In **(b)**, we show the tip of the trajectory, which can be very well captured by a parabola. These figures are identical to Fig. 7 in the main text. The color of the dots reflects the local numerical sharpness.

### A.6.3 TANH-ACTIVATED FULLY CONNECTED NETWORK

Here we show the results for the same experiment on a tanh-activated 5-layer FC network. The phenomena are qualitatively identical to the ELU case described above.

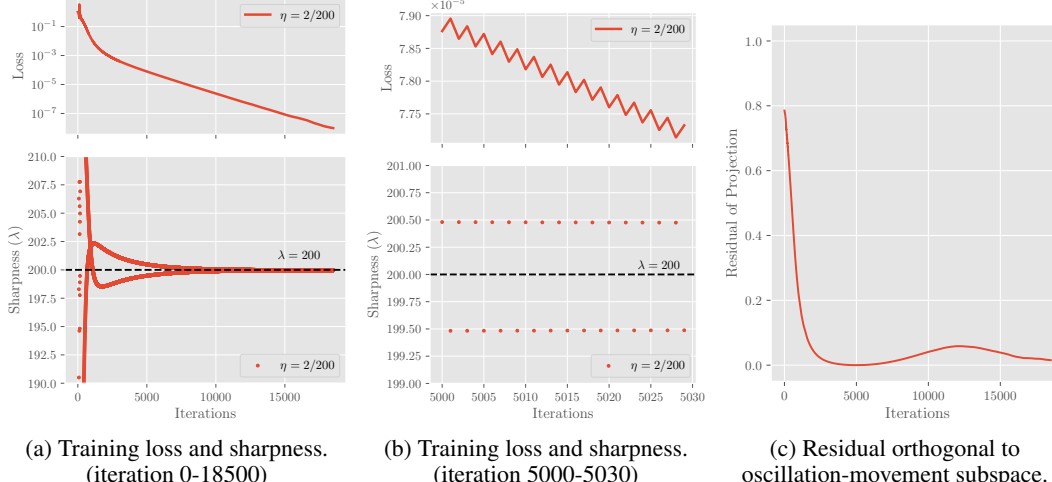

(a) Training loss and sharpness. (iteration 0-18500)

(b) Training loss and sharpness. (iteration 5000-5030)

(c) Residual orthogonal to oscillation-movement subspace.

Figure 25: **Training Statistics for tanh-activated 5-layer FC network.** ($\eta = 0.01, 2/\eta = 200$) Please refer to the caption of Fig. 23 for detailed explanation.

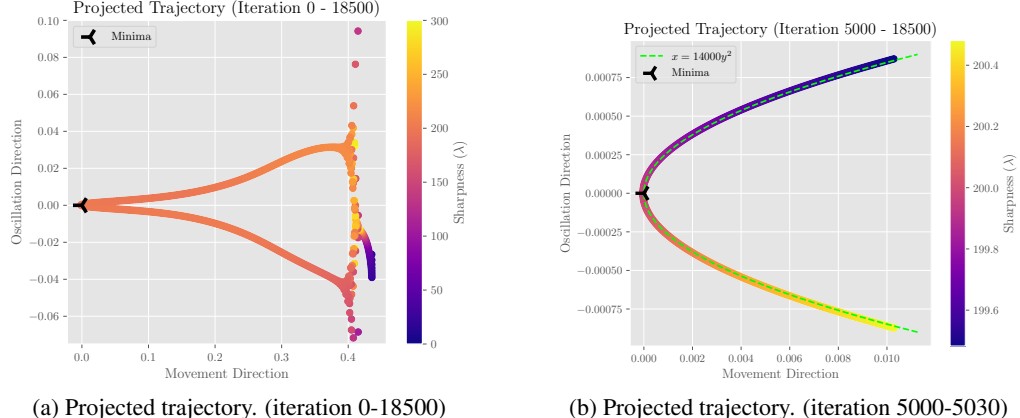

(a) Projected trajectory. (iteration 0-18500)

(b) Projected trajectory. (iteration 5000-5030)

Figure 26: **Projected Trajectory of tanh-activated 5-layer FC network.** ($\eta = 0.01, 2/\eta = 200$) Please refer to the caption of Fig. 24 for detailed explanation.

### A.7 EDGE OF STABILITY AND STOCHASTIC GRADIENT DESCENT

In this section, we will briefly discuss some empirical observations of EoS in stochastic gradient descent (SGD). In Appendix A.7.1, we will first empirically present the effects of different forms of noise on our scalar model. Then in Appendix A.7.2 we will compare it with the observations made on real world models trained with mini-batch gradient descent. Finally, we will discuss the limitations of our scalar model in explaining what people observe about EoS when the model is trained with SGD.

### A.7.1 GD WITH NOISE ON SCALAR NETWORK

We first look into the training trajectory of our degree-4 scalar network example with noise injected to the gradient descent process. We consider *label noise*, which perturbs the target by a small amount per iteration, and *gradient noise*, which perturbs the gradient by a small amount per iteration before updating the parameter according to it.

**Label Noise:**

To simulate the existence of label noise, at each iteration we compute the gradient for the objective

$$\mathcal{L}_{\text{LN}}(x, y, \delta) = \frac{1}{4}(1 + \delta - x^2 y^2)^2$$

where $\delta$ is sampled from a zero-mean Gaussian $\mathcal{N}(0, \sigma^2)$ for each iteration. This is equivalent to adding a perturbation of $\delta$ to the label (which is 1 in our original model). We start from the same initialization as in Fig. 8 and plot the trajectory in Fig. 27.

As shown in Fig. 27a and Fig. 27b, the trajectory first roughly follows a parabolic boundary and reaches close to the set of global minima near the EoS-minimum relatively quickly (for around 200 iterations). This part of the trajectory resembles our analysis for the case without label noise.

After the model reaches the tip of the parabola, the dynamics is mainly dominated by the label noise. The gradient is dominated by its noise component of $\delta xy(y, x)$, which is orthogonal to the manifold of global minima $xy = 1$, thus the model starts oscillating around the global minima. As shown in Fig. 27c and Fig. 27d, the sharpness further decreases very slowly (for $10^6$ iterations) and eventually reaches the flattest global minimum at $(1, 1)$ with sharpness of 4. We believe this is within the regime of the sharpness reduction flow near the manifold of minima, which is comprehensively studied by Damian et al. (2021); Li et al. (2021; 2022b); Lyu et al. (2022).

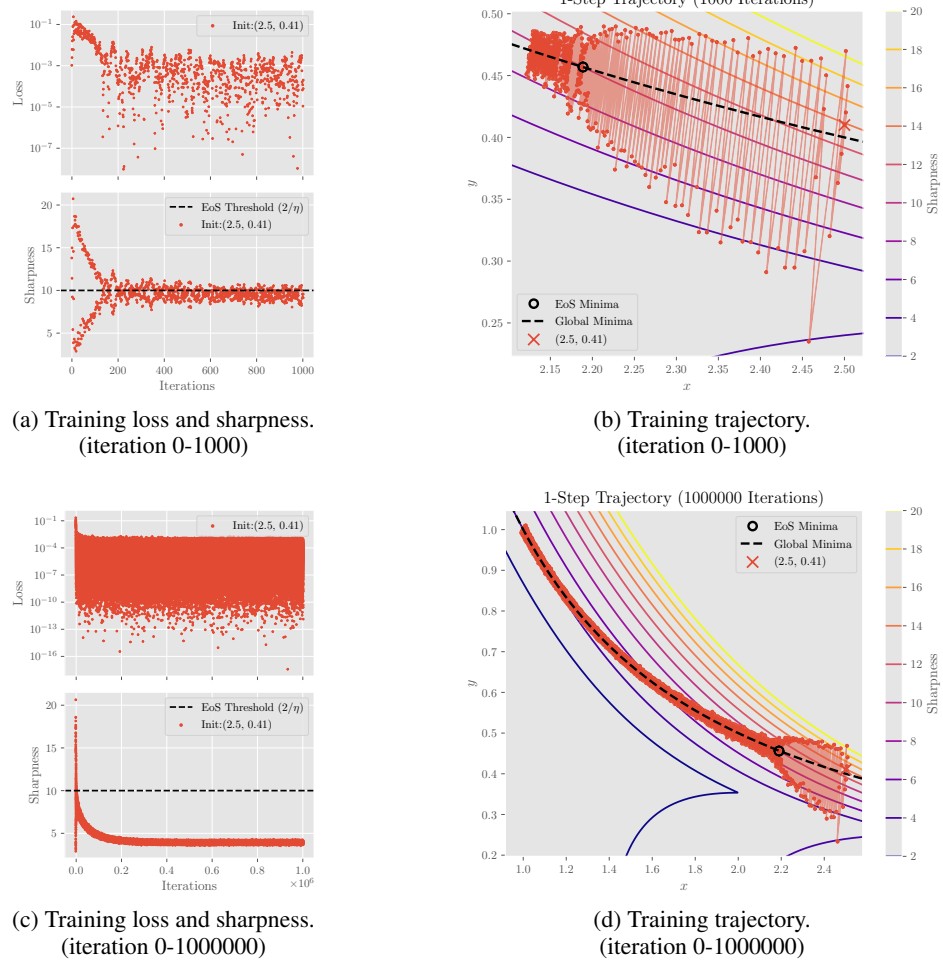

(a) Training loss and sharpness.
(iteration 0-1000)

(b) Training trajectory.
(iteration 0-1000)

(c) Training loss and sharpness.
(iteration 0-1000000)

(d) Training trajectory.
(iteration 0-1000000)

Figure 27: **Training Trajectory of Degree-4 Model with Label Noise.** ($\eta = 0.2$)
We plot the loss, sharpness, and training trajectory of the label noise model with $\sigma = 0.01$. The label noise model first follows a trajectory similar to the original GD training trajectory and reaches near the EoS minimum as shown in **(a, b)**. Then it follows the sharpness reduction flow along the manifold of minima and reaches the flattest minima as shown in **(c, d)**.

**Gradient Noise**:

For gradient noise model, we sample a perturbation of $(\delta, \delta')$ from a 2-dimensional spherical Gaussian $\mathcal{N}(0, \sigma^2 I_2)$ at each iteration and apply this perturbation to the gradient before we update the parameter. The one-step dynamics with gradient noise $(\delta, \delta')$ is then:

$$x_{t+1} = x_t - \eta(x_t y_t^2(x_t^2 y_t^2 - 1) + \delta), \quad y_{t+1} = y_t - \eta(x_t^2 y_t(x_t^2 y_t^2 - 1) + \delta').$$

With gradient noise, the initial parabolic trajectory can still be observed as shown in Fig. 28b. As the model reaches close to the manifold of global minima near the tip of the parabola, it no longer follows a monotone sharpness reduction flow (as in Fig. 27c for the label noise case) but instead randomly oscillates along the manifold of global minima between the two EoS minima. We believe this is due to the component of the gradient noise parallel to the minima manifold, which dominates the sharpness reduction effect.

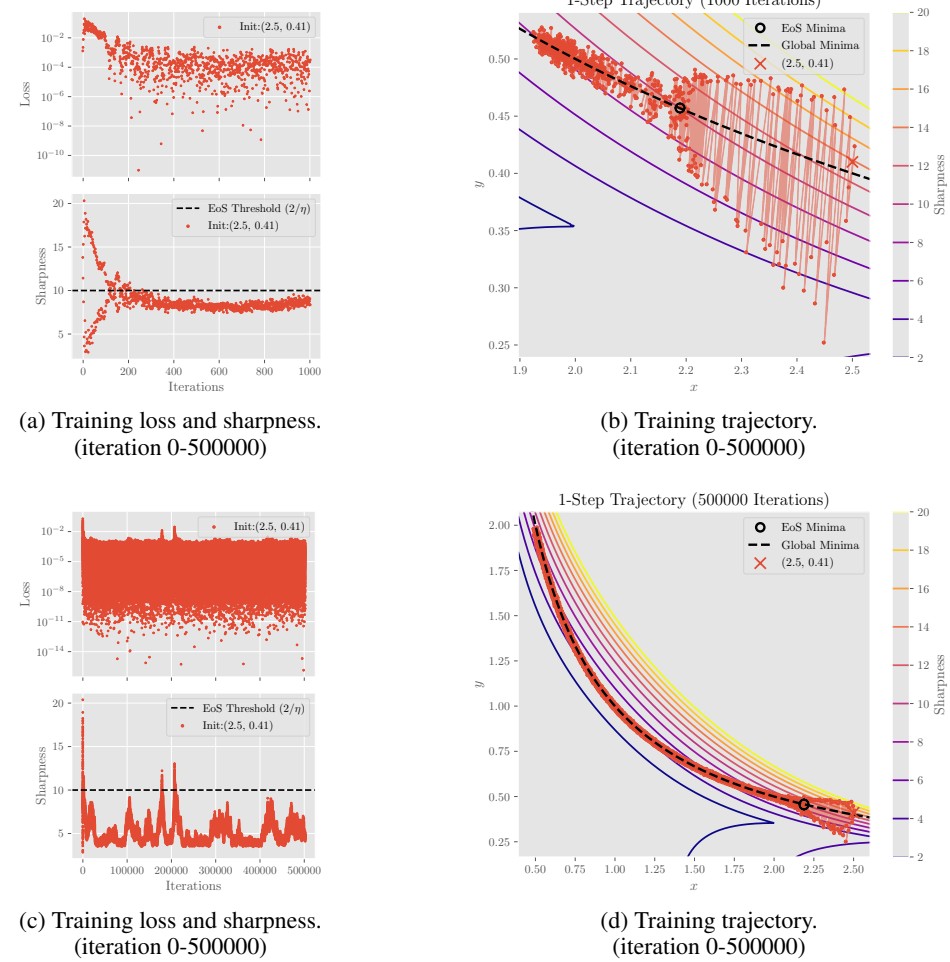

(a) Training loss and sharpness.
(iteration 0-500000)

(b) Training trajectory.
(iteration 0-500000)

(c) Training loss and sharpness.
(iteration 0-500000)

(d) Training trajectory.
(iteration 0-500000)

Figure 28: **Training Trajectory of Degree-4 Model with Gradient Noise.** ($\eta = 0.2$)
We plot the loss, sharpness, and training trajectory of the gradient noise model with $\sigma = 0.01$. Like the label noise model, the gradient noise model first follows a trajectory similar to the original GD training trajectory and reaches near the EoS minimum as shown in **(a, b)**. After getting around the EoS-minimum, the parameter begins to randomly oscillate and traverse around the manifold of global minima between the two EoS minima as shown in **(c,d)**. It is likely that the gradient noise finally converges to some distribution along the manifold of global minima.

### A.7.2 MINIBATCH SGD FOR OVER-PARAMETERIZED MODELS

In this section, we empirically investigate what happens to the converging sharpness when over-parameterized network are trained with minibatch SGD. We use the same 5-layer FC models and dataset as used in Section 6. Other than full-batch gradient descent, we also train the models with mini-batch gradient descent with varying batchsizes and record their converging sharpness.

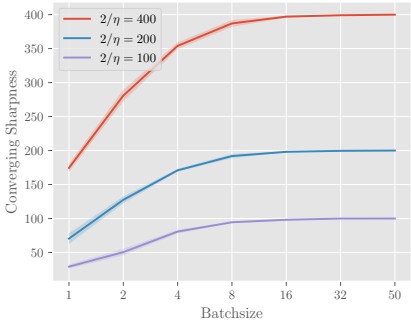
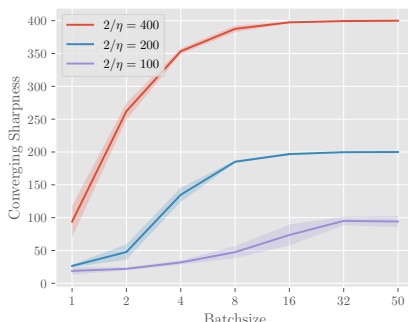

(a) 5-layer FC network with ELU activation   (b) 5-layer FC network with tanh activation

Figure 29: **FC networks trained with SGD of varying batchsizes.** ($\eta = 0.005, 0.01, 0.02$) For each learning rate and batchsize, we train 10 models from different random initialization for 20000 epochs and record their converging sharpness. The standard deviation of sharpness is represented by the shaded area. (The loss of all models converges to lower than $10^{-8}$). When the batch size of SGD is large, the converging sharpness is close to $2/\eta$, which is very similar to the gradient descent cases. On the other hand, the converging sharpness is significantly lower when the batch size is small compared with the dataset. It is worth noting that the converging sharpness for each batch size is quite concentrated.

Instead of going to the flattest minima (as in the label noise case) or randomly oscillating below the EoS threshold (as in the gradient noise case), the converging minima of overparameterized deep networks trained with mini-batch SGD have highly concentrated sharpness that is correlated to the batch size.

We note that a key difference between the over-parameterized mini-batch SGD and the noisy GD experiment discussed in Appendix A.7.1 is that the loss for mini-batch SGD can converge to a fixed point with loss 0 (i.e. the model can memorize all training data) while the models with fixed additive noise will not converge to a fixed point. Currently the minimalist scalar model example we analyzed can only memorize one data point. We believe it is an interesting future direction to generalize the model to memorize more data and understand why mini-batch SGD converges below the EoS threshold.

## B   THEORETICAL ANALYSIS ON THE DEGREE-4 EXAMPLE

In this section we present the complete rigorous analysis on the training dynamics of the degree-4 example discussed in Section 3. The section will be organized in the following way:

In Appendix B.1 we will first define the problem and two reparameterizations we used for analysis, this serves as a more comprehensive version of Section 2 in the main text.

In Appendix B.2 we will first restate our main theorem for the degree-4 example (Theorem 3.1) along with two corollaries (Corollary 3.1, and Corollary 3.2). These theoretical results characterized the sharpness concentration and sharpness adaptivity phenomenon of the degree-4 example. Then we will provide a more comprehensive proof sketch for Theorem 3.1 that is similar to the discussion in Section 3.2 of the main text.

Then we will provide the lemmas for dynamics approximation (Appendix B.3), phase I convergence (Appendix B.4), and phase II convergence (Appendix B.5).

Finally, in Appendix B.6 we will use the lemmas to complete the proof for the main theorem along with its corollaries.

### B.1   PRELIMINARIES

We consider a simple objective function $\mathcal{L}(x, y, z, w) = \frac{1}{2}(xyzw - 1)^2$. Denote $\gamma := xyzw$, then

$$\nabla \mathcal{L}(x, y, z, w) = (\gamma^2 - \gamma)[x, y, z, w]^{-1}, \tag{21}$$

$$\nabla^2 \mathcal{L}(x, y, z, w) = (\gamma^2 - \gamma) \begin{bmatrix} \gamma^2/x^2 & (2\gamma^2 - \gamma)/xy & (2\gamma^2 - \gamma)/xz & (2\gamma^2 - \gamma)/xw \\ (2\gamma^2 - \gamma)/xy & \gamma^2/y^2 & (2\gamma^2 - \gamma)/yz & (2\gamma^2 - \gamma)/yw \\ (2\gamma^2 - \gamma)/xz & (2\gamma^2 - \gamma)/yz & \gamma^2/z^2 & (2\gamma^2 - \gamma)/zw \\ (2\gamma^2 - \gamma)/xw & (2\gamma^2 - \gamma)/yw & (2\gamma^2 - \gamma)/zw & \gamma^2/w^2 \end{bmatrix}. \tag{22}$$

Let the parameter $[x, y, z, w]$ to be optimized by gradient descent with step size $\eta$, that

$$[x_{(t+1)}, y_{(t+1)}, z_{(t+1)}, w_{(t+1)}] = [x_{(t)}, y_{(t)}, z_{(t)}, w_{(t)}] - \eta \nabla \mathcal{L}(x_{(t)}, y_{(t)}, z_{(t)}, w_{(t)}). \tag{23}$$

To further simplify the problem, we consider the symmetric initialization of $z_0 = x_0$, $w_0 = y_0$. Note that due to symmetry of objective, the identical entries will remain identical throughout the training process, so the training dynamics reduces to two dimensional, and the global minima is simply $S = \{(x, y) \in \mathbb{R}^2 : x^2 y^2 = 1\}$. Computing the closed-form of the gradient, we know the 1-step update of $x$ and $y$ follows

$$x_{t+1} = x_t - x_t y_t^2 \eta(x_t^2 y_t^2 - 1), \quad y_{t+1} = y_t - x_t^2 y_t \eta(x_t^2 y_t^2 - 1). \tag{24}$$

Denote $\gamma = xy$, the parameter Hessian of the objective $\mathcal{L}$ at $(x, y, x, y)$ admits eigenvalues $\lambda_1 = x^2(1 - \gamma), \lambda_2 = y^2(1 - \gamma)$ and

$$\lambda_3 = \frac{1}{2} \left( (x^2 + y^2)(3\gamma^2 - 1) - \sqrt{(x^2 + y^2)^2(1 - 3\gamma^2)^2 + 4\gamma^2(3 - 10\gamma^2 + 7\gamma^4)} \right),$$
$$\lambda_4 = \frac{1}{2} \left( (x^2 + y^2)(3\gamma^2 - 1) + \sqrt{(x^2 + y^2)^2(1 - 3\gamma^2)^2 + 4\gamma^2(3 - 10\gamma^2 + 7\gamma^4)} \right). \tag{25}$$

When $(x, y)$ converges to any minimum, $\gamma = x^2 y^2 = 1$, so $\lambda_1, \lambda_2, \lambda_3$ all vanishes. Therefore it is $\lambda_4$ that corresponds to the EoS phenomenon people observe. When $\eta < \frac{1}{2}$, solving $\lambda_4 = 2/\eta$ with $x^2 y^2 = 1$ gives

$$x = \pm \frac{1}{\sqrt{2}} ((-4 + \eta^{-2})^{\frac{1}{2}} + \eta^{-1})^{\frac{1}{2}},$$
$$y = \pm \sqrt{2} ((-4 + \eta^{-2})^{\frac{1}{2}} + \eta^{-1})^{-\frac{1}{2}} \tag{26}$$

and their multiplicative inverses. These solutions correspond to the minima with sharpness exactly equal to the EoS threshold of $2/\eta$. Since they are all symmetric with each other, without loss of generality we pick the minimum of interest as

$$(\check{x}, \check{y}) \triangleq (\frac{1}{\sqrt{2}}((-4 + \eta^{-2})^{\frac{1}{2}} + \eta^{-1})^{\frac{1}{2}}, \sqrt{2}((-4 + \eta^{-2})^{\frac{1}{2}} + \eta^{-1})^{-\frac{1}{2}}). \tag{27}$$

To better analyze the dynamics under a more natural coordinate, we consider the reparameterization that For any $(x, y) \in \{(x, y) \in \mathbb{R}^+ \times \mathbb{R}^+ : x > y\}$, define

$$c \triangleq (x^2 - y^2)^{\frac{1}{2}}, \quad d \triangleq xy. \tag{28}$$

This gives a bijective continuous mapping between $\{(x, y) \in \mathbb{R}^+ \times \mathbb{R}^+ : x > y\}$ and $\{(c, d) \in \mathbb{R}^+ \times \mathbb{R}^+\}$. Intuitively, we are taking the lower half of $y = 1/x$ on the positive quadrant as $d = 1$. With $c, d$ as defined, the $\eta$-EoS minimum simplifies to $(\check{c}, \check{d}) \triangleq ((\eta^{-2} - 4)^{\frac{1}{4}}, 1)$. The inverse map can be computed as

$$x = \frac{\sqrt{c^2 + \sqrt{c^4 + 4d^2}}}{\sqrt{2}}, \quad y = \frac{\sqrt{2}d}{\sqrt{c^2 + \sqrt{c^4 + 4d^2}}}. \tag{29}$$

To expand the dynamics near the $\eta$-EoS minimum, we define

$$a \triangleq c - (\eta^{-2} - 4)^{\frac{1}{4}}, \quad b \triangleq d - 1 \tag{30}$$

to be the offset from $(\check{c}, \check{d})$. Our analysis will primarily be using the $(a, b)$-parameterization. To summarize, the $(c, d)$ and $(a, b)$ reparameterization of $(x, y)$ are respectively given by

$$(c, d) \triangleq \left( \left( x^2 - y^2 \right)^{\frac{1}{2}}, xy \right), \quad (a, b) \triangleq \left( \left( x^2 - y^2 \right)^{\frac{1}{2}} - \left( \eta^{-2} - 4 \right)^{\frac{1}{4}}, xy - 1 \right). \tag{31}$$

Let $\kappa \triangleq \sqrt{\eta}$, under the reparameterization Eq. (24) becomes.

$$
\begin{aligned}
a_{t+1} &= (\kappa^{-4} - 4)^{\frac{1}{4}} + \left( a_t + (\kappa^{-4} - 4)^{\frac{1}{4}} \right) \left( 1 - \left( (1 + b_t)^3 - (1 + b_t) \right)^2 \kappa^4 \right)^{\frac{1}{2}}, \\
b_{t+1} &= b_t + ((1 + b_t)^3 - 2(1 + b_t)^5 + (1 + b_t)^7)\kappa^4 \\
&\quad + \left( (1 + b_t) - (1 + b_t)^3 \right) \left( 4(1 + b_t)^2 \kappa^4 + (a_t \kappa + (1 - 4\kappa^4)^{\frac{1}{4}})^4 \right)^{\frac{1}{2}}.
\end{aligned}
\tag{32}
$$

## B.2 THEORETICAL RESULTS AND PROOF SKETCH

Now with the reparameterization defined, we restate our convergence result on the 4 scalar objective and discuss the proof sketch.

**Theorem 3.1** (Sharpness Concentration). *For a large enough absolute constant $K$, suppose $\kappa < \frac{1}{2000\sqrt{2}} K^{-1}$, and the initialization $(a_0, b_0)$ satisfies $a_0 \in (12\kappa^{\frac{5}{2}}, \frac{1}{4} K^{-2}\kappa^{-1})$ and $b_0 \in (-K^{-1}, K^{-1})\backslash\{0\}$. Consider the GD trajectory characterized in Eq. (6) with fixed step size $\kappa^2$ from $(a_0, b_0)$, for any $\epsilon > 0$ there exists $T = \mathcal{O}(K^{-2}\kappa^{-\frac{15}{2}} + \log(\epsilon^{-1}) + \log(|b_0|^{-1})\kappa^{-\frac{7}{2}})$ such that for all $t > T$, $|b_t| < \epsilon$ and $a_t \in (-\frac{5}{3}\kappa^3, -\frac{1}{10}\kappa^3)$.*

### B.2.1 PROOF SKETCH OF THEOREM 3.1

Our analysis begins with approximating the local movement using primarily Taylor expansion around the $\kappa^2$-EoS Minimum (Appendix B.3). We show that for initialization within a local region of width $2K^{-2}\kappa^{-1}$ and height $2K^{-1}$ centered at the $\kappa^2$-EoS minimum (condition B.1), the local two-step update of $a$ and $b$ can be characterized by

$$a'' = a - 4b^2\kappa^3 + R_a, \quad b'' = b - 16b^3 + 8ab\kappa + R_b. \tag{33}$$

Where $R_a, R_b$ are remainders that we can effectively bound (Corollary B.1). We note that in the region we are considering, $a$ is always monotonically decreasing at $b^2\kappa^3$ per 2 steps (Lemma 5).

With the approximation ready, we will conduct our convergence analysis with 2 phases.

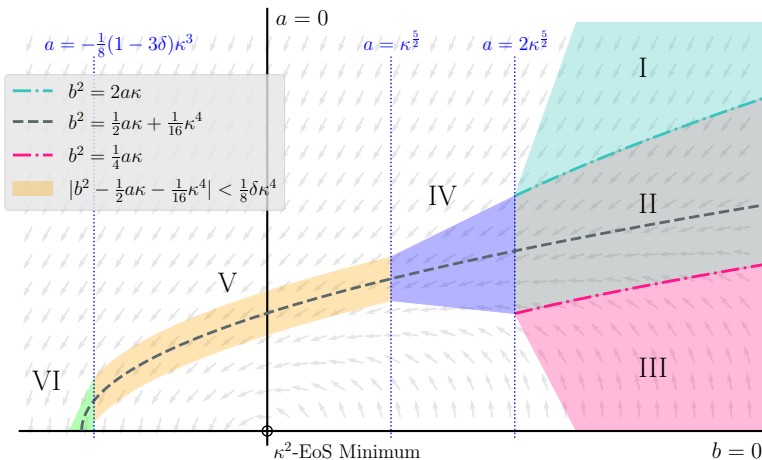

Figure 30: **Convergence Diagram** for GD on the 4 scalar example. The horizontal directions represents $a$ and the vertical direction represents $b$. The arrows indicate the directions of local 2-step movement. This diagram is for demonstration and ratios are not exact.

In **Phase 1** (Appendix B.4), we consider all possible initializations $(a_0, b_0)$ such that $a_0 \in (12\kappa^{\frac{5}{2}}, \frac{1}{4}K^{-2}\kappa^{-1})$ and $b_0 \in (-K^{-1}, K^{-1})\backslash\{0\}$. We partition the region of initializations into three parts separated by $b^2 = 2a\kappa$ and $b^2 = \frac{1}{4}a\kappa$ (shown as region I, II, III in Fig. 30).

- For initializations in region I where $b^2 > 2a\kappa$ (condition B.3), we show that the cubic term $b^3$ in the expression of $b''$ in Eq. (33) dominates the $ab\kappa$ term as well as the remainder, so that the two step update on $|b|$ is monotonically decreasing with at least an additive update of $-|b^3|$. Combining with slow movement of $a$, we show that initializations in region I will quickly enter region II (Lemma 8).

- For initializations in region III where $b^2 \in (0, \frac{1}{4}a\kappa)$ (condition B.4), we show that the $ab\kappa$ term will dominate the $b^3$ term and other remainders. Thus the two-step update of $|b|$ will monotonically increase with a multiplicative rate of at least $(1 + a\kappa)$. Combining with slow movement of $a$, we show that initializations in region III will also quickly enter region II (Lemma 9).

- For the last step in Phase 1, we show that the 2-step trajectories entering region II will stay in the region in the sense that $b^2 \in (\frac{1}{4}a\kappa, 2a\kappa)$. (Lemma 11) We also show that $a$ will keep decreasing and enter $(\frac{3}{2}\kappa^{\frac{5}{2}}, 2\kappa^{\frac{5}{2}})$. In the diagram (Fig. 30) this corresponds to the phase of entering region IV from II.

At the end of Phase I, we would have shown that all trajectories starting from the required initialization will converge to near the parabola $b^2 = \frac{1}{2}a\kappa$ and enter region IV from the right.

In **Phase 2** (Appendix B.5), we begin with initialization $(a_0, b_0)$ such that $a_0 \in (\frac{3}{2}\kappa^{\frac{5}{2}}, 2\kappa^{\frac{5}{2}})$ and $b_0^2 \in (\frac{1}{4}a_0\kappa, 2a_0\kappa)$. In phase 1 we have shown a rough convergence result close to the parabola $b^2 = \frac{1}{2}a\kappa$ with the extreme point of $(0, 0)$. In phase 2 we will change the parabola of interest to be $b^2 = \frac{1}{2}a\kappa + \frac{1}{16}\kappa^4$ which is characterized by the ODE approximation as discussed in Section 3.1. In particular we will focus on the residual $\xi \triangleq b^2 - \frac{1}{2}a\kappa - \frac{1}{16}\kappa^4$. The phase 2 convergence has the following three stages. Throughout the analysis we fix a small constant $\delta = 0.04$

- **Stage 1.** After the trajectory enters $a \in (\frac{3}{2}\kappa^{\frac{5}{2}}, 2\kappa^{\frac{5}{2}})$ and $b^2 \in (\frac{1}{4}a\kappa^{\frac{5}{2}}, 2a\kappa^{\frac{5}{2}})$, we will show that the two-step update of $\xi$ follows $|\xi'' - (1 - 32c^2)\xi| < \delta b^2 \kappa^4$. (Lemma 12). Then we show that $|\xi|$ will further decrease to less than $\frac{1}{8}\delta\kappa^4$ before $a$ decreases to less than $\kappa^{\frac{5}{2}}$, and the trajectory will enter region V from the right (Lemma 13).

- **Stage 2.** After the trajectory enters region V, we will show that it will remain close to the parabola $\xi \triangleq b^2 - \frac{1}{2}a\kappa - \frac{1}{16}\kappa^4$ that $|\xi|$ will remain less than $\frac{1}{8}\delta\kappa^4$ while $a$ decreases

into the interval $a \in (-\frac{1}{8}(1 - 3\delta\kappa^3), -\frac{1}{10}(1 + 2\delta)\kappa^3)$ and the trajectory enters region VI (Lemma 14).

- **Stage 3.** Finally we conclude the proof by a convergence analysis in region VI. The two-step dynamics approximation in region VI is very similar to region III, that the $ab\kappa$ term in the two-step update of $b$ will dominate. Since $a$ is now negative, $|b|$ will follow the multiplicative update $|b''| < (1 + a\kappa)|b|$. We will also show that the movement of $a$ will be small, and the final converging minima will not be far from the extreme point $a = \frac{1}{8}\kappa^3$ (Lemma 15).

### B.3    1 AND 2-STEP DYNAMICS OF $a$ AND $b$

Now we begin our rigorous analysis on the dynamics of $(a_t, b_t)$.

For simplicity of notations, when analyzing the 1-step and 2-step dynamics of $(a_t, b_t)$, we use $a, a', a''$ to denote $a_t, a_{t+1}, a_{t+2}$ and $b, b', b''$ to denote $b_t, b_{t+1}, b_{t+2}$. For simplicity of calculation, we consider the change of variable $\kappa = \sqrt{\eta}$.

In the following analysis, use operator $\mathcal{O}(\cdot)$ to only hides absolute constants that are independent of $\epsilon, \kappa, a, b$ and no asymptotic limits are taken. Concretely, for monomial $x$ and polynomial $y$ of some variables, we denote $y = \mathcal{O}(x)$ if there exists some absolute constant $K$ independent of the variables such that for any parameterization of the variables, $|y| < K|x|$. Note that this is stronger than the usual big-$O$ notations. Throughout the analysis we will use $K$ to represent the absolute constant that uniformly upper bounds all the absolute constants of the $\mathcal{O}(\cdot)$ terms. This is well defined as we will only be considering finite number of $\mathcal{O}(\cdot)$ terms.

#### B.3.1    ONE STEP DYNAMICS APPROXIMATION OF $a$ AND $b$

**Lemma 1.** *Fix any positive $\epsilon$ that $\epsilon < 0.5$, for any $\kappa \in (0, \epsilon^{\frac{1}{4}})$, for all $(a, b)$ such that $|a| < \epsilon\kappa^{-1}$ and $|b| < \min\{1, \frac{1}{5}\epsilon\kappa^{-2}\}$, we have*

$$a' = a - 2b^2\kappa^3 + \mathcal{O}(\epsilon b^2\kappa^3) + \mathcal{O}(b^2\kappa^4). \tag{34}$$

*Proof of Lemma 1.* Recall from Eq. (32) that

$$a' = \left(a + \left(\kappa^{-4} - 4\right)^{\frac{1}{4}}\right)\left(1 - b^2(1 + b)^2(2 + b)^2\kappa^4\right)^{\frac{1}{2}} - \left(\kappa^{-4} - 4\right)^{\frac{1}{4}}. \tag{35}$$

Since $\left(\kappa^{-4} - 4\right)^{\frac{1}{4}}$ will approach infinity as $\kappa$ goes to 0, we instead analyze

$$\kappa a' = \left(\kappa a + \left(1 - 4\kappa^4\right)^{\frac{1}{4}}\right)\left(1 - b^2(1 + b)^2(2 + b)^2\kappa^4\right)^{\frac{1}{2}} - \left(1 - 4\kappa^4\right)^{\frac{1}{4}}. \tag{36}$$

Note that $\left(1 - b^2(1 + b)^2(2 + b)^2\kappa^4\right)^{\frac{1}{2}}$ will be close to $1 - 2b^2(1 + b)^2(2 + b)^2\kappa^4$ for not so large $b$ and $1 - 4\kappa^4$ will be close to 1 for small $\kappa$, we will leverage these two properties to approximate $a'$.

First observe that for any $x \in (0, 8\epsilon/(1 + 2\epsilon)^2)$ we have $1 - \frac{x}{2} - \epsilon x < \sqrt{1 - x} < 1 - \frac{x}{2}$. Since $\epsilon < 0.5$, $(1 + 2\epsilon)^2 < 4$, so it is sufficient to let $x < 2\epsilon$ for the inequality to hold.

Now we substitute $x$ by $b^2(1 + b)^2(2 + b)^2\kappa^4$. Since $|b| < 1$, $(1 + b)^2(2 + b)^2 < 36$. Thus when $|b| < \frac{1}{5}\epsilon\kappa^{-2}$ we have $b^2(1 + b)^2(2 + b)^2\kappa^4 < 36b^2\kappa^4 < 36(\frac{1}{5}\epsilon\kappa^{-2})^2\kappa^4 < 2\epsilon$, and therefore

$$\begin{aligned}
\left(1 - b^2(1 + b)^2(2 + b)^2\kappa^4\right)^{\frac{1}{2}} &= 1 - \frac{1}{2}\left(b^2(1 + b)^2(2 + b)^2\kappa^4\right) + \mathcal{O}(\epsilon b^2\kappa^4) \\
&= 1 - 2b^2\kappa^4 + \mathcal{O}(b^2\kappa^5) + \mathcal{O}(\epsilon b^2\kappa^4).
\end{aligned} \tag{37}$$

Since $1 - 4\kappa^4 > 0$, by Bernoulli inequality we have $(1 - \kappa^4)^4 \geq 1 - 4\kappa^4$, combining with the requirement of $\kappa < \epsilon^{\frac{1}{4}}$, we have $(1 - 4\kappa^4)^{\frac{1}{4}} \geq 1 - \kappa^4 > 1 - \epsilon$. Meanwhile since we required $|d| < \epsilon\kappa^{-1}$, $|\kappa d| < \epsilon$, so

$$\left(\kappa d + (1 - 4\kappa^4)^{\frac{1}{4}}\right) = 1 + \mathcal{O}(\epsilon). \tag{38}$$

Combining Eq. (37) and Eq. (38) we have

$$
\begin{aligned}
\kappa a' &= \left(\kappa a + \left(1 - 4\kappa^4\right)^{\frac{1}{4}}\right)\left(1 - b^2(1+b)^2(2+b)^2\kappa^4\right)^{\frac{1}{2}} - \left(1 - 4\kappa^4\right)^{\frac{1}{4}} \\
&= \left(\kappa a + \left(1 - 4\kappa^4\right)^{\frac{1}{4}}\right)\left(1 - 2b^2\kappa^4 + \mathcal{O}(b^2\kappa^5) + \mathcal{O}(\epsilon b^2\kappa^4)\right) - \left(1 - 4\kappa^4\right)^{\frac{1}{4}} \\
&= \kappa a\left(1 - 2b^2\kappa^4 + \mathcal{O}(b^2\kappa^5) + \mathcal{O}(\epsilon b^2\kappa^4)\right) + \left(1 - 4\kappa^4\right)^{\frac{1}{4}}\left(-2b^2\kappa^4 + \mathcal{O}(b^2\kappa^5) + \mathcal{O}(\epsilon b^2\kappa^4)\right) \\
&= \kappa a - \left(\kappa a + \left(1 - 4\kappa^4\right)^{\frac{1}{4}}\right)\left(-2b^2\kappa^4 + \mathcal{O}(b^2\kappa^5) + \mathcal{O}(\epsilon b^2\kappa^4)\right) \\
&= \kappa a - (1 + \mathcal{O}(\epsilon))\left(-2b^2\kappa^4 + \mathcal{O}(b^2\kappa^5) + \mathcal{O}(\epsilon b^2\kappa^4)\right) \\
&= \kappa a - 2b^2\kappa^4 + \mathcal{O}(b^2\kappa^5) + \mathcal{O}(\epsilon b^2\kappa^4).
\end{aligned}
\tag{39}
$$

Dividing both sides by $\kappa$ completes the proof. $\qquad\square$

**Lemma 2.** *For any $\kappa \in (0, 0.1)$, for all $(a, b)$ such that $|a| < \kappa^{-1}$ and $|b| < 1$,*

$$
b' = -b - 4ab\kappa - 3b^2 - b^3 + \mathcal{O}(ab^2\kappa) + \mathcal{O}(a^2b\kappa^2) + \mathcal{O}(b^2\kappa^4) + \mathcal{O}(b\kappa^5). \tag{40}
$$

*Proof of Lemma 2.* For simplicity of notation, let

$$
\delta \triangleq \sqrt{4\eta^2(1+b)^2 + \left(a\eta^{\frac{1}{2}} + (1 - 4\eta^2)^{\frac{1}{4}}\right)^4} = \sqrt{4\kappa^4(1+b)^2 + \left(a\kappa + (1 - 4\kappa^4)^{\frac{1}{4}}\right)^4}. \tag{41}
$$

Plugging Eq. (41) into Eq. (32), since $|b| < 1$ we have

$$
4b^2\kappa^4 + 16b^3\kappa^4 + 25b^4\kappa^4 + 19b^5\kappa^4 + 7b^6\kappa^4 + b^7\kappa^4 = \mathcal{O}(b^2\kappa^4). \tag{42}
$$

Thus

$$
\begin{aligned}
b' &= b - 2b\delta - 3b^2\delta - b^3\delta + 4b^2\kappa^4 + 16b^3\kappa^4 + 25b^4\kappa^4 + 19b^5\kappa^4 + 7b^6\kappa^4 + b^7\kappa^4 \\
&= b - 2b\delta - 3b^2\delta - b^3\delta + \mathcal{O}(b^2\kappa^4)
\end{aligned}
\tag{43}
$$

Since $\kappa < 0.1$, $|a| < \kappa^{-1}$ and $|b| < 1$, by Lemma 18 we have

$$
\delta = 1 + 2a\kappa + a^2\kappa^2 + (4b + 2b^2)\kappa^4 + \mathcal{O}(\kappa^5). \tag{44}
$$

Thus

$$
\begin{aligned}
b' &= b - 2b - 4ab\kappa - 3b^2 - 6ab^2\kappa - b^3 - 2ab^3\kappa \\
&\quad + (2b + 3b^2 + b^3)\left(\mathcal{O}(a^2\kappa^2) + \mathcal{O}(b\kappa^4) + \mathcal{O}(\kappa^5)\right) \\
&= -b - 4ab\kappa - 3b^2 - b^3 + \mathcal{O}(ab^2\kappa) + \mathcal{O}(a^2b\kappa^2) + \mathcal{O}(b^2\kappa^4) + \mathcal{O}(b\kappa^5).
\end{aligned}
\tag{45}
$$

$\qquad\square$

Combining the conditions required by Lemma 1 and Lemma 2 we have

**Condition B.1** (One-step Dynamics Approximation Condition)**.**

$$
\begin{aligned}
&\epsilon < 0.5, \\
&\kappa < \min\{0.1, \epsilon^{\frac{1}{4}}\}, \\
&|a| < \epsilon\kappa^{-1}, \\
&|b| < \min\{1, \tfrac{1}{5}\epsilon\kappa^{-2}\}.
\end{aligned}
\tag{46}
$$

When condition B.1 is satisfied, we have

$$
\begin{aligned}
a' &= a - 2b^2\kappa^3 + \mathcal{O}(\epsilon b^2\kappa^3) + \mathcal{O}(b^2\kappa^4), \\
b' &= -b - 4ab\kappa - 3b^2 - b^3 + \mathcal{O}(ab^2\kappa) + \mathcal{O}(ab^2\kappa^2) + \mathcal{O}(b^2\kappa^4) + \mathcal{O}(b\kappa^5).
\end{aligned}
\tag{47}
$$

### B.3.2 Two Step Dynamics Approximation of $a$ and $b$

Now we approximate the 2-step dynamics for $(a, b)$.

**Lemma 3.** *Fix some positive constant $\epsilon < 0.1$, with $\kappa, a, b$ satisfying condition B.1,*

$$b'' = b - 16b^3 + 8ab\kappa + \mathcal{O}(b^4) + \mathcal{O}(ab^2\kappa) + \mathcal{O}(a^2b\kappa^2) + \mathcal{O}(b^2\kappa^4) + \mathcal{O}(b\kappa^5) + \mathcal{O}(\epsilon b^2\kappa^3). \quad (48)$$

*Proof.* Combining the one step dynamics characterized in Lemma 1 and Lemma 2 we have

$$b'' = -b' - 4a'b'\kappa - 3b'^2 - b'^3 + \mathcal{O}(a'^2b'\kappa^2) + \mathcal{O}(a'b'^2\kappa) + \mathcal{O}(b'^2\kappa^4) + \mathcal{O}(b'\kappa^5). \quad (49)$$

We will analyze these terms one by one.

$$
\begin{aligned}
a'b'\kappa &= \kappa \left( a - 2b^2\kappa^3 + \mathcal{O}(\epsilon b^2\kappa^3) + \mathcal{O}(b^2\kappa^4) \right) \left( -b - 4ab\kappa - 3b^2 - b^3 + \mathcal{O}(ab^2\kappa) \right. \\
&\qquad \left. + \mathcal{O}(a^2b\kappa^2) + \mathcal{O}(b^2\kappa^4) + \mathcal{O}(b\kappa^5) \right) \\
&= a\kappa \left( -b - 4ab\kappa - 3b^2 - b^3 + \mathcal{O}(ab^2\kappa) + \mathcal{O}(a^2b\kappa^2) + \mathcal{O}(b^2\kappa^4) + \mathcal{O}(b\kappa^5) \right) \\
&\quad - 2b^2\kappa^4 \left( -b - 4ab\kappa - 3b^2 - b^3 + \mathcal{O}(ab^2\kappa) + \mathcal{O}(a^2b\kappa^2) + \mathcal{O}(b^2\kappa^4) + \mathcal{O}(b\kappa^5) \right) \\
&\quad + \mathcal{O}(\epsilon b^2\kappa^3) + \mathcal{O}(b^2\kappa^4) \\
&= -ab\kappa + \mathcal{O}(ab^2\kappa) + \mathcal{O}(a^2b\kappa^2) + \mathcal{O}(b^2\kappa^4) + \mathcal{O}(b\kappa^5) + \mathcal{O}(\epsilon b^2\kappa^3),
\end{aligned}
\quad (50)
$$

$$
\begin{aligned}
b'^2 &= \left( -b - 4ab\kappa - 3b^2 - b^3 + \mathcal{O}(ab^2\kappa) + \mathcal{O}(a^2b\kappa^2) + \mathcal{O}(b^2\kappa^4) + \mathcal{O}(b\kappa^5) \right)^2 \\
&= \left( b \left( -1 - 4a\kappa - 3b - b^2 \right) + \mathcal{O}(ab^2\kappa) + \mathcal{O}(a^2b\kappa^2) + \mathcal{O}(b^2\kappa^4) + \mathcal{O}(b\kappa^5) \right)^2 \\
&= \left( -b - 4ab\kappa - 3b^2 - b^3 \right)^2 + \mathcal{O}(b^3a\kappa) + \mathcal{O}(a^2b^2\kappa^2) + \mathcal{O}(b^3\kappa^4) + \mathcal{O}(b^2\kappa^5) \\
&= b^2 + 6b^3 + \mathcal{O}(b^4) + \mathcal{O}(a^2b^2\kappa^2) + \mathcal{O}(b^3\kappa^4) + \mathcal{O}(b^2\kappa^5),
\end{aligned}
\quad (51)
$$

$$
\begin{aligned}
b'^3 &= \left( -b - 4ab\kappa - 3b^2 - b^3 + \mathcal{O}(ab^2\kappa) + \mathcal{O}(a^2b\kappa^2) + \mathcal{O}(b^2\kappa^4) + \mathcal{O}(b\kappa^5) \right)^3 \\
&= \left( b \left( -1 - 4a\kappa - 3b - b^2 \right) + \mathcal{O}(ab^2\kappa) + \mathcal{O}(a^2b\kappa^2) + \mathcal{O}(b^2\kappa^4) + \mathcal{O}(b\kappa^5) \right)^3 \\
&= \left( -b - 4ab\kappa - 3b^2 - b^3 \right)^3 + \mathcal{O}(ab^2\kappa) + \mathcal{O}(a^2b\kappa^2) + \mathcal{O}(b^2\kappa^4) + \mathcal{O}(b\kappa^5) \\
&= -b^3 + \mathcal{O}(b^4) + \mathcal{O}(a^2b\kappa^2) + \mathcal{O}(b^2\kappa^4) + \mathcal{O}(b\kappa^5).
\end{aligned}
\quad (52)
$$

Note that $b' = \mathcal{O}(b) + \mathcal{O}(ab\kappa)$ and $a' = a + \mathcal{O}(b^2\kappa^3)$, so

$$
\begin{aligned}
\mathcal{O}(a'b'^2\kappa) &= \mathcal{O}(ab^2\kappa) + \mathcal{O}(b^3\kappa^4), \\
\mathcal{O}(a'^2b'\kappa^2) &= \mathcal{O}(a^2b\kappa^2) + \mathcal{O}(a^2b\kappa^2), \\
\mathcal{O}(b'^2\kappa^4) &= \mathcal{O}(b^2\kappa^4) + \mathcal{O}(ab^2\kappa^5), \\
\mathcal{O}(b'\kappa^5) &= \mathcal{O}(b\kappa^5) + \mathcal{O}(ab\kappa^6).
\end{aligned}
\quad (53)
$$

Hence

$$\mathcal{O}(b'^2a'\kappa) + \mathcal{O}(b'a'^2\kappa^2) + \mathcal{O}(b'^2\kappa^4) + \mathcal{O}(b'\kappa^5) = \mathcal{O}(ab^2\kappa) + \mathcal{O}(a^2b\kappa^2) + \mathcal{O}(b^2\kappa^4) + \mathcal{O}(b\kappa^5). \quad (54)$$

Combining above we have

$$
\begin{aligned}
b'' &= -\left( -b - 4ab\kappa - 3b^2 - b^3 \right) - 4(-ab\kappa) - 3(b^2 + 6b^3) - (-b^3) \\
&\quad + \mathcal{O}(b^4) + \mathcal{O}(ab^2\kappa) + \mathcal{O}(a^2b\kappa^2) + \mathcal{O}(b^2\kappa^4) + \mathcal{O}(b\kappa^5) + \mathcal{O}(\epsilon b^2\kappa^3) \\
&= b + 8ab\kappa - 16b^3 + \mathcal{O}(b^4) + \mathcal{O}(ab^2\kappa) + \mathcal{O}(a^2b\kappa^2) + \mathcal{O}(b^2\kappa^4) + \mathcal{O}(b\kappa^5) + \mathcal{O}(\epsilon b^2\kappa^3).
\end{aligned}
\quad (55)
$$

$\square$

**Lemma 4.** *Fix some positive constant $\epsilon < 0.5$, with $\kappa, a, b$ satisfying condition B.1,*

$$a'' = a - 4b^2\kappa^4 + \mathcal{O}(\epsilon b^2\kappa^3) + \mathcal{O}(b^2\kappa^4). \quad (56)$$

*Proof of Lemma 4.* Combining Lemma 1 and Lemma 2 we have

$$a'' = a' - 2b'^2\kappa^3 + \mathcal{O}(\epsilon b'^2\kappa^3) + \mathcal{O}(b'^2\kappa^4). \tag{57}$$

From Eq. (51) in the proof of Lemma 3 we have

$$b'^2 = b^2 + \mathcal{O}(b^3) + \mathcal{O}(a^2b^2\kappa^2) + \mathcal{O}(b^3\kappa^4) + \mathcal{O}(b^2\kappa^5). \tag{58}$$

Hence the last three terms of Eq. (57) can be calculated that

$$b'^2\kappa^3 = b^2\kappa^3 + \mathcal{O}(b^4\kappa^3) + \mathcal{O}(a^2b^2\kappa^5) + \mathcal{O}(b^3\kappa^7) + \mathcal{O}(b^2\kappa^8), \tag{59}$$

$$\begin{aligned}
\mathcal{O}(\epsilon b'^2\kappa^3) &= \mathcal{O}(b'^2)\epsilon\kappa^3 \\
&= \mathcal{O}(\epsilon b^3\kappa^3) + \mathcal{O}(\epsilon b^4\kappa^3) + \mathcal{O}(\epsilon a^2b^2\kappa^5) + \mathcal{O}(\epsilon b^3\kappa^7) + \mathcal{O}(\epsilon b^2\kappa^8) \\
&= \mathcal{O}(\epsilon b^2\kappa^3) + \mathcal{O}(\epsilon b\kappa^8),
\end{aligned} \tag{60}$$

$$\begin{aligned}
\mathcal{O}(b'^2\kappa^4) &= \mathcal{O}(b'^2)\kappa^4 \\
&= \mathcal{O}(b^2\kappa^4) + \mathcal{O}(b^3\kappa^4) + \mathcal{O}(a^2b^2\kappa^6) + \mathcal{O}(b^3\kappa^8) + \mathcal{O}(b^2\kappa^9) \\
&= \mathcal{O}(b^2\kappa^4) + \mathcal{O}(a^2b^2\kappa^6) + \mathcal{O}(b^2\kappa^9).
\end{aligned} \tag{61}$$

Combining above we have that

$$\begin{aligned}
a'' &= a' - 2b'^2\kappa^3 + \mathcal{O}(\epsilon b'^2\kappa^3) + \mathcal{O}(b'^2\kappa^4) \\
&= a - 2b^2\kappa^3 + \mathcal{O}(\epsilon b^2\kappa^3) + \mathcal{O}(b^2\kappa^4) - 2b^2\kappa^3 \\
&\quad + \mathcal{O}(b^3\kappa^3) + \mathcal{O}(a^2b^2\kappa^5) + \mathcal{O}(b^3\kappa^7) + \mathcal{O}(b^2\kappa^8) + \mathcal{O}(\epsilon b^2\kappa^3) + \mathcal{O}(\epsilon a^2b^2\kappa^5) \\
&\quad + \mathcal{O}(\epsilon b^2\kappa^8) + \mathcal{O}(b^2\kappa^4) + \mathcal{O}(a^2b^2\kappa^6) + \mathcal{O}(b^2\kappa^9) \\
&= a - 4b^2\kappa^3 + \mathcal{O}(\epsilon b^2\kappa^3) + \mathcal{O}(b^3\kappa^3) + \mathcal{O}(b^2\kappa^4).
\end{aligned} \tag{62}$$

This completes the proof. $\square$

Combining Lemma 4 and Lemma 3, the 2-step dynamics approximation can be summarized as:

**Corollary B.1.** *For any positive constant $\epsilon < 0.5$, with $\kappa, a, b$ satisfying condition B.1 we have*

$$\begin{aligned}
b'' &= b - 16b^3 + 8ab\kappa + \mathcal{O}(b^4) + \mathcal{O}(ab^2\kappa) + \mathcal{O}(a^2b\kappa^2) + \mathcal{O}(b^2\kappa^4) + \mathcal{O}(b\kappa^5) + \mathcal{O}(\epsilon b^2\kappa^3), \\
a'' &= a - 4b^2\kappa^3 + \mathcal{O}(\epsilon b^2\kappa^3) + \mathcal{O}(b^3\kappa^3) + \mathcal{O}(b^2\kappa^4).
\end{aligned} \tag{63}$$

### B.4 Phase I: Convergence to Near Parabola

Now we show that under the $(a, b)$ parameterization, any initialization $(a_0, b_0)$ such that $a_0 \in (3\kappa^{\frac{5}{2}}, \frac{1}{4}K^{-2}\kappa^{-1})$ and $b_0 \in (-K^{-1}, K^{-1})$ will converge near the parabola very fast. As mentioned above, there are mainly three regimes of interest. We will first determine the region in which the two step update $b'' - b$ is solely dominated by $-b^3$ or $ab\kappa$.

To formally analyze the different dynamics and characterizing the regimes for them, in this section we use $K$ to denote the uniform upper bound over the absolute constants hidden by the $\mathcal{O}(\cdot)$ operator in the 2-step dynamics approximation characterized by Corollary B.1. Since there is only finite terms with $\mathcal{O}$-notation, such constant $K$ is well defined and independent of $\epsilon, \kappa, a, b$. Without loss of generality we assume $K > 512$.

Rewriting Corollary B.1 with the uniform upper bound $K$, we have the following corollary:

**Corollary B.2.** *(2-step dynamics approximation of $a$ and $b$) There exists some absolute constants $K$ such that for any constant $\epsilon < 0.5$, for all $\kappa, a, b$ satisfying condition B.1, the 2-step update of $(a, b)$ can be characterized by*

$$a'' = a - 4b^2\kappa^3 + R_a, \quad b'' = b - 16b^3 + 8ab\kappa + R_b. \tag{64}$$

*where the remainder $R_a$ and $R_b$ have upper bound:*

$$\begin{aligned}
|R_a| &< K\left|\epsilon c^2\kappa^3\right| + K\left|c^3\kappa^3\right| + K\left|c^2\kappa^4\right|, \\
|R_b| &< K\left|c^4\right| + K\left|c^2d\kappa\right| + K\left|cd^2\kappa^2\right| + K\left|c^2\kappa^4\right| + K\left|c\kappa^5\right| + K\left|\epsilon c^2\kappa^3\right|.
\end{aligned} \tag{65}$$

Now we establish the conditions to characterize the work zones in order for the analysis to be more tractable in different regimes.

### B.4.1 WHEN $a'' - a$ IS CLOSE TO $-4b^2\kappa^3$

Here we formalize the observation that when $b$ and $\kappa$ are not large, the two-step movement of $a$ is monotone and always close to $4b^2\kappa^3$. Concretely we have the following lemma:

**Lemma 5.** *Fix $\epsilon = K^{-1}$, for all $\kappa, a, b$ satisfying condition B.1 as well as the extra condition of $|\kappa| < K^{-1}$ and $|b| < K^{-1}$, we have $|a'' - (a - 4b^2\kappa^3)| < 3b^2\kappa^3$.*

*Proof of Lemma 5.* Since we assume $K > 512$, we have $\epsilon = K^{-1} < 0.5$ as required by Corollary B.2. Since $\kappa, a, b$ satisfies condition B.1, by Corollary B.2 we have $a'' = a - 4b^2\kappa^3 + R_a$ where

$$|R_a| < K \left| \epsilon b^2\kappa^3 \right| + K \left| b^3\kappa^3 \right| + K \left| b^2\kappa^4 \right|. \tag{66}$$

Thus to prove the lemma we only need to bound every term on RHS of Eq. (66) by $b^2\kappa^3$.

(i) Since we fixed $\epsilon = K^{-1}$, $K \left| \epsilon b^2\kappa^3 \right| = b^2\kappa^3$.

(ii) Since $|b| < K^{-1}$, $K \left| b^3\kappa^3 \right| < K \left| K^{-1}b^2\kappa^3 \right| < b^2\kappa^3$.

(iii) Since $\kappa < K^{-1}$, $K \left| b^2\kappa^4 \right| < K \left| K^{-1}b^2\kappa^3 \right| < b^2\kappa^3$

Therefore $|R_a| < 3b^2\kappa^3$ which completes the proof. $\qquad\square$

Note that when we fix $\epsilon = K^{-1}$, condition B.1 becomes

$$\begin{aligned} &\kappa < \min\{0.1, K^{-\frac{1}{4}}\}, \\ &|a| < K^{-1}\kappa^{-1}, \\ &|b| < \min\{1, \tfrac{1}{5}K^{-1}\kappa^{-2}\}. \end{aligned} \tag{67}$$

Combining with the additional condition of Lemma 5, we can summarize the condition for Lemma 5 to hold as

$$\begin{aligned} &\kappa < \min\{0.1, K^{-\frac{1}{4}}, K^{-1}\}, &&(1) \\ &|a| < K^{-1}\kappa^{-1}, &&(2) \\ &|b| < \min\{1, K^{-1}, \tfrac{1}{5}K^{-1}\kappa^{-2}\}. &&(3) \end{aligned} \tag{68}$$

With $K > 512$, $K^{-1} < 0.1$, so (1) can be reduced to $\kappa < K^{-1}$. Since $\kappa < K^{-1}$, $\tfrac{1}{5}\kappa^{-2} > \tfrac{1}{5}K^2$, so $\tfrac{1}{5}K^{-1}\kappa^{-2} > \tfrac{1}{5}K > 1 > K^{-1}$, and thus (3) can be reduced to $|b| < K^{-1}$. In conclusion, the following condition is sufficient for $|a'' - (a - 4b^2\kappa^3)| < 3b^2\kappa^3$.

**Condition B.2** (Condition for $(4 \pm 3)b^2\kappa^3$ movement of $a$)**.**

$$\begin{aligned} &\kappa < K^{-1}, \\ &|a| < K^{-1}\kappa^{-1}, \\ &|b| < K^{-1}. \end{aligned} \tag{69}$$

### B.4.2 WHEN $-b^3$ DOMINATES THE DYNAMICS OF $b$

**Lemma 6.** *For all $\kappa, a, b$ satisfying*

$$\begin{aligned} &\kappa < K^{-1}, \\ &|a| \in (\kappa^3, K^{-2}\kappa^{-1}), \\ &|b| \in (\sqrt{|a|\kappa}, K^{-1}). \end{aligned} \tag{70}$$

*We have $|b'' - (b - 16b^3)| \le 14|b^3|$.*

*Proof of Lemma 6.* First it is straightforward to check that the condition on $\kappa, a, b$ is stronger than condition B.1 if we set $\epsilon = 0.1$, thus by Corollary B.2 we have $b'' = b - 16b^3 + 8ab\kappa + R_b$ where

$$|R_b| < K \left| b^4 \right| + K \left| ab^2\kappa \right| + K \left| a^2b\kappa^2 \right| + K \left| b^2\kappa^4 \right| + K \left| b\kappa^5 \right| + K \left| \epsilon b^2\kappa^3 \right|. \tag{71}$$

Thus to prove the claim, it is sufficient to bound $|8ab\kappa|$ by $8|b^3|$ and every term on RHS of Eq. (71) by $|b^3|$. We will now bound them term by term.

(i) Since $\sqrt{|a|\kappa} < |b|$, $|a\kappa| < b^2$, so $8|ab\kappa| = 8|b||a\kappa| < 8|b^3|$.

(ii) Since $|b| < K^{-1}$, $K|b^4| < K|K^{-1}b^3| = |b^3|$.

(iii) Since $|a| < K^{-2}\kappa^{-1}$, multiplying $K^2|a\kappa^2|$ on both side gives $K^2a^2\kappa^2 < |a|\kappa$. it follows by taking square root that $K|a|\kappa < \sqrt{|a|\kappa}$. Since $|b| > \sqrt{|a|\kappa}$, we have $K|a|\kappa < |b|$. Multiply $b^2$ on both side gives $K|ab^2\kappa| < |b^3|$.

(iv) Since $|a| < K^{-2}\kappa^{-1} < K^{-1}\kappa^{-1}$, multiply $K|a|\kappa$ on both side gives $Ka^2\kappa^2 < |a|\kappa$ and hence $\sqrt{K}|a|\kappa < \sqrt{|a|\kappa} < |b|$. Squaring both sides and multiply by $b$ gives $K|a^2b\kappa^2| < |b^3|$.

(v) Since $|a| > \kappa^3$ and $|b| > \sqrt{|a|\kappa}$, we have $|b| > \kappa^2 > K\kappa^4$. Multiplying $b^2$ on both sides, we have $K|b^2\kappa^4| < |b^3|$.

(vi) In (v) we observe that $|b| > \kappa^2$, so $|b| > \sqrt{K}\kappa^{\frac{5}{2}}$ and hence $b^2 > K\kappa^5$. Multiplying $|b|$ on both side gives $K|b\kappa^3| < |b^3|$.

(vii) Since we fixed $\epsilon = 0.1$ while $\kappa < K^{-1} < 0.1$, we have $\epsilon\kappa < 1$. Since $|b| > \kappa^2$, we have $|b| > \kappa^3\epsilon > K\epsilon\kappa^3$. Multiply $b^2$ on both side gives $K|\epsilon c^2\kappa^3| < |c^3|$.

Therefore we have $|8ab\kappa| + |R_b| < 14|c^3|$, which completes the proof. $\qquad\square$

We restate the sufficient condition for $|b'' - (b - 16b^3)| \leq 14|b^3|$ as follow

**Condition B.3** (Condition for $-b^3$ Dominated $b$ Movement)**.**

$$\begin{aligned} \kappa &< K^{-1}, \\ |a| &\in (\kappa^3, K^{-2}\kappa^{-1}), \\ |b| &\in (\sqrt{|a|\kappa}, K^{-1}). \end{aligned} \tag{72}$$

### B.4.3 WHEN $ab\kappa$ DOMINATES THE $b'' - b$

**Lemma 7.** *For any $\kappa, a, b$ satisfying*

$$\begin{aligned} \kappa &< K^{-1}, \\ |a| &\in (\kappa^3, K^{-2}\kappa^{-1}), \\ |b| &< \min\{\tfrac{1}{2\sqrt{2}}\sqrt{|a|\kappa}, K^{-1}\}. \end{aligned} \tag{73}$$

*We have $|b'' - (b + 8ab\kappa)| \leq 7|ab\kappa|$.*

*Proof of Lemma 7.* First fix $\epsilon = 0.5$. It is straightforward to check that for all $\kappa, a, b$ satisfying the given condition, condition B.1 is also satisfied, so by Corollary B.2 we have $b'' = b - 16b^3 + 8ab\kappa + R_b$ where

$$|R_b| < K\left|b^4\right| + K\left|ab^2\kappa\right| + K\left|a^2b\kappa^2\right| + K\left|b^2\kappa^4\right| + K\left|b\kappa^5\right| + K\left|\epsilon b^2\kappa^3\right|. \tag{74}$$

Thus to prove the claim, it is sufficient to bound $16|b^3|$ by $2|ab\kappa|$, two terms in RHS of Eq. (74) by $\frac{1}{2}|b^3|$ and the remaining 4 terms by $|b^3|$. We will now bound them term by term.

(i) Since $|b| < \frac{1}{2\sqrt{2}}\sqrt{|a|\kappa}$, $b^2 < \frac{1}{8}|a|\kappa$. Multiply $16|b|$ on both side gives $16|b^3| < 2|ab\kappa|$.

(ii) Since $|a| < K^{-2}\kappa$, we have $|a|\kappa < K^{-2}$. Multiplying $a^2\kappa^2$ on both side gives $|a^3|\kappa^3 < K^{-2}a^2\kappa^3$. If we take the 6-th root, we have $\sqrt{|a|\kappa} < (K^{-1}|a|\kappa)^{\frac{1}{3}}$. Since $|b| < \sqrt{|a|\kappa}$, we have $|b| < (K^{-1}|a|\kappa)^{\frac{1}{3}}$ and hence $|b^3| < K^{-1}|a|\kappa$. Multiply $K|b|$ on both side gives $K|b^4| < |ab\kappa|$.

(iii) Since $|b| < K^{-1}$, multiply both side by $K|ab\kappa|$ gives $K|ab^2\kappa| < |ab\kappa|$.

(iv) Since $|a| < K^{-2}\kappa^{-1} < \frac{1}{2}K^{-1}\kappa^{-1}$ (as we assumed $K > 512$), multiply both side by $|ab|\kappa^2$ gives $K|a^2b\kappa^2| < \frac{1}{2}|ab\kappa|$.

(v) Since $|a| > \kappa^3$ and $\kappa < K^{-1}$, we have $|a| > \kappa^3 > K^{-2}\kappa^5 > K^2\kappa^7$. Multiplying $K^{-2}|a|\kappa^{-6}$ on both side gives $K^{-2}a^2\kappa^{-6} > |a|\kappa$. It follows by taking the square root that $\sqrt{|a|\kappa} < K^{-1}|a|\kappa^{-3}$. Since we have $|b| < \frac{1}{2\sqrt{2}}\sqrt{|a|\kappa} < \frac{1}{2}\sqrt{|a|\kappa}$, combining with above gives $|b| < \frac{1}{2}K^{-1}|a|\kappa^{-3}$. Multiply $K|b|\kappa^4$ on both side gives $K|b^2\kappa^4| < \frac{1}{2}|ab\kappa|$.

(vi) Since $|a| > \kappa^3$, we have $|a| > K\kappa^4$. Multiplying $|b|\kappa$ on both side gives $K|b\kappa^5| < |ab\kappa|$.

(vii) Since $|a| > \kappa^3$ and $\epsilon = 0.5$, we have $|a| > K^2\epsilon^2\kappa^5$. Taking the multiplicative inverse and multiply $a^2\kappa$ on both side gives $|a|\kappa < K^{-2}\epsilon^{-2}\kappa^{-6}a^2$. It follows by taking the square root that $\sqrt{|a|\kappa} < K^{-1}\epsilon^{-1}\kappa^{-3}|a|$. Since $|b| < \sqrt{|a|\kappa}$, we have $|b| < K^{-1}\epsilon^{-1}\kappa^{-3}|a|$. Finally multiply $K\epsilon|b|\kappa^3$ on both side, we have $K|\epsilon b^2\kappa^3| < |ab\kappa|$.

From (i) we have $16|b^3| < 2|ab\kappa|$, from (ii) - (vii) we have $R_b = K\left|b^4\right| + K\left|ab^2\kappa\right| + K\left|a^2b\kappa^2\right| + K\left|b^2\kappa^4\right| + K\left|b\kappa^5\right| + K\left|\epsilon b^2\kappa^3\right| < 5|ab\kappa|$. Therefore $|b'' - (b + 8ab\kappa)| \le 16|b^3| + R_b \le 7|ab\kappa|$, which completes the proof. $\qquad\square$

We restate the sufficient condition for $|b'' - (b + 8ab\kappa)| \le 7|ab\kappa|$ as follow:

**Condition B.4** (Condition for $ab\kappa$ Dominated $b'' - b$ Movement)**.**

$$
\begin{aligned}
&\kappa < K^{-1}, \\
&|a| \in (\kappa^3, K^{-2}\kappa^{-1}), \\
&|b| < \min\{\tfrac{1}{2\sqrt{2}}\sqrt{|a|\kappa}, K^{-1}\}.
\end{aligned}
\tag{75}
$$

### B.4.4 Convergence when $-b^3$ dominates $ab\kappa$

Now we are ready to analyze the dynamics when $c^3$ dominates the movement of $c$. For $t \in \mathbb{N}$, we abuse the notation to let $(c_t, d_t)$ denote $c$ and $d$ after the $t$-th **2-step update** with step size $\eta = \kappa^2$ from the initialization $(c_0, d_0)$.

**Lemma 8** (Convergence to near parabola from large $b$)**.** *For any $\kappa < K^{-1}$, for any initialization $(a_0, b_0)$ satisfying $a_0 \in (12\kappa^{\frac{5}{2}}, \frac{1}{4}K^{-2}\kappa^{-1})$ and $|b_0| \in [2\sqrt{|a_0|\kappa}, K^{-1})$, there exists some $T < \kappa^{-4}$ such that $a_T \in (2\kappa^{\frac{5}{2}}, K^{-2}\kappa^{-1})$ and $|b_T| \in (\sqrt{a_T\kappa}, 2\sqrt{a_T\kappa})$.*

*Proof of Lemma 8.* We will prove the claim using induction.

Consider the inductive hypothesis for $k \in \{0, 1, \cdots, \lfloor \kappa^{-4}\rfloor\}$ that

$$
P(k)\colon |a_k| \in (2\kappa^{\frac{5}{2}}, \tfrac{1}{4}K^{-2}\kappa^{-1}) \text{ and } |b_k| \in (2\sqrt{a_k\kappa}, \min\{K^{-1}, (k+1)^{-\frac{1}{2}}\}).
$$

Since $|b_0| < K^{-1} < 1 = (0+1)^{-\frac{1}{2}}$ and $a_0 \in (3\kappa^{\frac{5}{2}}, \frac{1}{4}K^{-2}\kappa^{-1}) \subset (2\kappa^{\frac{5}{2}}, \frac{1}{4}K^{-2}\kappa^{-1})$ as required by the initialization, the base case $P(0)$ holds trivially. Now we can proceed to the inductive step.

Assume $P(l)$ holds for all $l \le k$, we want to show that $P(k+1)$ holds unless $|b_{k+1}| < 2\sqrt{|a_{k+1}|\kappa}$.

By the inductive hypothesis, $(a_k, b_k)$ satisfies condition B.2 and condition B.3. Thus by Lemma 5 and Lemma 6 we have

$$
a_{k+1} \in (a_k - 7b_k^2\kappa^3, a_k - b_k^2\kappa^3), \qquad |b_{k+1}| \in (|b_k| - 30|b_k^3|, |b_k| - 2|b_k^3|).
\tag{76}
$$

By the strong inductive hypothesis, these properties also holds when substituting $k$ by any $l < k$.

We first check the lower bound for $a_{k+1}$ under the assumption that $k < \kappa^{-4}$. Since $a_{l+1} > a_l - 7b_l^2\kappa^3$ for all $l \leq k$, we have

$$
\begin{aligned}
a_{k+1} &> a_0 - \sum_{l=0}^{k} 7b_l^2\kappa^3 \\
&> a_0 - \sum_{l=0}^{k} 7\left((l+1)^{-\frac{1}{2}}\right)^2\kappa^3 \qquad \text{(Since } b_l < (l+1)^{-\frac{1}{2}} \text{ by IH)} \\
&> 12\kappa^{\frac{5}{2}} - 7\kappa^3\sum_{l=0}^{k}\frac{1}{l+1} \\
&> 12\kappa^{\frac{5}{2}} - 7\kappa^3\left(1 + \int_1^{k+1}\frac{1}{\tau}d\tau\right) \\
&= 12\kappa^{\frac{5}{2}} - 7\kappa^3(1 + \log(k+1)).
\end{aligned}
\tag{77}
$$

Under the assumption that $k < \kappa^{-4}$ and $\kappa < K^{-1} < \frac{1}{512}$, it is not hard to check that

$$
1 + \log(k+1) < 2 + \log(\kappa^{-4}) = 2 - 4\log(\kappa) < \frac{10}{7}\kappa^{-\frac{1}{2}}.
\tag{78}
$$

The last inequality holds since when $\kappa = \frac{1}{512}$ we have $2 - 4\log(\frac{1}{512}) - \frac{10}{7}(\frac{1}{512})^{-\frac{1}{2}} < -5$ and $2 + \log(\kappa^{-4}) = 2 - 4\log(\kappa)$ is monotonically increasing when $\kappa < \frac{1}{512}$. Plugging back into Eq. (77), we know that if $P(l)$ holds for all $l \leq k$ and $k < \kappa^{-4}$, then

$$
a_k + 1 > 12\kappa^{\frac{5}{2}} - 7\kappa^3\left(\frac{10}{7}\kappa^{-\frac{1}{2}}\right) = 12\kappa^{\frac{5}{2}} - 10\kappa^{\frac{5}{2}} = 2\kappa^{\frac{5}{2}}.
\tag{79}
$$

The upper bound of $a_k < \frac{1}{4}K^{-2}\kappa^{-1}$ always holds since $a$ is monotonically decreasing by Eq. (76).

Now we check the upper bound for $|b_{k+1}|$. Consider $f(x) = x^{-\frac{1}{2}}$, we have $f'(x) = -\frac{1}{2}x^{-\frac{3}{2}} = -\frac{1}{2}f(x)^3$ and $f''(x) = \frac{3}{4}x^{-\frac{5}{2}}$. Note that $f''(x) > 0$ for all $x > 0$, so by a first order Taylor expansion around $x = k + 1$ we have $f(k+2) > f(k+1) - \frac{1}{2}f(k+1)^3$. Combined with $|b_{k+1}| < |b_k| - |b_k^3|$, it follows

$$
\begin{aligned}
f(k+2) - |b_{k+1}| &\geq f(k+1) - \frac{1}{2}f(k+1)^3 - |b_k| + |b_k^3| \\
&= (f(k+1) - |b_k|) - \frac{1}{2}(f(k+1)^3 - |b_k^3|) + \frac{1}{2}b_k^3 \\
&\geq (f(k+1) - |b_k|) - \frac{1}{2}(f(k+1) - |b_k|)(b_k^2 + |b_k|f(k+1) + f(k+1)^2) \\
&= (f(k+1) - |b_k|)\left(1 - \frac{1}{2}\left(b_k^2 + |b_k|f(k+1) + f(k+1)^2\right)\right).
\end{aligned}
\tag{80}
$$

Since $f(k+1) \leq 1$ and $|b_k| < |b_0| < K^{-1} < \frac{1}{2}$. for all $k \geq 1$, we have $b_k^2 + |b_k|f(k+1) + f(k+1)^2 < 2$, and hence $\left(1 - \frac{1}{2}\left(b_k^2 + |b_k|f(k+1) + f(k+1)^2\right)\right) > 0$. Since $|b_k| < (k+1)^{-\frac{1}{2}} = f(k+1)$ by the induction hypothesis, we have $f(k+2) - |b_{t+1}| > 0$, so $|b_{k+1}| < (k+2)^{-\frac{1}{2}}$. The other upper bound $|b_{k+1}| < K^{-1}$ holds trivially since $|b_0| < K^{-1}$ as required by the initialization and $|b_k|$ is monotonically decreasing according to Eq. (76).

Now we will show that there exists some $\tau < \kappa^{-4}$ that $|b_\tau| < 2\sqrt{|a_\tau|\kappa}$. Assume toward contradiction that there is no such $\tau$, then the induction may proceed to $h \triangleq \lfloor\kappa^{-4}\rfloor$ so that $a_h > 2\kappa^{\frac{5}{2}}$, $|b_h| < (h+1)^{\frac{1}{2}} < (\kappa^{-4})^{\frac{1}{2}} < \kappa^2$ and $|b_h| > 2\sqrt{a_h\kappa} > 2\sqrt{2\kappa^{\frac{5}{2}}\kappa} = 2\sqrt{2}\kappa^{\frac{7}{4}}$. The last two inequalities lead to contradiction as $2\sqrt{2}\kappa^{\frac{7}{4}} > \kappa^2$, so the assumption does not hold and there exists some $\tau < \kappa^{-4}$ for $|b_\tau| < 2\sqrt{a_\tau\kappa}$.

Let $T < \lfloor\kappa^{-4}\rfloor$ be the smallest such $\tau$, then the induction may proceed to $k = T - 1$, which guarantees $P(t)$ for all $t < T$. Thus for all $t < T$, $a_t > (2\kappa^{\frac{5}{2}}, K^{-2}\kappa^{-1})$ and $|b_t| \in (2\sqrt{a_T\kappa}, K^{-1})$. Since $P(T-1)$ holds, following Eq. (77) we also have $a_T > 2\kappa^{\frac{5}{2}}$.

Now we still need to show $|b_T| > \sqrt{a_T \kappa}$. Since $a_k$ is monotonically decreasing, it is sufficient to show $|b_T| > \sqrt{a_{T-1} \kappa}$. From Eq. (76) we have $|b_T| > |b_{T-1}| - 30|b_{T-1}^3| = |b_{T-1}|(1 - 30b_{T-1}^2)$. Since $b_{T-1} < K^{-1} < \frac{1}{512}$, $(1 - 30b_{T-1}^2) > \frac{1}{2}$. Combined with $|b_{T-1}| > 2\sqrt{a_{T-1}\kappa}$ as shown above, we have $|b_T| > \frac{1}{2}(2\sqrt{a_{T-1}\kappa}) > \sqrt{a_{T-1}\kappa}$, which completes the proof.

$\square$

### B.4.5 Convergence when $ab\kappa$ dominates $b^3$

**Lemma 9** (Convergence to near parabola from small $b$). *For any $\kappa < K^{-1}$, for any initialization $(a_0, b_0)$ satisfying $a_0 \in (12\kappa^{\frac{5}{2}}, \frac{1}{4}K^{-2}\kappa^{-1})$ and $|b_0| \in (0, \frac{1}{4}\sqrt{a_0 \kappa}]$, there exists some $T < \frac{1}{2}\log(|b_0|^{-1})\kappa^{-\frac{7}{2}}$ such that $|b_T| \in (\frac{1}{4}\sqrt{a_T\kappa}, \frac{1}{2}\sqrt{a_T\kappa})$ and $a_T \in (2\kappa^{\frac{5}{2}}, \frac{1}{4}K^{-2}\kappa^{-1})$.*

*Proof of Lemma Lemma 9.* We will prove this claim using induction.

Consider the inductive hypothesis for $k \in \mathbb{N}$ that

$$P(k) : |b_k| \in (|b_0|(1 + 4\kappa^{\frac{7}{2}})^k, \frac{1}{4}\sqrt{a_k \kappa}), a_0 \in (2\kappa^{\frac{5}{2}}, \frac{1}{4}K^{-2}\kappa^{-1})$$

$$\text{and if } k \geq 1, (|b_k| - |b_0|)/(a_0 - a_k) > \kappa^{-\frac{5}{4}}.$$

The base case when $k = 0$ holds from the initialization, so we proceed to the inductive step. Assume $P(l)$ holds for all $l \leq k$, we want to show $P(k + 1)$ holds unless $|b_k| > \frac{1}{4}\sqrt{a_k\kappa}$.

First note that by the inductive hypothesis, since $a_k < \frac{1}{4}K^{-2}\kappa^{-1}$, we have $|b_k| < \frac{1}{4}\sqrt{a_k\kappa} < \frac{1}{4}\sqrt{K^{-2}\kappa^{-1}\kappa} < K^{-1}$. Combining with conditions on $|a_k|$ and $|b_k|$ from the inductive hypothesis we have $(a_k, b_k)$ satisfying condition B.2 and condition B.4. Thus by Lemma 5 and Lemma 7 we have

$$a_{k+1} \in (a_k - 7b_k^2\kappa^3, a_k - b_k^2\kappa^3), \qquad |b_{k+1}| \in (|b_k| + 2|a_k b_k \kappa|, |b_k| + 14|a_k b_k \kappa|). \tag{81}$$

Observe that when $a$ is not too small, the movement of $b$ is significantly larger than the movement of $a$. From Eq. (81) we have $a_k - a_{k+1} < 8b_k^2\kappa^3$ and $|b_{k+1}| - |b_k| > 2a_k|b_k|\kappa$, so

$$\begin{aligned}
\frac{|b_{k+1}| - |b_k|}{a_k - a_{k+1}} &> \frac{2a_k|b_k|\kappa}{8b_k^2\kappa^3} = \frac{1}{|b_k|}\frac{a_k}{4\kappa^2} \\
&> \frac{4}{\sqrt{a_k\kappa}}\frac{a_k}{4\kappa^2} &&\text{(since } |b_t| < \tfrac{1}{4}\sqrt{a_t\kappa}\text{)} \\
&= \frac{\sqrt{a_k}}{\kappa^{\frac{5}{2}}} > \kappa^{-\frac{5}{4}}. &&\text{(since } a_k > \kappa^{\frac{5}{2}}\text{)}
\end{aligned} \tag{82}$$

When $k = 0$, we directly have $(|b_1| - |b_0|)/(a_0 - a_1) > \kappa^{-\frac{5}{4}}$. When $k \geq 1$, from the inductive hypothesis we have $(|b_k| - |b_0|)/(a_0 - a_k) > \kappa^{-\frac{5}{4}}$, combining with $(|b_{k+1}| - |b_k|)/(a_k - a_{k+1}) > \kappa^{-\frac{5}{4}}$, we have $(|b_{k+1}| - |b_0|)/(a_0 - a_{k+1}) > \kappa^{-\frac{5}{4}}$ by the mediant inequality.

Now we check the bounds on $|a_{k+1}|$ and $|b_{k+1}|$.

Since $a$ is monotonically decreasing from Eq. (81), we have $a_{k+1} < a_0 < \frac{1}{4}K^{-2}\kappa^{-1}$.

Since $(|b_k| - |b_0|)/(a_0 - a_k) > \kappa^{-\frac{5}{4}}$ according to the inductive hypothesis and $(a_0 - a_k) > 0$ by monotonicity of $a$, $a_0 - a_k < (|b_k| - |b_0|)\kappa^{\frac{5}{4}} < |b_k|\kappa^{\frac{5}{4}} < \frac{1}{4}\sqrt{a_k\kappa}\kappa^{\frac{5}{4}} = \frac{1}{4}\sqrt{a_k}\kappa^{\frac{7}{4}} < \frac{1}{4}\sqrt{a_0}\kappa^{\frac{7}{4}}$. Here the last step holds again by monotonicity of $a$. Reorganizing the inequality we have $a_k > a_0 - \frac{1}{4}\sqrt{a_0}\kappa^{\frac{7}{4}} = \sqrt{a_0}(\sqrt{a_0} - \frac{1}{4}\kappa^{\frac{7}{4}})$. Since $a_0 > 12\kappa^{\frac{5}{2}}$ by the initialization condition, $\sqrt{a_0} > 3\kappa^{\frac{5}{4}}$ and $\sqrt{a_0} - \frac{1}{4}\kappa^{\frac{7}{4}} > 2\kappa^{\frac{5}{4}}$. Thus $a_k > \sqrt{a_0}(\sqrt{a_0} - \frac{1}{4}\kappa^{\frac{7}{4}}) > 6\kappa^{\frac{5}{2}}$. Note that since $|b_k| < K^{-1} < \frac{1}{512}$, $7b_k^2\kappa^3 < 7K^{-2}\kappa^3 < \kappa^3$. From Eq. (81) we have $a_{k+1} > a_k - 8b_k^2\kappa^3 > 6\kappa^{\frac{5}{2}} - \kappa^3 > 2\kappa^{\frac{5}{2}}$. This gives the desired lower bound for $a_{k+1}$

Since $|b_{k+1}| > |b_k| + 2a_k|b_k|\kappa$ from Eq. (81), combining with $a_k > 2\kappa^{\frac{5}{2}}$ we have $|b_{k+1}| > |b_k|(1 + 2a_k\kappa) > |b_k|(1 + 4\kappa^{\frac{5}{2}}\kappa)$. Since $|b_k| > |b_0|(1 + 4\kappa^{\frac{7}{2}})^k$ by the inductive hypothesis, we have $|b_{k+1}| > |b_0|(1 + 4\kappa^{\frac{7}{2}})^{k+1}$.

With the guarantees on $|a_{k+1}|$, $|b_{k+1}|$, and $(|b_k| - |b_0|)/(a_0 - a_k)$ as shown above, if we additionally assume that $|b_{k+1}| < \frac{1}{4}\sqrt{a_{k+1}\kappa}$, $P(k+1)$ will hold and the induction can proceed.

Now claim that there exists some $\tau < \frac{1}{2}\log(|b_0|^{-1})\kappa^{-\frac{7}{2}}$ that $|b_\tau| \geq \frac{1}{4}\sqrt{a_\tau\kappa}$. Assume toward contradiction that there is no such $\tau$, then the induction can proceed for any $t \in \mathbb{N}$.

Consider $t \geq \frac{1}{2}\log(|b_0|^{-1})\kappa^{-\frac{7}{2}}$, we have

$$
\begin{aligned}
t &\geq \log(|b_0|^{-1})(\tfrac{1}{4}\kappa^{-\frac{7}{2}} + \tfrac{1}{4}\kappa^{-\frac{7}{2}}) \\
&> \log(|b_0|^{-1})(1 + \tfrac{1}{4}\kappa^{-\frac{7}{2}}) && (\text{Since } \kappa^{-\frac{7}{2}} > \tfrac{1}{2}) \\
&> \left(\tfrac{1}{2}\log(K^{-2}) + \log(|b_0|^{-1})\right)(1 + \tfrac{1}{4}\kappa^{-\frac{7}{2}}) && (\text{Since } \log(K^{-2}) < 0) \\
&= \left(\tfrac{1}{2}\log((K^{-2}\kappa^{-1})\kappa) + \log(|b_0|^{-1})\right)(1 + \tfrac{1}{4}\kappa^{-\frac{7}{2}}) && (83) \\
&> \left(\tfrac{1}{2}\log(a_k\kappa) + \log(|b_0|^{-1})\right)(1 + \tfrac{1}{4}\kappa^{-\frac{7}{2}}) && (\text{Since } K^{-2}\kappa^{-1} > a_k) \\
&= \left(\tfrac{1}{2}\log(a_k\kappa) + \log(|b_0|^{-1})\right)(1 + 4\kappa^{\frac{7}{2}})/4\kappa^{\frac{7}{2}} \\
&= \left(\tfrac{1}{2}\log(a_k\kappa) + \log(|b_0|^{-1})\right)\log(1 + 4\kappa^{\frac{7}{2}})^{-1} && (\text{Since } \log(1+x) \geq x/(1+x)).
\end{aligned}
$$

It follows that

$$
\begin{aligned}
t &> \left(\tfrac{1}{2}\log(a_k\kappa) + \log(|b_0|^{-1})\right)\log(1 + 4\kappa^{\frac{7}{2}})^{-1} \\
&= \left(\log(\sqrt{a_k\kappa}) + \log(|b_0|^{-1})\right)\log(1 + 4\kappa^{\frac{7}{2}})^{-1} \\
&> \left(\log(\tfrac{1}{4}\sqrt{a_k\kappa}) + \log(|b_0|^{-1})\right)\log(1 + 4\kappa^{\frac{7}{2}})^{-1} \\
&= \log_{(1+4\kappa^{\frac{7}{2}})}(\tfrac{1}{4}\sqrt{a_t\kappa}/|b_0|).
\end{aligned}
\tag{84}
$$

Hence $(1 + 4\kappa^{\frac{7}{2}})^t > \frac{1}{4}\sqrt{a_t\kappa}/|b_0|$, and therefore $|b_t| > |b_0|(1 + 4\kappa^{\frac{7}{2}})^t > \frac{1}{4}\sqrt{a_t\kappa}$, which contradicts that $P(t)$ holds and lead to contradiction.

Therefore there must exists some $\tau$ such that $|b_\tau| > \frac{1}{4}\sqrt{a_\tau\kappa}$. Let $T < \frac{1}{2}\log(|b_0|^{-1})\kappa^{-\frac{7}{2}}$ be the smallest such $\tau$, then the induction will proceed to $k = T - 1$. Moreover since $P(T-1)$ holds, following the previous analysis, the bounds on $|a_T|$ also holds, so we have $|a_T| \in (2\kappa^{\frac{5}{2}}, \frac{1}{4}K^{-2}\kappa^{-1})$.

Now to complete the proof we only need to show that $|b_T| < \frac{1}{2}\sqrt{a_T\kappa}$.

From Eq. (81) we have $|b_T| < |b_{T-1}| + 14a_{T-1}|b_{T-1}|\kappa = (1 + 14a_{T-1}\kappa)|b_{T-1}|$. Since $a_{T-1} < \frac{1}{4}K^{-2}\kappa^{-1}$ and $K > 512$, $(1 + 14a_{T-1}\kappa) < (1 + 14K^{-2}) < 1.1$. So

$$
|b_T| < 1.1|b_{T-1}| < 1.1(\tfrac{1}{4}\sqrt{a_{T-1}\kappa}).
\tag{85}
$$

On the other side, note that $a_T > a_{T-1} - 7b_{T-1}^2\kappa^3$ by Eq. (81), where $b_{T-1}^2 < (\frac{1}{4}\sqrt{a_{T-1}\kappa})^2 = \frac{1}{16}a_{T-1}\kappa$. We have $a_T > a_{T-1} - \frac{7}{16}a_{T-1}\kappa > (1 - \kappa)a_{T-1}$. Since $\kappa < K^{-1} < \frac{1}{512}$, we have $a_T > 0.99a_{T-1}$ and hence

$$
\tfrac{1}{4}\sqrt{a_{T-1}\kappa} < 1.1(\tfrac{1}{4}\sqrt{a_T\kappa}).
\tag{86}
$$

Combining Eq. (85) and Eq. (86) we have $|b_T| < 1.21(\frac{1}{4}\sqrt{a_T\kappa}) < \frac{1}{2}\sqrt{a_T\kappa}$. This concludes the proof for this lemma.

$\square$

### B.4.6 $|b|$ STAYS IN $(\frac{1}{4}\sqrt{a\kappa}, 2\sqrt{a\kappa})$ AS $a$ DECREASES WHEN $a$ IS NOT TOO SMALL

After showing that $b$ will enter $(\frac{1}{4}\sqrt{a\kappa}, 2\sqrt{a\kappa})$, we now show that it will not leave this region unless $a$ is very small. To do so, we first determine a regime in which we can effectively bound the two-step movement of $b$.

**Lemma 10.** *For any $\kappa, a, b$ satisfying*

$$
\begin{aligned}
\kappa &< K^{-1}, \\
|a| &< \frac{1}{4}K^{-2}\kappa^{-1}, \\
|b| &< 2\sqrt{a\kappa}.
\end{aligned}
\tag{87}
$$

*we have* $|b'' - b| < \frac{1}{16}\sqrt{a\kappa}$.

*Proof of Lemma 10.* Fix $\epsilon = 0.1$, it is straightforward to check that $\kappa, a, b$ satisfies condition B.1. Hence from Eq. (64) and Eq. (65) we have that

$$|b'' - b| < 16|b^3| + 8|ab\kappa| + Kb^4 + K|ab^2\kappa| + K|a^2b\kappa^2| + Kb^2\kappa^4 + K|b\kappa^5| + K|\epsilon|b^2\kappa^3. \quad (88)$$

To prove the claim it is sufficient to bound all terms on RHS under $\frac{1}{128}\sqrt{a\kappa}$. Note that with $b < 2\sqrt{a\kappa}$ and $a < K^{-2}\kappa^{-1}$, $\sqrt{a\kappa} < \sqrt{K^{-2}} = K^{-1} < \frac{1}{128}$ and hence $b < 2K^{-1}$.

(i) Since $K > 512$, $a\kappa < K^{-2} < \frac{1}{16384}$. Multiplying $128\sqrt{a\kappa}$ on both sides we have $128(a\kappa)^{\frac{3}{2}} < \frac{1}{128}\sqrt{a\kappa}$. Since $|b| < 2\sqrt{a\kappa}$, $|b^3| < 8(a\kappa)^{\frac{3}{2}}$, thus $16|b^3| < 128(a\kappa)^{\frac{3}{2}} < \frac{1}{128}\sqrt{a\kappa}$.

(ii) Since $|b| < 2\sqrt{a\kappa}$, $8|ab\kappa| < 16(a\kappa)^{\frac{3}{2}} < \frac{1}{1024}\sqrt{a\kappa}$. The last inequality holds directly from (i).

(iii) Since $|b| < 2\sqrt{a\kappa}$, $Kb^4 < 16Ka^2\kappa^2 = 16K(a\kappa)^{\frac{3}{2}}\sqrt{a\kappa}$. Since $\sqrt{a\kappa} < K^{-1}$, $16K(a\kappa)^{\frac{3}{2}}\sqrt{a\kappa} < 16K^{-2}\sqrt{a\kappa} < \frac{16}{16284}\sqrt{a\kappa} < \frac{1}{128}\sqrt{a\kappa}$. Therefore $Kb^4 < \frac{1}{128}\sqrt{ab\kappa}$.

(iv) Since $|b| < 2K^{-1}$, $K|ab^2\kappa| < 2|ab\kappa|$, which is less than $\frac{1}{512}\sqrt{a\kappa}$ from (ii).

(v) Since $|a\kappa| < K^{-1}$, $K|ab^2\kappa^2| < |ab\kappa| < \frac{1}{1024}\sqrt{ab\kappa}$.

(vi) Note that as $\kappa < 0.1$ in condition B.1, $\kappa^4 < \frac{1}{512}$. Since $|b| < 2\sqrt{a\kappa} < 2K^{-1}$, $b^2 < 4K^{-1}\sqrt{a\kappa}$, and thus $Kb^2\kappa^4 < 4\sqrt{a\kappa}\kappa^4 < \frac{1}{128}\sqrt{a\kappa}$.

(vii) Since $b < 2\sqrt{a\kappa}$, $K|b\kappa^5| < 2K\kappa^5\sqrt{a\kappa}$. Given $\kappa < K^{-1} < \frac{1}{4}K^{-\frac{1}{5}}$, we have $\kappa^5 < \frac{1}{1024}K^{-1}$, and hence $2K\kappa^5\sqrt{a\kappa} < \frac{1}{512}\sqrt{a\kappa}$. Thus $K|b\kappa^5| < \frac{1}{512}\sqrt{ab\kappa}$.

(viii) Since we fixed $\epsilon = 0.1$, $\kappa < K^{-1} < \frac{1}{512} < \frac{1}{8}\epsilon^{-\frac{1}{3}}$, $\kappa^3 < \frac{1}{512}\epsilon^{-1}$. Multiplying $4\epsilon\sqrt{a\kappa}$ on both sides we have $4\epsilon\sqrt{a\kappa}\kappa^3 < \frac{1}{128}\sqrt{a\kappa}$. Since $b^2 < 4K^{-1}\sqrt{a\kappa}$ from (vi), we have $K\epsilon b^2\kappa^3 < K(4K^{-1}\sqrt{a\kappa})\kappa^3 = 4\epsilon\sqrt{a\kappa}\kappa^3 < \frac{1}{128}\sqrt{a\kappa}$.

Now that we have bounded every monomial term on RHS of Eq. (88) by $\frac{1}{128}\sqrt{a\kappa}$, we have $|b'' - b| < \frac{1}{16}\sqrt{a\kappa}$, which completes the proof. $\qquad \square$

Here we restate the condition for Lemma 10

**Condition B.5** (Condition for small $b$ movement)**.**

$$\begin{aligned} \kappa &< K^{-1}, \\ |a| &< K^{-2}\kappa^{-1}, \\ |b| &< 2\sqrt{d\kappa}. \end{aligned} \quad (89)$$

With the two-step movement of $c$ bounded above, we may proceed to state the lemma that guarantees $c$ will not leave $(\frac{1}{4}\sqrt{d\kappa}, 2\sqrt{d\kappa})$ unless $d$ is very small.

**Lemma 11.** *For any $\kappa$ and initialization $a_0, b_0$ satisfying*

$$\begin{aligned} \kappa &< \frac{1}{16}K^{-1}, \\ a_0 &\in (\kappa^{\frac{5}{2}}, \frac{1}{4}K^{-2}\kappa^{-1}), \\ |b_0| &\in (\frac{1}{4}\sqrt{a_0\kappa}, 2\sqrt{a_0\kappa}). \end{aligned} \quad (90)$$

*There exists some $T \leq 16a_0\kappa^{-\frac{13}{2}}$ such that $a_T < \kappa^{\frac{5}{2}}$ and for all $t < T$, $a_t > \kappa^{\frac{5}{2}}$ and $b_t \in (\frac{1}{4}\sqrt{a_t\kappa}, 2\sqrt{a_t\kappa})$.*

*Proof of Lemma 11.* First we check that the region defined is not empty. This is true since given $\kappa < \frac{1}{16}K^{-1}$, we have $\frac{1}{4}K^{-2}\kappa^{-1} > 4K^{-1} > K^{-\frac{5}{2}} > \kappa^{\frac{5}{2}}$.

To prove the claim we consider the inductive hypothesis

$$P(k): a_k \in (\kappa^{\frac{5}{2}}, a_0 - \tfrac{1}{16}(k-1)\kappa^{\frac{13}{2}}] \text{ and } |b_k| \in (\tfrac{1}{4}\sqrt{a_k\kappa}, 2\sqrt{a_k\kappa}).$$

Assume that $P(k)$ holds for some $k$. Since $a_k < a_0 < \frac{1}{4}K^{-2}\kappa^{-1}$, we have $|b_k| < 2\sqrt{a_k\kappa} < 2\sqrt{\frac{1}{4}K^{-2}\kappa^{-1}\kappa} = K^{-1}$. With $\kappa < \frac{1}{16}K^{-1}$ and $a_k < a_0 < \frac{1}{4}K^{-2}\kappa^{-1}$, we have $\kappa, b_k, a_k$ satisfying condition B.2 and condition B.5. Thus $a_{k+1} < a_k - b_k^2\kappa^3$, $a_{k+1} > a_k - 8b_k^2\kappa^3$ (by Lemma 5), and $|b_{k+1} - b_k| < \frac{1}{16}\sqrt{a_k\kappa}$ (by Lemma 10).

Observe that since $a_{k+1} > a_k - 8b_k^2\kappa^3$,

$$
\begin{aligned}
2\sqrt{a_{k+1}\kappa} &> 2\sqrt{(a_k - 8b_k^2\kappa^3)\kappa} \\
&> 2\sqrt{(a_k - 8(4a_k\kappa)\kappa^3)\kappa} \quad (\text{sinbe } b_k < 2\sqrt{a_k\kappa}) \\
&= 2\sqrt{(a_k\kappa)(1 - 32\kappa^4)} \\
&= 2\sqrt{1 - 32\kappa^4}\sqrt{a_k\kappa}
\end{aligned}
\tag{91}
$$

Note that with $\kappa < \frac{1}{16}K^{-1}$ where we assume $K > 128$, we have $\sqrt{1 - 32\kappa^4} > 0.99$ and hence $\sqrt{a_{k+1}\kappa} > 0.99\sqrt{a_k\kappa}$. Now we will show that $b_{k+1}$ will not leave $(\frac{1}{4}\sqrt{a_{k+1}\kappa}, 2\sqrt{a_{k+1}\kappa})$. There are three cases to consider:

1. When $|b_k| \in (\sqrt{a_k\kappa}, 2\sqrt{a_k\kappa})$, along with $|b_k| < K^{-1}$ we have condition B.3 satisfied and thus $|b_{k+1}| < |b_k|(1 - b_k^2)$ by Lemma 6. Since $\kappa < \frac{1}{16}K^{-1} < \frac{1}{2048}$, we have $\kappa^{\frac{1}{4}} < \frac{1}{4\sqrt{2}}$, and thus $\kappa^{\frac{7}{4}} > 4\sqrt{2}\kappa^2$. Moreover, since $a_k > \kappa^{\frac{5}{2}}$, we have $|b_k| > \sqrt{a_k\kappa} > \kappa^{\frac{7}{4}} > 4\sqrt{2}\kappa^2$. Squaring both sides we have $b_k^2 > 32\kappa^4$. Hence $1 - 32\kappa^4 > 1 - b_k^2 > 1 - 2b_k^2 + b_k^4 = (1 - b_k^2)^2$. The last inequality holds since $b_k^2 > b_k^4$ as $|b_k| < 1$. Now taking the square root on both sides we have $1 - b_k^2 < \sqrt{1 - 32\kappa^4}$. Since $|b_k| < 2\sqrt{a_k\kappa}$, combining with Eq. (91) and $1 - b_k^2 < \sqrt{1 - 32\kappa^4}$ we have $|b_{k+1}| < 2\sqrt{a_k\kappa}(1 - b_k^2) < 2\sqrt{1 - 32\kappa^4}\sqrt{a_k\kappa} < 2\sqrt{a_{k+1}\kappa}$, which gives the desired upper bound to $|b_{k+1}|$.

   Now we prove the lower bound for $|b_{k+1}|$. Since we know $|b_{k+1} - b_k| < \frac{1}{16}\sqrt{a_k\kappa}$, by triangle inequality, $|b_k| > \sqrt{a\kappa}$ implies $|b_{k+1}| > \frac{15}{16}\sqrt{a_k\kappa} > \frac{1}{4}\sqrt{a_k\kappa} > \frac{1}{4}\sqrt{a_{k+1}\kappa}$.

2. When $|b_k| \in [\frac{1}{2\sqrt{2}}\sqrt{a_k\kappa}, \sqrt{a_k\kappa}]$, since $|b_{k+1} - b_k| < \frac{1}{16}\sqrt{a_k\kappa}$, by triangle inequality we have $|b_{k+1}| \in [(\frac{1}{2\sqrt{2}} - \frac{1}{16})\sqrt{a_k\kappa}, \frac{17}{16}\sqrt{a_k\kappa}]$. Since $(\frac{1}{2\sqrt{2}} - \frac{1}{16})\sqrt{a_k\kappa} > \frac{1}{4}\sqrt{a_{k+1}\kappa}$ and $\frac{17}{16}\sqrt{a_k\kappa} < 2\sqrt{a_{k+1}\kappa}$ as $\sqrt{a_{k+1}\kappa} > 0.99\sqrt{a_k\kappa}$, we have $|b_{k+1}| \in (\frac{1}{4}\sqrt{a_k\kappa}, 2\sqrt{a_k\kappa})$.

3. When $|b_k| \in (\frac{1}{4}\sqrt{a_k\kappa}, \frac{1}{2\sqrt{2}}\sqrt{a_k\kappa})$, along with $|b_k| < K^{-1}$ we have condition B.4 satisfied, and thus $|b_{k+1}| > |b_k| + |b_k a_k\kappa| > |b_k|$. Since $\sqrt{a_{k+1}\kappa} < \sqrt{a_k\kappa}$, $|b_{k+1}| > \frac{1}{4}\sqrt{a_{k+1}\kappa}$.

   On the other side, since $|b_{k+1} - b_k| < \frac{1}{16}\sqrt{a_k\kappa}$, and $|b_k| < \frac{1}{2\sqrt{2}}\sqrt{a_k\kappa}$, by triangle inequality we have $|b_{k+1}| < (\frac{1}{16} + \frac{1}{2\sqrt{2}})\sqrt{a_k\kappa} < 2\sqrt{a_{k+1}\kappa}$. The last step holds since $\sqrt{a_{k+1}\kappa} > 0.99\sqrt{a_k\kappa}$.

Summarizing the three cases, we know that $b_{k+1} \in (\frac{1}{4}\sqrt{a_{k+1}\kappa}, 2\sqrt{a_{k+1}\kappa})$.

By the assumption of $P(k)$ we also have $a_k > \kappa^{\frac{5}{2}}$ and $a_k < a_0 - \frac{1}{16}(k-1)\kappa^{\frac{13}{2}}$. Since we know $b_k > \frac{1}{4}\sqrt{a_k\kappa}$, we have $b_k^2\kappa^3 > \frac{1}{16}a_k\kappa^4 > \frac{1}{16}\kappa^{\frac{13}{2}}$. Thus $a_{k+1} < a_k - b_k^2\kappa^3 < a_k - \frac{1}{16}\kappa^{\frac{13}{2}} < a_0 - \frac{1}{16}k\kappa^{\frac{13}{2}}$. Therefore unless $b_{k+1} < \kappa^{\frac{5}{2}}$, $P(k+1)$ holds. Note that there must be some $t$ such that $a_t < \kappa^{\frac{5}{2}}$ since when $t > 16a_0\kappa^{-\frac{13}{2}} + 1$, $a_0 - \frac{1}{16}(t-1)\kappa^{\frac{13}{2}} < 0$. We induct on $k$ from 1, the

base case holds by the initialization of $b_0$ and $a_0$. Let $T$ be the smallest $t$ such that $a_t < \kappa^{\frac{5}{2}}$, at which we terminate the induction. Then for all $t < T$, $a_t \in (\kappa^{\frac{5}{2}}, a_0]$ and $b_t \in (\frac{1}{4}\sqrt{a_t\kappa}, 2\sqrt{a_t\kappa})$. This concludes the proof. $\qquad\square$

**Corollary B.3.** *Following the initialization condition and notation of Lemma 11, if $a_0 > 2\kappa^{\frac{5}{2}}$, there exists some $\tau < T$ such that $a_t \in (\frac{3}{2}\kappa^{\frac{5}{2}}, 2\kappa^{\frac{5}{2}})$.*

*Proof of Corollary B.3.* We will follow the notations defined in the proof of Lemma 11. By definition of $T$, $P(t)$ holds for all $t < T$. Then for all $t < T$, we have $a_t > \kappa^{\frac{5}{2}}$, $b_t < K^{-1}$, and $a_{t+1} < a_t - b_t^2\kappa^3$. Hence $|a_{t+1} - a_t| < K^{-2}\kappa^3 < \frac{1}{2}\kappa^{\frac{5}{2}}$ since we assumed $K > 128$.

Since $a_0 > 2\kappa^{\frac{5}{2}}$ and $a_T < \kappa^{\frac{5}{2}} < \frac{3}{2}\kappa^{\frac{5}{2}}$, combining with $|a_{t+1} - a_t| < \frac{1}{2}\kappa^{\frac{5}{2}}$ we know that there must exist some $\tau$ such that $a_\tau \in (\frac{3}{2}\kappa^{\frac{5}{2}}, 2\kappa^{\frac{5}{2}})$. $\qquad\square$

**Corollary B.4.** *Following Lemma 11, if $a_0 \in (\frac{3}{2}\kappa^{\frac{5}{2}}, 2\kappa^{\frac{5}{2}})$, $T > \frac{1}{128}\kappa^{-4}$*

*Proof of Corollary B.4.* We will follow the notations defined in the proof of Lemma 11. By definition of $T$, $P(t)$ holds for all $t < T$. Then for all $t < T$, we have $a_t < a_0 = 2\kappa^{\frac{5}{2}}$ and $b_t < 2\sqrt{a_t\kappa} < 2\sqrt{2}\kappa^{\frac{7}{4}}$. It follows that $a_{t+1} > a_t - 8b_t^2\kappa^3 > a_t - 64\kappa^{\frac{13}{2}}$. Since $a_0 - a_T > \frac{3}{2}\kappa^{\frac{5}{2}} - \kappa^{\frac{5}{2}} = \frac{1}{2}\kappa^{\frac{5}{2}}$, we must have $T > \frac{1}{2}\kappa^{\frac{5}{2}}/64\kappa^{\frac{13}{2}} = \frac{1}{128}\kappa^{-4}$. $\qquad\square$

### B.5 PHASE II: CONVERGENCE ALONG THE PARABOLA

In Appendix B.4 we have shown that for a certain range of initializations, $(a, b)$ converges close to the parabola $2b^2 = a\kappa$ very fast. In this section, we will show that $(a, b)$ will slowly move along the parabola, and will eventually converge to a point with sharpness just below the EoS threshold $2/\eta = 2/\kappa^2$.

To facilitate the analysis, we define the residual $\xi \triangleq b^2 - \frac{1}{2}a\kappa - \frac{1}{16}\kappa^4$ and consider a small perturbation constant threshold $\delta = 0.04$.

Follow from Corollary B.1 we have that for any $\epsilon < 0.5$, for any $\kappa, a, b$ satisfying condition B.1,

$$
\begin{aligned}
\xi'' &= b''^2 - \frac{1}{2}a''\kappa - \frac{1}{16}\kappa^4 \\
&= \left(b^2 + 8ab\kappa - 16b^3\right)^2 \\
&\quad + \mathcal{O}(b)\left(\mathcal{O}(b^4) + \mathcal{O}(ab^2\kappa) + \mathcal{O}(ab^2\kappa^2) + \mathcal{O}(b^2\kappa^4) + \mathcal{O}(b\kappa^5) + \mathcal{O}(\epsilon b^2\kappa^3)\right) \\
&\quad - \frac{1}{2}a\kappa + 2b^2\kappa^4 + \kappa\left(\mathcal{O}(\epsilon b^2\kappa^3) + \mathcal{O}(b^3\kappa^3) + \mathcal{O}(b^2\kappa^4)\right) - \frac{1}{16}\kappa^4 \\
&= b^2 - 32b^4 + 16b^2a\kappa - \frac{1}{2}a\kappa + 2b^2\kappa^4 - \frac{1}{16}\kappa^4 \\
&\quad + \mathcal{O}(b^5) + \mathcal{O}(ab^3\kappa) + \mathcal{O}(a^2b^2\kappa^2) + \mathcal{O}(b^3\kappa^4) + \mathcal{O}(b^2\kappa^5) + \mathcal{O}(\epsilon b^3\kappa^3) + \mathcal{O}(\epsilon b^2\kappa^4). \\
&= \left(1 - 32b^2\right)\left(b^2 - \frac{1}{2}a\kappa - \frac{1}{16}\kappa^4\right) + \\
&\quad + \mathcal{O}(b^5) + \mathcal{O}(ab^3\kappa) + \mathcal{O}(a^2b^2\kappa^2) + \mathcal{O}(b^3\kappa^4) + \mathcal{O}(b^2\kappa^5) + \mathcal{O}(\epsilon b^3\kappa^3) + \mathcal{O}(\epsilon b^2\kappa^4) \\
&= \left(1 - 32b^2\right)\xi + \mathcal{O}(b^5) + \mathcal{O}(ab^3\kappa) + \mathcal{O}(a^2b^2\kappa^2) + \mathcal{O}(b^3\kappa^4) + \mathcal{O}(b^2\kappa^5) + \mathcal{O}(\epsilon b^3\kappa^3) + \mathcal{O}(\epsilon b^2\kappa^4).
\end{aligned}
\tag{92}
$$

When $|b| < \kappa$, the above expression can be further reduced to

$$
\xi'' = \left(1 - 32b^2\right)\xi + \mathcal{O}(b^5) + \mathcal{O}(ab^3\kappa) + \mathcal{O}(a^2b^2\kappa^2) + \mathcal{O}(b^3\kappa^4) + \mathcal{O}(\epsilon b^2\kappa^4). \tag{93}
$$

Hence there exists absolute constants $K$ such that for all $\epsilon < 0.5$, for all $a, b, \kappa$ satisfying condition B.1 and $|b| < \kappa$, we have $\xi'' = (1 - 32b^2)\xi + R_\xi$ where

$$
|R_\xi| < K\left|b^5\right| + K\left|ab^3\kappa\right| + K\left|a^2b^2\kappa^2\right| + K\left|b^3\kappa^4\right| + K\left|\epsilon b^2\kappa^4\right| \tag{94}
$$

Now fix $\delta = 0.04$, we will determine the regime such that $|R_\xi|$ is less than $\delta b^2 \kappa^4$.

**Lemma 12.** *For any $\kappa, a, b$ satisfying*

$$\kappa < \frac{1}{80\sqrt{2}}\delta K^{-1}, \quad |b| < 2\sqrt{2}\kappa^{\frac{7}{4}}, \quad |a| < 2\kappa^{\frac{5}{2}} \tag{95}$$

*where $\delta = 0.04$, we have*

$$\left|\xi'' - (1 - 32c^2)\xi\right| < \delta b^2 \kappa^4. \tag{96}$$

*Proof of Lemma 12.* Fix $\epsilon = \frac{1}{5}\delta K^{-1}$, claim that $\kappa, b, a$ in the given regime satisfies condition B.1. We check the conditions one by one:

(i) Since we assume $K > 128$, fixing $\epsilon = \frac{1}{5}\delta K^{-1}$ satisfies $\epsilon < 0.5$

(ii) With both $\delta$ and $K^{-1}$ less than 0, $\kappa < \frac{1}{80\sqrt{2}}\delta K^{-1} < 0.1$. Also $\frac{1}{80\sqrt{2}}\delta K^{-1} < \frac{1}{5}\delta K^{-1} = \epsilon < \epsilon^{\frac{1}{4}}$. Thus $\kappa < \min\{0.1, \epsilon^{\frac{1}{4}}\}$.

(iii) With $\epsilon = \frac{1}{5}\delta K^{-1}$ and $\kappa < \frac{1}{80\sqrt{2}}\delta K^{-1}$, $\epsilon\kappa^{-1} > 16\sqrt{2} > 2\kappa^{\frac{5}{2}} > |a|$. Thus $|a| < \epsilon\kappa^{-1}$.

(iv) Since $\epsilon\kappa^{-1} > 16\sqrt{2}$ and $\kappa < 1$, $\epsilon\kappa^{-2}/5 > \frac{16}{5}\sqrt{2} > 1 > 2\sqrt{2}\kappa^{\frac{7}{4}}$. Thus $|b| < \min\{1, \epsilon\kappa^{-2}/5\}$.

Thus Eq. (94) applies, and we only need to bound every term on its RHS by $\frac{1}{5}\delta b^2 \kappa^4$ to complete the proof. We will do that term by term.

(i) Since $\kappa < \frac{1}{80\sqrt{2}}\delta K^{-1} < 1$, we have $\kappa^{\frac{5}{4}} < \frac{1}{80\sqrt{2}}\delta K^{-1}$. Multiplying $16\sqrt{2}\kappa^4$ on both sides we have $16\sqrt{2}\kappa^{\frac{21}{4}} < \frac{1}{5}\delta K^{-1}\kappa^4$. Note that since $|b| < 2\sqrt{2}\kappa^{\frac{7}{2}}$, $|b^3| < 16\sqrt{2}\kappa^{\frac{21}{4}}$, so $|b^3| < \frac{1}{5}\delta K^{-1}\kappa^4$. Multiplying $Kb^2$ on both sides gives $K|b^5| < \frac{1}{5}\delta b^2 \kappa^4$.

(ii) Since $\kappa^{\frac{5}{4}} < \frac{1}{80\sqrt{2}}\delta K^{-1} < \frac{1}{20\sqrt{2}}\delta K^{-1}$, multiplying $4\sqrt{2}\kappa^3$ on both sides we have $4\sqrt{2}\kappa^{\frac{17}{4}} < \frac{1}{5}\delta K^{-1}\kappa^3$. Note that since $|b| < 2\sqrt{2}\kappa^{\frac{7}{4}}$ and $|a| < 2\kappa^{\frac{5}{2}}$, $|ab| < 4\sqrt{2}\kappa^{\frac{17}{4}}$, we have $|ab| < \frac{1}{5}\delta K^{-1}\kappa^3$. Multiplying $Kb^2\kappa$ on both sides gives $K|ab^3\kappa| < \frac{1}{5}\delta b^2 \kappa^4$.

(iii) Since $\kappa < \frac{1}{80\sqrt{2}}\delta K^{-1} < 1$, we have $\kappa^3 < \kappa < \frac{1}{20}\delta K^{-1}$. Multiplying $4\kappa^2$ on both side gives $4\kappa^5 < \frac{1}{5}K^{-1}\delta\kappa^2$. Since $|a| < 2\kappa^{\frac{5}{2}}$, we have $a^2 < 4\kappa^5 < \frac{1}{5}K^{-1}\delta\kappa^2$. Multiplying $Kb^2\kappa^2$ on both side, we have $Ka^2b^2\kappa^2 < \frac{1}{5}\delta b^2 \kappa^4$.

(iv) Since $\kappa < \frac{1}{2}$, $2\kappa^{\frac{5}{2}} < \kappa$. Thus $|b| < 2\kappa^{\frac{5}{2}} < \kappa < \frac{1}{80\sqrt{2}}\delta K^{-1} < \frac{1}{5}\delta K^{-1}$. Multiplying $Kb^2\kappa^4$ on both side gives $K|b^3\kappa^4| < \frac{1}{5}\delta b^2 \kappa^4$.

(v) Since we fixed $\epsilon = \frac{1}{5}\delta K^{-1}$, $K\epsilon b^2 \kappa^4 = \frac{1}{5}\delta b^2 \kappa^4$.

Therefore we have

$$|R_\xi| < K\left|b^5\right| + K\left|ab^3\kappa\right| + K\left|a^2b^2\kappa^2\right| + K\left|b^3\kappa^4\right| + K\left|\epsilon b^2\kappa^4\right| < \delta b^2 \kappa^4. \tag{97}$$

Plugging back to $\xi'' = (1 - 32b^2)\xi + R_\xi$ completes the proof. $\qquad\square$

Here we restate the condition for Lemma 13:

**Condition B.6.** With $\delta = 0.04$ and $K > 512$,

$$\kappa < \frac{1}{80\sqrt{2}}\delta K^{-1}, \quad |b| < 2\sqrt{2}\kappa^{\frac{7}{4}}, \quad |a| < 2\kappa^{\frac{5}{2}}. \tag{98}$$

**Corollary B.5.** *For any $\kappa, a, b$ satisfying condition B.6, if $|\xi| > \frac{1}{16}\delta\kappa^4$, we have $|\xi''| < (1 - 4\kappa^{\frac{7}{2}})|\xi|$.*

*Proof of Corollary B.5.* Since condition B.6 holds, by Lemma 13 we have

$$|\xi''| < |(1 - 32b^2)\xi| + \delta b^2 \kappa^4. \tag{99}$$

Thus if $|\xi| > \frac{1}{16}\delta\kappa^4$, it follows that

$$
\begin{aligned}
|\xi''| &< |(1 - 32b^2)\xi| + \delta b^2 \kappa^4 \\
&= (1 - 32b^2)|\xi| + \delta b^2 \kappa^4 && \text{(Since } 32b^2 < 1\text{)} \\
&= |\xi| - 16b^2|\xi| - (16b^2|\xi| - \delta b^2 \kappa^4) \\
&< |\xi| - 16b^2|\xi| - b^2(16(\tfrac{1}{16}\delta\kappa^4) - \delta\kappa^4) && \text{(Since } |\xi| > \tfrac{1}{16}\delta\kappa^4\text{)} \\
&= |\xi| - 16b^2|\xi| \\
&< |\xi| - 4\kappa^{\frac{7}{2}}|\xi| && \text{(Since } |b| > \tfrac{1}{2}\kappa^{\frac{7}{4}}\text{)} \\
&= (1 - 4\kappa^{\frac{7}{2}})|\xi|.
\end{aligned}
\tag{100}
$$

$\square$

### B.5.1 PHASE II STAGE 1

In this stage we will show that after $|b|$ gets close to $\sqrt{a\kappa/2}$ while $a$ decreases to around $2\kappa^{\frac{5}{2}}$ from Phase I of the convergence, the residual of $(b, a)$ to the parabola will further decrease to below $\frac{1}{8}\delta\kappa^4$.

**Lemma 13.** *For any $\kappa < \frac{1}{80\sqrt{2}}\delta K^{-1}$, for all initialization $(b_0, a_0)$ such that $a_0 \in (\frac{3}{2}\kappa^{\frac{5}{2}}, 2\kappa^{\frac{5}{2}})$ and $|b_0| \in (\frac{1}{4}\sqrt{a_0\kappa}, 2\sqrt{a_0\kappa})$. Let $T$ be the time that $a$ exits $(\kappa^{\frac{5}{2}}, 2\kappa^{\frac{5}{2}})$ as characterized in Lemma 11 and Corollary B.4. There exists some $\tau < T$ such that $\xi_\tau < \frac{1}{8}\delta\kappa^4$.*

*Proof of Lemma 13.* First note that since $\kappa < \frac{1}{80\sqrt{2}}\delta K^{-1} < \frac{1}{16}K^{-1}$, the initialization condition given is a subset of the valid initialization for Lemma 11. Thus for all $t < T$, we have $a_t \in (\kappa^{\frac{5}{2}}, 2\kappa^{\frac{5}{2}})$ and $|b_t| \in (\frac{1}{4}\sqrt{a_t\kappa}, 2\sqrt{a_t\kappa})$. Hence $|b_t| < 2\sqrt{2\kappa^{\frac{5}{2}}\kappa} = 2\sqrt{2}\kappa^{\frac{7}{4}}$ and $|b_t| > \sqrt{\frac{1}{4}\kappa^{\frac{5}{2}}\kappa} = \frac{1}{2}\kappa^{\frac{7}{4}}$. Therefore $(b_t, a_t)$ satisfies condition B.6 and $\left|\xi_{t+1} - (1 - 32b_t^2)\xi_t\right| < \delta b_t^2 \kappa^4$. Also note that with $K > 512$ and $\delta = 0.05$, $b_t^2 < 8\kappa^{\frac{7}{2}} < 8\kappa < \frac{1}{10\sqrt{2}}\delta K^{-1} < 1$.

For all $t < T$ that $|\xi_t| > \frac{1}{16}\delta\kappa^4$, by Corollary B.5 we have $|\xi_{t+1}| < (1 - 4\kappa^{\frac{7}{2}})|\xi_t|$. At the initialization, we have $|\xi_1| \leq |b_0^2| + |\frac{1}{2}a_0\kappa| + \frac{1}{16}\kappa^4 < \frac{9}{2}a_0\kappa + \frac{1}{16}\kappa^4 < 9\kappa^{\frac{7}{2}} + \frac{1}{16}\kappa^4 < 10\kappa^{\frac{7}{2}}$. Thus for all $\tau < T$ such that for all $t < \tau, |\xi_t| > \frac{1}{16}\delta\kappa^4$, we have $|\xi_\tau| < (1 - 4\kappa^{\frac{7}{2}})^\tau 10\kappa^{\frac{7}{2}}$. Now we only need to show that $|\xi_\tau|$ will decrease sufficiently fast.

Consider

$$\tau = \left\lceil \log_{(1 - 4\kappa^{\frac{7}{2}})}\left(\frac{\frac{1}{8}\delta\kappa^4}{10\kappa^{\frac{7}{2}}}\right)\right\rceil \geq \frac{\log(\frac{1}{80}\delta\kappa^{\frac{1}{2}})}{\log(1 - 4\kappa^{\frac{7}{2}})} = \frac{\log(80\delta^{-1}\kappa^{-\frac{1}{2}})}{-\log(1 - 4\kappa^{\frac{7}{2}})}. \tag{101}$$

Since $\log(1 - 4\kappa^{\frac{7}{2}}) < -4\kappa^{\frac{7}{2}}$ by a second order Taylor expansion, we have $\tau < \log(80\delta^{-1}\kappa^{-\frac{1}{2}})/4\kappa^{\frac{7}{2}}$. Substituting $\delta = 0.04$ in the expression, $\log(80\delta^{-1}\kappa^{-\frac{1}{2}}) = \log(2000) + \log(\kappa^{-\frac{1}{2}}) < 8 + \log(\kappa^{-\frac{1}{2}})$. Observe that for all $x > 175$, we have $8 + \log(x) < \sqrt{x}$.

Since $\kappa < \frac{1}{80\sqrt{2}}\delta K^{-1} < \frac{1}{512 \times 2000\sqrt{2}} < \frac{1}{175^2}$, let $x = \kappa^{-\frac{1}{2}}$, we have $8 + \log(\kappa^{-\frac{1}{2}}) < \kappa^{-\frac{1}{4}}$. Hence $\log(80\delta^{-1}\kappa^{-\frac{1}{2}}) < \kappa^{-\frac{1}{4}}$ and $\tau < \kappa^{-\frac{1}{4}}/4\kappa^{\frac{7}{2}} = \frac{1}{4}\kappa^{-\frac{15}{4}}$. Recall from Corollary B.4 we have $T > \frac{1}{512}\kappa^{-4}$. If we assume $K > 512$, then $\kappa < \frac{1}{512 \times 2000\sqrt{2}} < \frac{1}{32^4}$, and thus $\kappa^{-\frac{1}{4}} > 32$ and $T > \frac{1}{512}\kappa^{-4} > \frac{1}{4}\kappa^{-\frac{15}{4}} > \tau$.

Since $\tau < T$, $a_\tau > \kappa^{\frac{5}{2}}$. If for all $t < \tau, |\xi_t| > \frac{1}{16}\delta\kappa^4$, then following the analysis above, by definition of $\tau$ we have $|\xi_\tau| < \frac{1}{8}\delta\kappa^4$. If there exists some $t < \tau$ that $|\xi_t| \leq \frac{1}{16}\delta\kappa^4$, then setting $\tau = t$ directly completes the proof. $\square$

### B.5.2 Phase II Stage 2

In this phase, we show that once $|\xi|$ is smaller than $\frac{1}{8}\delta\kappa^4$, $a$ will decrease slowly while $|\xi|$ does not increase beyond $\frac{1}{8}\delta\kappa^4$.

**Lemma 14.** *For all $\kappa < \frac{1}{80\sqrt{2}}\delta K^{-1}$, for all initialization $(a_0, b_0)$ satisfying $a_1 \in (\kappa^{\frac{5}{2}}, 2\kappa^{\frac{5}{2}})$ and $|\xi_1| < \frac{1}{8}\delta\kappa^4$, there exists some $T < 48\delta^{-1}\kappa^{-\frac{9}{2}}+1$ such that $a_T < -\frac{1}{8}(1-3\delta)\kappa^3$ and for all $t < T$, $a_t > -\frac{1}{8}(1-3\delta)\kappa^3$ and $|\xi_k| < \frac{1}{8}\delta\kappa^4$.*

We will prove the claim using induction. Consider the inductive hypothesis

$$P(k): |\xi_k| < \tfrac{1}{8}\delta\kappa^4 \text{ and } a_k \in (-\tfrac{1}{8}(1-3\delta)\kappa^3, a_1 - \tfrac{1}{16}\delta\kappa^7 k].$$

Note that $P(0)$ holds directly from construction, so we proceed to the inductive step. Assuming $P(k)$ holds, We will show that either $a_k < -\frac{1}{8}(1-3\delta)\kappa^3$ or $P(k+1)$ holds.

First we verify that $b_k$, $a_k$ satisfies condition B.6. Since $\delta = 0.05$, $|-\frac{1}{8}(1-3\delta)\kappa^3| < |\frac{1}{8}\kappa^3| < 2\kappa^{\frac{5}{2}}$. Also since $a_1 < 2\kappa^{\frac{5}{2}}$ as required, we have $|a_t| < \max\{|-\frac{1}{8}(1-3\delta)\kappa^3|, |a_1 - \frac{1}{16}\delta\kappa^7 k|\} < 2\kappa^{\frac{5}{2}}$. Since $|\xi_k| < \frac{1}{8}\delta\kappa^4$, we have $b_k^2 < \frac{1}{2}a_k\kappa + \frac{1}{16}\kappa^4 + \frac{1}{8}\delta\kappa^4 < \frac{1}{2}(2\kappa^{\frac{5}{2}}) + \kappa^4 < 2\kappa^{\frac{7}{2}}$. Hence $|b_t| < 2\sqrt{2}\kappa^{\frac{7}{4}}$ as required. Therefore we have $|\xi_{k+1} - (1 - 32b^2)\xi_k| < \delta b_k^2 \kappa^4$.

Next we establish the lower bounds for $|b_k|$, which will give lower bound for the movement of $a$. Since $a_k > -\frac{1}{8}(1-3\delta)\kappa^3$ and $|\xi_k| = |b_k^2 - (\frac{1}{2}a_k\kappa + \frac{1}{16}\kappa^4)| < \frac{1}{8}\delta\kappa^4$, we have

$$c_k^2 > \tfrac{1}{2}d_k\kappa + \tfrac{1}{16}\kappa^4 - \tfrac{1}{8}\delta\kappa^4 > \tfrac{1}{2}\left(-\tfrac{1}{8}(1-3\delta)\kappa^3\right)\kappa + \tfrac{1}{16}\kappa^4 - \tfrac{1}{8}\delta\kappa^4 = \tfrac{1}{16}\delta\kappa^4. \tag{102}$$

Since $|b_k| < K^{-1}$ and $|a_k| < K^{-1}$, condition B.2 is satisfied and we have $a_{k+1} < a_k - b_k^2\kappa^3 < a_k - \frac{1}{16}\delta\kappa^4\kappa^3$. Given that $a_k \leq a_1 - \frac{1}{16}\delta\kappa^7 k$ by the inductive hypothesis, $a_{k+1} \leq a_1 - \frac{1}{16}\delta\kappa^7(k+1)$.

What remains to show for the inductive step is that $|\xi_{k+1}| < \frac{1}{8}\delta\kappa^3$. There are two cases to consider. When $|\xi_k| \in (\frac{1}{16}\delta\kappa^4, \frac{1}{8}\delta\kappa^4)$, by Corollary B.5 we know $|\xi_{k+1}| < (1 - 4\kappa^{\frac{7}{2}})|\xi_k| < |\xi_k| < \frac{1}{8}\delta\kappa^4$. When $|\xi_k| \leq \frac{1}{16}\delta\kappa^4$, since $|\xi_{k+1} - (1 - 32b^2)\xi_k| < \delta b_k^2\kappa^4$, we have

$$|\xi_{k+1} - \xi_k| < \delta b_k^2\kappa^4 + 32b_k^2|\xi_k| \leq \delta b_k^2\kappa^4 + 32b_k^2(\tfrac{1}{16}\delta\kappa^4) = 3\delta b_k^2\kappa^4. \tag{103}$$

Since $b_k^2 < 2\kappa^{\frac{7}{2}} < \frac{1}{48}$, $|\xi_{k+1} - \xi_k| < \frac{1}{48}3\delta\kappa^4 = \frac{1}{16}\delta\kappa^4$. Since $|\xi_k| \leq \frac{1}{16}\delta\kappa^4$, we have $|\xi_{k+1}| \leq |\xi_{k+1} - \xi_k| + |\xi_k| \leq \frac{1}{8}\delta\kappa^4$ as desired.

In summary we have $P(k)$ implies $P(k+1)$ unless $a_{k+1} < -\frac{1}{8}(1-3\delta)\kappa^3$. Note that there must exists some $t$ such that $a_{t+1} < -\frac{1}{8}(1-3\delta)\kappa^3$ since for any $\tau \geq 48\delta^{-1}\kappa^{-\frac{9}{2}} + 1$, if the induction proceed to $P(\tau)$, then $a_1 - \frac{1}{16}\delta\kappa^7(\tau - 1) < a_1 - 3\kappa^{\frac{5}{2}} < -\kappa^{\frac{5}{2}} < -\frac{1}{8}(1-3\delta)\kappa^3$, which violates $P(\tau)$. Let $T < 48\delta^{-1}\kappa^{-\frac{9}{2}} + 1$ be the first $t$ such that $a_t < -\frac{1}{8}(1-3\delta)\kappa^3$, then by construction we have for all $t < T$, $P(t)$ holds. This completes the proof of the lemma.

**Corollary B.6.** *Following Lemma 14, $a_{T-1} \in (-\frac{1}{8}(1-3\delta)\kappa^3, -\frac{1}{10}(1+2\delta)\kappa^3)$.*

*Proof of Corollary B.6.* Denote $T - 1$ by $\tau$, by definition of $T$, we know $P(\tau)$ holds and therefore we have $b_\tau^2 < 2\kappa^{\frac{7}{2}}$, $a_\tau > -\frac{1}{8}(1-3\delta)\kappa^3$ and $a_T > a_\tau - 5b_\tau^2\kappa^3$. Combining above we have $a_T > a_\tau - 10\kappa^{\frac{7}{2}}\kappa^3$, so $a_\tau < a_T + 10\kappa^{\frac{13}{2}} < -\frac{1}{8}(1-3\delta)\kappa^3 + 10\kappa^{\frac{13}{2}}$. Note that since we set $\delta = 0.04$, we have

$$-\tfrac{1}{10}(1+2\delta)\kappa^3 - (-\tfrac{1}{8}(1-3\delta)\kappa^3) = (\tfrac{1}{40} - (\tfrac{2}{10} + \tfrac{3}{8})\delta)\kappa^3 = \tfrac{1}{500}\kappa^3. \tag{104}$$

Since $\kappa^{-\frac{7}{2}} > K^{\frac{7}{2}} > 5000$, we have $10\kappa^{\frac{13}{2}} < \frac{1}{500}\kappa^3 = -\frac{1}{10}(1+2\delta)\kappa^3 - (-\frac{1}{8}(1-3\delta)\kappa^3)$. Adding $-\frac{1}{8}(1-3\delta)\kappa^3$ on both sides, we have $a_\tau < -\frac{1}{8}(1-3\delta)\kappa^3 + 10\kappa^{\frac{13}{2}} < -\frac{1}{10}(1+2\delta)\kappa^3$. Combining with $a_\tau > -\frac{1}{8}(1-3\delta)\kappa^3$ concludes the proof. $\square$

### B.5.3 PHASE II STAGE 3

Here we state the lemma which proves the final convergence of the two step trajectory. The proof is very similar to that of Lemma 7 except $a$ is negative now and $|b|$ is decreasing.

**Lemma 15** (Final Convergence). *For all $\kappa < \frac{1}{16}K^{-1}$, for all $a_0, b_0$ satisfying $a_0 \in (-\frac{1}{8}(1 - 3\delta)\kappa^3, -\frac{1}{10}(1 + 2\delta)\kappa^3)$ and $|\xi_0| = |b_0^2 - \frac{1}{2}a_0\kappa - \frac{1}{16}\kappa^4| < \frac{1}{8}\delta\kappa^4$, for all $\epsilon > 0$, there exists some $T < 25\log(\epsilon^{-1})$ such that for all $t \geq T$, $|b_t| < \epsilon$ and $a_t \in (-\frac{5}{3}\kappa^3, -\frac{1}{10}\kappa^3)$.*

*Proof.* We will prove the claim using induction. Consider the inductive hypothesis

$$P(k): |b_k| \leq |b_0|(1 - \frac{1}{10}(1 + 2\delta))^k, a_0 - a_k \in [0, 8\sqrt{2}\kappa(|b_0| - |b_k|)).$$

When $k = 0$, the statement holds trivially, so we proceed to the inductive step. Assume $P(k)$ holds for some $k \in \mathbb{N}$, we want to show that $P(k+1)$ holds as well.

First we check that $(a_k, b_k)$ with $\kappa < \frac{1}{16}K^{-1}$ satisfies condition B.4.

Since $(1 - \frac{1}{10}(1 + 2\delta)) < 1$, by the inductive hypothesis and the initialization condition on $|b_0|$ we have $|b_k| < |b_0|$. It is also from the inductive hypothesis that $|a_k| > |a_0|$, thus to show $|b_k| < \frac{1}{2\sqrt{2}}\sqrt{|a_k|\kappa}$, one only need to show for $k = 0$ case, which is equivalent to $b_0^2 < -\frac{1}{8}a_0\kappa$.

From the initialization condition, $\xi_0 = |b_0^2 - \frac{1}{2}a_0\kappa - \frac{1}{16}\kappa^4| < \frac{1}{8}\delta\kappa^4$. Since $a_0 < -\frac{1}{10}(1 + 2\delta)\kappa^3$, we must have $\frac{1}{2}a_0 + \frac{1}{16}\kappa^4 < 0 \leq b_0^2$. It follows that

$$
\begin{aligned}
b_0^2 &< \frac{1}{2}a_0\kappa + \frac{1}{16}\kappa^4 + \frac{1}{8}\kappa^4 \\
&= \frac{5}{8}a_0\kappa + \frac{1}{16}(1 + 2\delta) - \frac{1}{8}a_0\kappa \\
&< \frac{5}{8}(-\frac{1}{10}(1 + 2\delta)\kappa^3)\kappa + \frac{1}{16}(1 + 2\delta)\kappa^4 - \frac{1}{8}a_0\kappa \quad \text{(Since } a_0 < -\frac{1}{10}(1 + 2\delta)\kappa^3) \quad (105) \\
&= -\frac{1}{16}(1 + 2\delta)\kappa^4 + \frac{1}{16}(1 + 2\delta)\kappa^4 - \frac{1}{8}a_0\kappa \\
&= \frac{1}{8}|a_0|\kappa.
\end{aligned}
$$

From the initialization condition we have $a_0 < -\frac{1}{10}(1 + 2\delta)\kappa^3 = -0.108\kappa^3 < -\frac{1}{16}\kappa^3$, so $|a_k| > |a_0| > \frac{1}{16}\kappa^3$. For upper-bound on $|a_k|$ we note that $a_0 > -\frac{1}{8}(1 - 3\delta)\kappa^3 > -\kappa^3$ by initialization, combining with the inductive hypothesis we have $a_k > -\kappa^3 - 2\kappa(|b_0| - |b_t|) > -\kappa^3 - 2|b_0|$. Since $b_0^2 < \frac{1}{8}|a_0|\kappa < \frac{1}{64}(1 - 3\delta)\kappa^4 < (\frac{1}{8}\kappa^2)^2$, $|b_0| < \frac{1}{8}\kappa^2$, so $a_k > -\kappa^3 - \frac{1}{4}\kappa^3$ and hence $|a_k| < \frac{5}{4}\kappa^3 < K^{-2}\kappa^{-1}$ since we assumed $\kappa < \frac{1}{16}K^{-1}$. Therefore we have shown that $\kappa, a_k, b_k$ satisfies condition B.4 and by Lemma 7 we have $|b_{k+1}| < |b_k + a_kb_k\kappa|$.

Since $1 + a_k\kappa > 0$ as $|a_k| < \frac{1}{8}\kappa^3$ and $\kappa < K^{-1} < \frac{1}{512}$, we may write the update of $b_k$ as $|b_{k+1}| < |b_k|(1 + a_k\kappa)$. Since $a_k < a_0 < -\frac{1}{10}(1 + 2\delta)\kappa^3$, we have $|b_{k+1}| < |b_k|(1 - \frac{1}{10}(1 + 2\delta)\kappa^3)$. Combining with the inductive hypothesis that $|b_k| < |b_0|(1 - \frac{1}{10}(1 + 2\delta)\kappa^3)^k$, we have $|b_{k+1}| < |b_0|(1 - \frac{1}{10}(1 + 2\delta)\kappa^3)^{k+1}$.

Since condition B.4 is stronger than condition B.2, by Lemma 5 we have $a_k - a_{k+1} < 8b_k^2\kappa^3$. Combining with $|b_k| - |b_{k+1}| > |a_k||b_k|\kappa$, we have

$$
\begin{aligned}
\frac{a_k - a_{k+1}}{|b_k| - |b_{k+1}|} &< \frac{8b_k^2\kappa^3}{|a_kb_k|\kappa} \\
&= \frac{8|b_k|\kappa^2}{|a_k|} \\
&< \frac{8\frac{1}{2\sqrt{2}}\sqrt{|a_k|\kappa}\kappa^2}{|a_k|} \quad \text{(Since } |b_k| < \frac{1}{2\sqrt{2}}\sqrt{|a_k|\kappa}) \quad (106) \\
&= 2\sqrt{2}|a_k|^{-\frac{1}{2}}\kappa^{\frac{5}{2}} \\
&< 2\sqrt{2}(\frac{1}{16}\kappa^3)^{-\frac{1}{2}}\kappa^{\frac{5}{2}} \quad \text{(Since} |a_k| > \frac{1}{16}\kappa^3) \\
&< 8\sqrt{2}\kappa.
\end{aligned}
$$

Since the inductive hypothesis gives $(a_0 - a_k)/(|b_0| - |b_k|) < 8\sqrt{2}\kappa$, it follows by the mediant inequality that $(a_0 - a_{k+1})/(|b_0| - |b_{k+1}|) < 8\sqrt{2}\kappa$.

Thus we have shown $P(k) \to P(k+1)$, and by induction we know $P(k)$ holds for any $k \in \mathbb{N}$. Now we we can wrap up the convergence analysis leveraging this property.

Since for all $t$, $|b_t| \le |b_0|(1 - \frac{1}{10}(1 + 2\delta))^k$, for any $\epsilon > 0$ we may pick $T > \log_{(1-\frac{1}{10}(1+2\delta))}(\epsilon/|b_0|)$ such that for all $t > T$, $|b_t| < \epsilon$. Note that since $|b_0| < \frac{1}{8}\kappa^2 < 1$ and $1 - \frac{1}{10}(1 + 2\delta)$, we have

$$T < \log_{(1-\frac{1}{10}(1+2\delta))}(\epsilon/|b_0|) < \frac{\log(\epsilon)}{\log(1 - \frac{1}{10}(1.08))} < 25\log(\epsilon^{-1}). \tag{107}$$

For the region of final convergence, for any $t$ we know that from $P(t)$ that $a_0 - a_t \in [0, 8\sqrt{2}\kappa(|b_0| - |b_t|))$, so we have $a_t > a_0 - 8\sqrt{2}\kappa|b_0|$. Since we know $|b_0| < \frac{1}{8}\kappa^2$ and $a_0 > -\frac{1}{8}\kappa^3$ by initialization, it follows that $a_t > -\frac{1}{8} - \sqrt{2}\kappa^2 > -\frac{5}{3}\kappa^3$. The upper bound of $a_t < -\frac{1}{10}(1 + 2\delta)\kappa^3 < \frac{1}{10}\kappa^3$ is trivial since $a$ is monotonically decreasing.

Thus in summary we have shown that for any $\epsilon > 0$, there exists some $T < 25\log(\epsilon^{-1})$ such that for all $t > T$, $|b_t| < \epsilon$ and $a_t \in (-\frac{5}{3}\kappa^3, -\frac{1}{10}\kappa^3)$. $\qquad\square$

### B.6 Proof of Theorem 3.1 and its Corollaries

With all the lemmas ready, we may now prove Theorem 3.1 and its corollaries.

We first restate Theorem 3.1 here.

**Theorem 3.1** (Sharpness Concentration). *For a large enough absolute constant $K$, suppose $\kappa < \frac{1}{2000\sqrt{2}}K^{-1}$, and the initialization $(a_0, b_0)$ satisfies $a_0 \in (12\kappa^{\frac{5}{2}}, \frac{1}{4}K^{-2}\kappa^{-1})$ and $b_0 \in (-K^{-1}, K^{-1})\backslash\{0\}$. Consider the GD trajectory characterized in Eq. (6) with fixed step size $\kappa^2$ from $(a_0, b_0)$, for any $\epsilon > 0$ there exists $T = \mathcal{O}(K^{-2}\kappa^{-\frac{15}{2}} + \log(\epsilon^{-1}) + \log(|b_0|^{-1})\kappa^{-\frac{7}{2}})$ such that for all $t > T$, $|b_t| < \epsilon$ and $a_t \in (-\frac{5}{3}\kappa^3, -\frac{1}{10}\kappa^3)$.*

The proof for the main theorem is very simple after we have all the lemmas as discussed above.

*Proof of Theorem 3.1.* We consider any initialization $(a_0, b_0)$ satisfying $a_0 \in (12\kappa^{\frac{5}{2}}, \frac{1}{4}K^{-2}\kappa^{-1})$ and $b_0 \in (-K^{-1}, K^{-1})\backslash\{0\}$. We abuse the notation to let $a_t$ and $b_t$ be the value of $a$ and $b$ after the $t$-th two step update from $a_0$ and $b_0$.

If $|b_0| \ge 2\sqrt{a_0\kappa}$, then by Lemma 8 there exists some $\tau_1 < \kappa^{-4}$ such that $a_{\tau_1} \in (2\kappa^{\frac{5}{2}}, \frac{1}{4}K^{-2}\kappa^{-1})$ and $|b_{\tau_1}| \in (\sqrt{a_{\tau_1}\kappa}, 2\sqrt{a_{\tau_1}\kappa})$. If $|b_0| \le \frac{1}{4}\sqrt{|a_0\kappa|}$, by Lemma 9 there exists $\tau_2 < \frac{1}{2}\log(|b_0|^{-1})\kappa^{-\frac{7}{2}}$ such that $|b_{\tau_2}| \in (\frac{1}{4}\sqrt{a_{\tau_2}\kappa}, \frac{1}{2}\sqrt{a_{\tau_2}\kappa})$ and $a_{\tau_2} \in (2\kappa^{\frac{5}{2}}, \frac{1}{4}K^{-2}\kappa^{-1})$. Thus there exists some $T_1 = \mathcal{O}(\kappa^{-4} + \log(|b_0|^{-1})\kappa^{-\frac{7}{2}})$ such that $a_{T_1} \in (2\kappa^{\frac{5}{2}}, \frac{1}{4}K^{-2}\kappa^{-1})$ and $b_{T_1} \in (\frac{1}{4}\sqrt{a_{T_1}\kappa}, 2\sqrt{a_{T_1}\kappa})$.

Now by Lemma 11 and Corollary B.3, we know that there exists some $\tau_3 \le a_0\kappa^{-\frac{13}{2}}$ such that within $\tau_3$ two-step updates from $(a_{T_1}, b_{T_1})$ we have $a_{T_1+\tau_3} \in (\frac{3}{2}\kappa^{\frac{5}{2}}, 2\kappa^{\frac{5}{2}})$ and $b_{T_1+\tau_3} \in (\frac{1}{4}\sqrt{a_{T_1+\tau_3}\kappa}, 2\sqrt{a_{T_1+\tau_3}\kappa})$. Let $T_2 = T_1 + \tau_3$.

This completes phase 1 of convergence.

For phase 2, since $a_{T_2} \in (\frac{3}{2}\kappa^{\frac{5}{2}}, 2\kappa^{\frac{5}{2}})$ and $|b_{T_2}| \in (\frac{1}{4}\sqrt{a_{T_2}\kappa}, 2\sqrt{a_{T_2}\kappa})$, by Lemma 13, there exists $\tau_4 < |a_{T_2}|\kappa^{-\frac{13}{2}} < 2\kappa^{-6}$ such that within $\tau_4$ steps from $(a_{T_2}, b_{T_2})$, we have $a_{T_2+\tau_4} \in (\kappa^{-\frac{5}{2}}, 2\kappa^{-\frac{5}{2}})$ and $|\xi_{T_2+\tau_4}| < \frac{1}{8}\delta\kappa^4$ where we fixed $\delta = 0.04$. Let $T_3 = T_2 + \tau_4$.

After the residual $|\xi|$ decreases to less than $\frac{1}{8}\delta\kappa^4$ with $T_3$ two-step updates, by Lemma 14 and Corollary B.6 there exists some $\tau_5 < 48\delta_{-1}\kappa^{-\frac{9}{2}}$ such that $a_{T_3+\tau_5} \in (-\frac{1}{8}(1-3\delta)\kappa^3, -\frac{1}{10}(1+2\delta)\kappa^3)$ while $|\xi_{T_3+\tau_5}| < \frac{1}{8}\delta\kappa^4$. Let $T_4 = T_3 + \tau_5$.

Finally, by Lemma 15 we have that starting from $(a_{T_4}, b_{T_4})$, there exists some $\tau_6 < 25 \log(\epsilon^{-1})$ that for any $t > T_4 + \tau_6$, $|b_t| < \epsilon$ and $a_t \in (-\frac{5}{3}\kappa^3, -\frac{1}{10}\kappa^3)$. Thus we have the trajectory converging to some minima with $a \in (-\frac{5}{3}\kappa^3, -\frac{1}{10}\kappa^3)$.

Finally we bound the total number of steps required for convergence. Since $|a_0| < \frac{1}{4}K^{-2}\kappa^{-1}$, we have $\tau_3 < \frac{1}{4}K^{-2}\kappa^{-\frac{15}{2}}$. Moreover, since $\kappa < K^{-1}$, we have $\kappa^{-4} = \mathcal{O}(K^{-2}\kappa^{-\frac{15}{2}})$. Since $T_5 = T_1 + \tau_3 + \tau_4 + \tau_5 + \tau_6$, we have

$$T = \mathcal{O}\left(K^{-2}\kappa^{-\frac{15}{2}} + \log(|b_0|^{-1})\kappa^{-\frac{7}{2}} + \log(\epsilon^{-1})\right). \tag{108}$$

This completes the proof of the theorem. □

### B.6.1 Proof of Corollary 3.1

Before we proceed to prove Corollary 3.1, we first show a simple lemma on the approximity of $x$ and $c(x, y) \triangleq \sqrt{x^2 - y^2}$ when $x$ is large and $|1 - xy|$ is small. Recall that $c$ was previously defined in Eq. (31) and the $(a, b)$ coordinate that we have been focusing on is the offset from the $\kappa^2$-EoS minima in the $(c, d)$ coordinate.

**Lemma 16** (Approximity of $c$ to $x$). *For any large constant $K > 512$, fix any $\kappa < \frac{1}{2000\sqrt{2}}K^{-1}$. For any $x \in (\frac{\sqrt{2}}{2}\kappa^{-1}, 2\kappa^{-1})$ and any $y$ such that $|1 - xy| < K^{-1}$, we have $c(x, y) \triangleq \sqrt{x^2 - y^2} \in (x - 32\kappa^3, x)$.*

*Proof of Lemma 16.* Since $xy < 1 + K^{-1}$ and $x > \frac{\sqrt{2}}{2}\kappa^{-1}$, we must have $y < 2(1 + K^{-1})\kappa < 2\sqrt{2}\kappa$, where the last step holds since we assumed $K > 512$. Meanwhile $1 - xy < K^{-1}$ also implies $xy > 1 - K^{-1} > 0$, so $y > 0$.

Thus we have

$$\begin{aligned}
c(x, y) &= \sqrt{x^2 - y^2} \\
&> \sqrt{x^2 - (2\sqrt{2}\kappa)^2} = x\sqrt{1 - 8\kappa^2 x^{-2}} > x\sqrt{1 - 8\kappa^2(2\kappa^2)} > x(1 - 16\kappa^4).
\end{aligned} \tag{109}$$

where the last two inequality holds since $x > \frac{\sqrt{2}}{2}\kappa^{-1}$ and $\sqrt{x} > x$ when $x \in (0, 1)$. Thus $c(x, y) > x - 16x\kappa^4 > x - 32\kappa^3$ since we assume $x < 2\kappa^{-1}$. Since $y > 0$, $\sqrt{x^2 - y^2} < x$, so $c(x, y) \in (x - 32\kappa^3, x)$. □

Now we can proceed to proving Corollary 3.1. We first restate the result here:

**Corollary 3.1** (Sharpness Concentration under $(x, y)$-Parameterization). *For a large enough absolute constant $K$, suppose $\eta < \frac{1}{8000000}K^{-2}$, and the initialization $(x_0, y_0)$ satisfies $x_0 \in (\breve{x} + 13\eta^{\frac{5}{4}}, \breve{x} + \frac{1}{5}K^{-2}\eta^{-\frac{1}{2}})$ and $|x_0 y_0 - 1| \in (0, K^{-1})$ where $(\breve{x}, \breve{y})$ is the $\eta$-EoS minima defined in Definition 1. The GD trajectory characterized in Eq. (2) with fixed step size $\eta$ from $(x_0, y_0)$ will converge to a global minimum with sharpness $\lambda \in (\frac{2}{\eta} - \frac{20}{3}\eta, \frac{2}{\eta})$.*

*Proof of Corollary 3.1.* Following the convergence proof for Theorem 3.1, to prove this corollary we only need to show that all initializations $x_0, y_0$ satisfies the initialization condition of Theorem 3.1 after re-parameterized to $(a, b)$, and the sharpness $\lambda$ of the minima will satisfy $\lambda \in (\frac{2}{\eta} - \frac{20}{3}\eta, \frac{2}{\eta})$.

We first check the initialization in the $(x, y)$ coordinate satisfies the initialization condition in Theorem 3.1. Due to the different contexts, we will use $\kappa = \sqrt{\eta}$ and $\eta$ itself interchangeably.

Recall from Eq. (27) that $\breve{x} = \frac{\sqrt{2}}{2}((-4 + \eta^{-2})^{\frac{1}{2}} + \eta^{-1})^{\frac{1}{2}}$. It is not hard to check that $\breve{x} > \frac{\sqrt{2}}{2}\kappa^{-1}$ and $\breve{x} < \kappa^{-1}$ where $\kappa = \sqrt{\eta}$. Since $|x_0 y_0 - 1| < K^{-1}$ as required by the initialization, from Lemma 16 we have $c_0 \triangleq c(x_0, y_0) \in (x_0 - 32\kappa^3, x_0)$. By the same reasoning we also have $\breve{c} \triangleq c(\breve{x}, \breve{y}) \in (\breve{x} - 32\kappa^3, \breve{x})$. Thus $x_0 > \breve{x} + 13\kappa^{\frac{5}{2}}$ implies $c_0 > \breve{c} + 13\kappa^{\frac{5}{2}} - 32\kappa^3$. Since we assume $\kappa < \frac{1}{2000\sqrt{2}}K^{-1}$ where $K > 512$, $32\kappa^3 < \kappa^{\frac{5}{2}}$, and hence $c_0 > \breve{c} + 12\kappa^{\frac{5}{2}}$. Therefore $a_0 \triangleq c_0 - \breve{c} > 12\kappa^{\frac{5}{2}}$.

On the other end, since $\breve{x} < \kappa^{-1}$ as shown above, we have $\breve{x} + \frac{1}{4}K^{-2}\kappa^{-1} < 2\kappa^{-1}$. So we can again apply Lemma 16 so that $c_0 < c(\breve{x} + \frac{1}{4}K^{-2}\eta^{-\frac{1}{2}}, y_0) < \breve{x} + \frac{1}{4}K^{-2}\eta^{-\frac{1}{2}}$. Since $\breve{c} > \breve{x} - 32\kappa^3$, we have $c_0 < \breve{c} + \frac{1}{5}K^{-2}\eta^{-\frac{1}{2}} + 32\kappa^3 < \breve{c} + \frac{1}{4}K^{-2}\eta^{-\frac{1}{2}}$. The last step holds since $32\kappa^3 < K^{-2}\eta^{-\frac{1}{2}}$. Therefore $a_0 \triangleq c_0 - \breve{c} < \frac{1}{20}\frac{1}{4}K^{-2}\eta^{-\frac{1}{2}}$. Since $|b_0| = |1 - xy| \in (0, K^{-1})$ by construction, we know $(a_0, b_0)$ satisfies the initialization condition of Theorem 3.1, and we can have the trajectory converging to a global minima with $a \in [-\frac{5}{3}, -\frac{1}{10}]$ and $b = 0$.

Now we show that for global minima with satisfies $a_k \in [-\frac{5}{3}, -\frac{1}{10}]$ and $b_k = 0$, the sharpness $\lambda$ satisfies $\lambda \in (\frac{2}{\eta} - \frac{20}{3}\eta, \frac{2}{\eta})$.

Recall from Eq. (3) that the sharpness of $(x, y)$ near the global minima is given by

$$\lambda = \frac{1}{2}\left((x^2 + y^2)(3\gamma^2 - 1) + \sqrt{(x^2 + y^2)^2(1 - 3\gamma^2)^2 + 4\gamma^2(3 - 10\gamma^2 + 7\gamma^4)}\right) \quad (110)$$

where $\gamma \triangleq xy$. When $(x, y)$ is a global minima, $\gamma = 1$, and Eq. (110) reduces to

$$
\begin{aligned}
\lambda &= \frac{1}{2}\left((x^2 + y^2)(3 - 1) + \sqrt{(x^2 + y^2)^2(1 - 3)^2}\right) \\
&= \frac{1}{2}(2(x^2 + y^2) + 2(x^2 + y^2)) \\
&= 2(x^2 + y^2) \\
&= 2\sqrt{(x^2 + x^{-2})^2} \\
&= 2\sqrt{4 + (x^2 - x^{-2})^2} \\
&= 2\sqrt{4 + (\sqrt{x^2 - y^2})^4} \\
&= 2\sqrt{4 + c^4}.
\end{aligned} \quad (111)
$$

Since $c = (\kappa^{-4} - 4)^{\frac{1}{4}} + a$, where $a \in [-\frac{5}{3}\kappa^3, -\frac{1}{10}\kappa^3]$, we have $c < (\kappa^{-4} - 4)^{\frac{1}{4}}$ and hence $\lambda = 2\sqrt{4 + c^4} < 2\sqrt{4 + \kappa^{-4} - 4} = 2\kappa^{-2} = \frac{2}{\eta}$.

Now we prove the lower bound for $\lambda$. Follow from $c > (\kappa^{-4} - 4)^{\frac{1}{4}} - \frac{5}{3}\kappa^3$, we have

$$
\begin{aligned}
c^4 &> \left((\kappa^{-4} - 4)^{\frac{1}{4}} - \frac{5}{3}\kappa^3\right)^4 \\
&= \left((\kappa^{-4} - 4)^{-\frac{1}{4}}(\kappa^{-4} - 4)^{\frac{1}{4}} - \frac{5}{3}\kappa^3(\kappa^{-4} - 4)^{-\frac{1}{4}}\right)^4 (\kappa^{-4} - 4) \\
&= \left(1 - \frac{5}{3}\kappa^3(\kappa^{-4} - 4)^{-\frac{1}{4}}\right)^4 (\kappa^{-4} - 4) \\
&\geq \left(1 - \frac{20}{3}\kappa^3(\kappa^{-4} - 4)^{-\frac{1}{4}}\right)(\kappa^{-4} - 4) \quad \text{(Bernoulli inequality)} \\
&= \kappa^{-4} - 4 - \frac{20}{3}\kappa^3(\kappa^{-4} - 4)^{\frac{3}{4}} \\
&\geq \kappa^{-4} - 4 - \frac{20}{3}\kappa^3(\kappa^{-4})^{\frac{3}{4}} \\
&= \kappa^{-4} - 4 - \frac{20}{3}.
\end{aligned} \quad (112)
$$

Thus

$$
\begin{aligned}
\lambda &= 2\sqrt{4 + c^4} \\
&\geq 2\sqrt{4 + \kappa^{-4} - 4 - \frac{20}{3}} \\
&= 2\sqrt{\kappa^{-4} - \frac{20}{3}} \\
&= 2\sqrt{1 - \frac{20}{3}\kappa^4}\kappa^{-2} \\
&\geq 2(1 - \frac{20}{3}\kappa^4)\kappa^{-2} \\
&= 2\kappa^{-2} - \frac{20}{3}\kappa^2 = \frac{2}{\eta} - \frac{20}{3}\eta.
\end{aligned} \quad (113)
$$

Hence in conclusion, the converging minima has sharpness $\lambda \in (\frac{2}{\eta} - \frac{20}{3}\eta, \frac{2}{\eta})$, which completes the proof for the corollary. $\qquad\square$

### B.6.2 PROOF OF COROLLARY 3.2

Before proving Corollary 3.2, we first show a simple lemma on the approximity of $\breve{x}$ and $\kappa^{-1}$ where $\breve{x}$ is the $x$-coordinate for the $\kappa^2$-EoS minima.

**Lemma 17** (Approximity of $\breve{x}$ and $\kappa^{-1}$). *For any large constant $K > 512$, fix any $\kappa < \frac{1}{2000\sqrt{2}}K^{-1}$. With $\breve{x} \triangleq \frac{1}{\sqrt{2}}((-4 + \eta^{-2})^{\frac{1}{2}} + \eta^{-1})^{\frac{1}{2}}$, we have $|\breve{x} - \kappa^{-1}| < 36\kappa^3$.*

*Proof of Lemma 17.* Since the condition for $\kappa$ is identical to that of Lemma 16, we have $|\breve{x} - \breve{c}| < 32\kappa^3$ from Lemma 16 where $\breve{c} = (\kappa^{-4} - 4)^{\frac{1}{4}}$ from the calculation in Appendix B.1. Note that

$$\breve{c} = (\kappa^{-4} - 4)^{\frac{1}{4}} = \kappa^{-1}(1 - 4\kappa^{-4})^{\frac{1}{4}} > \kappa^{-1}(1 - 4\kappa^{-4}) = \kappa^{-1} - 4\kappa^3. \tag{114}$$

It is straightforward that $\breve{c} < \kappa^{-1}$, so $|\breve{c} - \kappa^{-1}| < 4\kappa^3$. Combining with $|\breve{x} - \breve{c}| < 32\kappa^3$, we have $|\breve{x} - \kappa^{-1}| < 36\kappa^3$, which completes the proof. $\square$

Now we can proceed to prove Corollary 3.2. We first restate the corollary.

**Corollary 3.2** (Sharpness Adaptivity). *For a large enough constant $K$, fix any $\alpha < \frac{1}{2000\sqrt{2}}K^{-1}$. For all initialization $(x_0, y_0)$ in the region characterized by*

$$x_0 \in (\alpha^{-1} + \tfrac{1}{15}K^{-2}\alpha^{-1}, \alpha^{-1} + \tfrac{1}{6}K^{-2}\alpha^{-1}) \tag{10}$$

*and $|x_0 y_0 - 1| \in (0, K^{-1})$, the GD trajectory from $(x_0, y_0)$ characterized by Eq. (2) with any step size $\eta \in (\alpha^2 - \frac{1}{10}K^{-2}\alpha^2, \alpha^2)$ will converge to a minima with sharpness $\lambda \in (\frac{2}{\eta} - \frac{20}{3}\eta, \frac{2}{\eta})$.*

*Proof of Corollary 3.2.* To prove this corollary, we only need to show that for all $\eta$ in the required range, the initialization region characterized by the corollary is a subset of the initialization region required by Corollary 3.1 for that particular $\eta$.

For the ease of derivation, we will use $\kappa^2$ to substitute for $\eta$. Since we are dealing with different step sizes, we augment our notation to let $(\breve{x}_{\kappa^2}, \breve{y}_{\kappa^2})$ denote the $\kappa^2$-EoS minimum. Note that the condition for $y_0$, namely $|x_0 y_0 - 1| \in (0, K^{-1})$ is identical to what is required by Corollary 3.1 so we only need to show for any learning rate $\kappa^2 \in (\alpha^2 - \frac{1}{10}K^{-2}\alpha^2, \alpha^2)$,

$$(\alpha^{-1} + \tfrac{1}{15}K^{-2}\alpha^{-1}, \alpha^{-1} + \tfrac{1}{6}K^{-2}\alpha^{-1}) \subseteq (\breve{x}_{\kappa^2} + 13\kappa^{\frac{5}{2}}, \breve{x}_{\kappa^2} + \tfrac{1}{5}K^{-2}\kappa^{-1}). \tag{115}$$

We will first show $\breve{x}_\eta + 13\kappa^{\frac{5}{2}} < \alpha^{-1} + \frac{1}{15}K^{-2}\alpha^{-1}$.

Since $\kappa^2 > \alpha^2 - \frac{1}{10}K^{-2}\alpha^2$, we have $\kappa > (1 - \frac{1}{10}K^{-2})^{\frac{1}{2}}\alpha > (1 - \frac{1}{9}K^{-2} + \frac{1}{324}K^{-4})^{\frac{1}{2}}\alpha = (1 - \frac{1}{16}K^{-2})\alpha$ where the last inequality holds since we may assume $K > 512$. Taking the multiplicative inverse, we have $\kappa^{-1} < (1 - \frac{1}{18}K^{-2})^{-1}\alpha^{-1}$.

Now note that since $K^{-1} < 1$, $(1 + \frac{1}{16}K^{-2})(1 - \frac{1}{18}K^{-2}) = 1 + \frac{1}{144}K^{-2} - \frac{1}{288}K^{-4} > 1$, so $(1 - \frac{1}{18}K^{-2})^{-1} < 1 + \frac{1}{16}K^{-2}$ and hence $\kappa^{-1} < (1 - \frac{1}{18}K^{-2})^{-1}\alpha^{-1} < (1 + \frac{1}{16}K^{-2})\alpha^{-1}$. From Lemma 17 we know $\breve{x}_{\kappa^2} < \kappa^{-1} + 36\kappa^3$, so

$$\breve{x}_{\kappa^2} + 13\kappa^{\frac{5}{2}} < \kappa^{-1} + 36\kappa^3 + 13\kappa^{\frac{5}{2}} < (1 + \tfrac{1}{16}K^{-2})\alpha^{-1} + 36\kappa^3 + 13\kappa^{\frac{5}{2}}. \tag{116}$$

Note that since $\kappa < \alpha < \frac{1}{2000\sqrt{2}}K^{-1}$ where $K > 512$, we have

$$36\kappa^3 + 13\kappa^{\frac{5}{2}} < \kappa^2 < \tfrac{1}{8000000}K^{-2} < \tfrac{1}{480}K^{-2} < (\tfrac{1}{15} - \tfrac{1}{16})K^{-2}\alpha^{-1}. \tag{117}$$

Thus combining with $\breve{x}_{\kappa^2} + 13\kappa^{\frac{5}{2}} < (1 + \frac{1}{16}K^{-2})\alpha^{-1} + 36\kappa^3 + 13\kappa^{\frac{5}{2}}$, we have

$$\breve{x}_{\kappa^2} + 13\kappa^{\frac{5}{2}} < (1 + \tfrac{1}{16}K^{-2})\alpha^{-1} + (\tfrac{1}{15} - \tfrac{1}{16})K^{-2}\alpha^{-1} < \alpha^{-1} + \tfrac{1}{15}K^{-2}\alpha^{-1}. \tag{118}$$

The other side is much simpler to show. Since $\kappa < \alpha$, we have $\kappa^{-1} > \alpha^{-1}$. From Lemma 17 we have $\breve{x}_{\kappa^2} > \kappa^{-1} - 36\kappa^3$, so $\breve{x}_{\kappa^2} + \frac{1}{5}K^{-2}\kappa^{-1} > \alpha^{-1} - 36\kappa^3 + \frac{1}{5}K^{-2}\alpha^{-1}$. From Eq. (117) we know $36\kappa^3 < \frac{1}{480}K^{-2} < \frac{1}{30}K^{-2}\alpha^{-1} = (\frac{1}{5} - \frac{1}{6})K^{-2}\alpha^{-1}$. Therefore $\breve{x}_{\kappa^2} + \frac{1}{5}K^{-2}\kappa^{-1} > \alpha^{-1} + \frac{1}{6}K^{-2}\alpha^{-1}$. This concludes the proof for the corollary. $\square$

### B.7 OTHER AUXILIARY LEMMAS

#### B.7.1 CONSTANT BOUND ON $R_\delta(\kappa)$

**Lemma 18.** *For all $\kappa < 0.1$, for all $(a,b) \in (-\kappa^{-1}, \kappa^{-1}) \times (-1, 1)$, there exists some absolute constant $K$ independent of $a, b, \kappa$ such that*

$$\delta \triangleq \sqrt{4\kappa^4(1+b)^2 + \left(a\kappa + (1 - 4\kappa^4)^{\frac{1}{4}}\right)^4} = 1 + 2a\kappa + a^2\kappa^2 + 2b(2+b)\kappa^4 + K\kappa^5. \quad (119)$$

*Proof of Lemma 18.* By explicitly computing the derivatives for $\delta$ with respect to $\kappa$, we know that $\delta$ is $C^6$ with respect to $\kappa$ and have the Taylor expansion

$$\delta = 1 + 2a\kappa + a^2\kappa^2 + 2b(2+b)\kappa^4 + R_\delta(\kappa)\kappa^5. \quad (120)$$

where $R_\delta(\kappa)$ is the Lagrangian remainder that $R_\delta(\kappa) = \frac{\partial^5 \delta}{\partial \kappa^5}(x)/120$ for some $x \in [0, \kappa]$. To prove the lemma we only need to bound the $R_\delta$ by some absolute constants. For simplicity of notation, denote $\alpha \triangleq (1 - 4x^4)^{\frac{1}{4}}$, $\beta = \alpha + ax$, $\gamma = 1 + b$, and $\phi = \left(4\gamma^2 x^4 + \beta^4\right)^{\frac{1}{2}}$. Moreover, let

$$
\begin{aligned}
\rho_1 &= a - \frac{4\kappa^3}{\alpha^3}, \\
\rho_2 &= -\frac{12\kappa^2}{\alpha^3} - \frac{48\kappa^6}{\alpha^7}, \\
\rho_3 &= -\frac{24\kappa}{\alpha^3} - \frac{432\kappa^5}{\alpha^7} - \frac{1344\kappa^9}{\alpha^{11}}, \\
\rho_4 &= -\frac{24}{\alpha^3} - \frac{2448\kappa^4}{\alpha^7} - \frac{24192\kappa^8}{\alpha^{11}} - \frac{59136\kappa^{12}}{\alpha^{15}}, \\
\rho_5 &= -\frac{10080\kappa^3}{\alpha^7} - \frac{262080\kappa^7}{\alpha^{11}} - \frac{1774080\kappa^{11}}{\alpha^{15}} - \frac{3548160\kappa^{15}}{\alpha^{19}}.
\end{aligned}
\quad (121)
$$

By doing some tedious calculation we have

$$
\begin{aligned}
\frac{\partial^5 \delta}{\partial \kappa^5}(x) = {} & \frac{105\left(16\gamma^2\kappa^3 + 4\beta^3\rho_1\right)^5}{32\phi^9} - \frac{75\left(16\gamma^2\kappa^3 + 4\beta^3\rho_1\right)^3\left(48\gamma^2\kappa^2 + 12\beta^2\rho_1^2 + 4\beta^3\rho_2\right)}{8\phi^7} \\
& + \frac{45\left(16\gamma^2\kappa^3 + 4\beta^3\rho_1\right)\left(48\gamma^2\kappa^2 + 12\beta^2\rho_1^2 + 4\beta^3\rho_2\right)^2}{8\phi^5} \\
& + \frac{15\left(16\gamma^2\kappa^3 + 4\beta^3\rho_1\right)^2\left(96\gamma^2\kappa + 24\beta\rho_1^3 + 36\beta^2\rho_1\rho_2 + 4\beta^3\rho_3\right)}{4\phi^5} \\
& - \frac{5\left(48\gamma^2\kappa^2 + 12\beta^2\rho_1^2 + 4\beta^3\rho_2\right)\left(96\gamma^2\kappa + 24\beta\rho_1^3 + 36\beta^2\rho_1\rho_2 + 4\beta^3\rho_3\right)}{2\phi^3} \\
& - \frac{5\left(16\gamma^2\kappa^3 + 4\beta^3\rho_1\right)\left(96\gamma^2 + 24\rho_1^4 + 144\beta\rho_1^2\rho_2 + 36\beta^2\rho_2^2 + 48\beta^2\rho_1\rho_3 + 4\beta^3\rho_4\right)}{4\phi^3} \\
& + \frac{240\rho_1^3\rho_2 + 360\beta\rho_1\rho_2^2 + 240\beta\rho_1^2\rho_3 + 120\beta^2\rho_2\rho_3 + 60\beta^2\rho_1\rho_4 + 4\beta^3\rho_5}{2\phi}
\end{aligned}
$$
$$(122)$$

Since $\kappa < 0.1$, we have $\alpha = (1 - 4x^4)^{\frac{1}{4}} > 0.99$ and $\phi = \left(4\gamma^2 x^4 + \beta^4\right)^{\frac{1}{2}} \geq \beta^2 = (\alpha + dx)^2 \geq \alpha^2 > 0.9$. Thus $\alpha$ and $\phi$ are bounded from below by some constants. Since $\kappa$ is bounded from above, we have $\rho_1, \ldots, \rho_5$ bounded from above by some constants.

Now we give upper bounds for $\beta$ and $\gamma$. Since $|x| < 0.1$, we have $(1 - 4x^4)^{\frac{1}{4}} \leq 1$. Since $|a| < \kappa^{-1}$, we have $\beta = \alpha + ax < \alpha + a\kappa < \alpha + 1 < 2$. Since $|b| < 1$, we have $\gamma = 1 + b < 2$. Note that it is straightforward from construction that all these terms are positive.

From Eq. (122) we have $\frac{\partial^6 \delta}{\partial \kappa^6}(x)$ as some finite degree polynomial of $\beta, \gamma, \kappa, \rho_1, \ldots, \rho_6$, and $\phi^{-1}$. Since the absolute value for all of these terms are bounded above by some constants independent of $x$, we have $\left| \frac{\partial^6 \delta}{\partial \kappa^6}(x) \right|$ uniformly bounded above by some constant $K$ for all $x \in [0, \kappa]$. This completes the proof of the lemma. $\qquad \square$

## C  THEORETICAL ANALYSIS ON RANK-1 APPROXIMATION OF ISOTROPIC MATRIX

In this section we prove the convergence of the vector case.

### C.1  PRELIMINARIES

**Model:** We consider a generalized model from the scalar product case.

$$\min_{\boldsymbol{x},\boldsymbol{y}\in\mathbb{R}^d} \frac{1}{4}\|\boldsymbol{I} - \boldsymbol{x}\boldsymbol{y}^\top\boldsymbol{x}\boldsymbol{y}^\top\|_F^2 \tag{123}$$

The normalization factor $\frac{1}{4}$ is added to show the equivalence of this rank-1 isotropic matrix factorization problem and the scalar case considered in Appendix B.6. We will show how the equivalence is achieved due to the alignment in the next section.

**Update Rule:** We use gradient descent to optimize this problem. By computing the closed-form of the gradient, we know the 1-step update of $\boldsymbol{x}$ and $\boldsymbol{y}$ follows:

$$\begin{aligned}
\boldsymbol{x}_{t+1} &= \boldsymbol{x}_t + \eta\left((\boldsymbol{x}_t^\top\boldsymbol{y}_t)\boldsymbol{y}_t - \frac{1}{2}(\boldsymbol{x}_t^\top\boldsymbol{y}_t)\|\boldsymbol{x}_t\|^2\|\boldsymbol{y}_t\|^2\boldsymbol{y}_t - \frac{1}{2}(\boldsymbol{x}_t^\top\boldsymbol{y}_t)^2\|\boldsymbol{y}_t\|^2\boldsymbol{x}_t\right), \\
\boldsymbol{y}_{t+1} &= \boldsymbol{y}_t + \eta\left((\boldsymbol{x}_t^\top\boldsymbol{y}_t)\boldsymbol{x}_t - \frac{1}{2}(\boldsymbol{x}_t^\top\boldsymbol{y}_t)\|\boldsymbol{x}_t\|^2\|\boldsymbol{y}_t\|^2\boldsymbol{x}_t - \frac{1}{2}(\boldsymbol{x}_t^\top\boldsymbol{y}_t)^2\|\boldsymbol{x}_t\|^2\boldsymbol{y}_t\right).
\end{aligned} \tag{124}$$

With the gradient descent update above, we can have the dynamics of the following four quantities: $\|\boldsymbol{x}\|^2, \|\boldsymbol{y}\|^2$, and $(\boldsymbol{x}^\top\boldsymbol{y})^2$.

$$\begin{aligned}
\|\boldsymbol{x}_{t+1}\|^2 &= \|\boldsymbol{x}_t\|^2 + 2\eta\left((\boldsymbol{x}_t^\top\boldsymbol{y}_t)^2 - (\boldsymbol{x}_t^\top\boldsymbol{y}_t)^2\|\boldsymbol{x}_t\|\|\boldsymbol{y}_t\|\right) \\
&\quad + \frac{\eta^2}{4}\left(4(\boldsymbol{x}_t^\top\boldsymbol{y}_t)^2\|\boldsymbol{y}_t\|^2 + (\boldsymbol{x}_t^\top\boldsymbol{y}_t)^2\|\boldsymbol{x}_t\|^4\|\boldsymbol{y}_t\|^6 + 3(\boldsymbol{x}_t^\top\boldsymbol{y}_t)^4\|\boldsymbol{x}_t\|^2\|\boldsymbol{y}_t\|^4\right. \\
&\quad \left. -4(\boldsymbol{x}_t^\top\boldsymbol{y}_t)^2\|\boldsymbol{x}_t\|^2\|\boldsymbol{y}_t\|^4 - 4(\boldsymbol{x}_t^\top\boldsymbol{y}_t)^4\|\boldsymbol{y}_t\|^2\right), \\
\|\boldsymbol{y}_{t+1}\|^2 &= \|\boldsymbol{y}_t\|^2 + 2\eta\left((\boldsymbol{x}_t^\top\boldsymbol{y}_t)^2 - (\boldsymbol{x}_t^\top\boldsymbol{y}_t)^2\|\boldsymbol{x}_t\|\|\boldsymbol{y}_t\|\right) \\
&\quad + \frac{\eta^2}{4}\left(4(\boldsymbol{x}_t^\top\boldsymbol{y}_t)^2\|\boldsymbol{x}_t\|^2 + (\boldsymbol{x}_t^\top\boldsymbol{y}_t)^2\|\boldsymbol{x}_t\|^6\|\boldsymbol{y}_t\|^4 + 3(\boldsymbol{x}_t^\top\boldsymbol{y}_t)^4\|\boldsymbol{x}_t\|^4\|\boldsymbol{y}_t\|^2\right. \\
&\quad \left. -4(\boldsymbol{x}_t^\top\boldsymbol{y}_t)^2\|\boldsymbol{x}_t\|^4\|\boldsymbol{y}_t\|^2 - 4(\boldsymbol{x}_t^\top\boldsymbol{y}_t)^4\|\boldsymbol{x}_t\|^2\right),
\end{aligned} \tag{125}$$

$$\begin{aligned}
(\boldsymbol{x}_{t+1}^\top\boldsymbol{y}_{t+1})^2 &= (\boldsymbol{x}_t^\top\boldsymbol{y}_t)^2\left(1 + \eta(\|\boldsymbol{x}_t\|^2 + \|\boldsymbol{y}_t\|^2)(1 - \frac{1}{2}\|\boldsymbol{x}_t\|^2\|\boldsymbol{y}_t\|^2 - \frac{1}{2}(\boldsymbol{x}_t^\top\boldsymbol{y}_t)^2)\right. \\
&\quad \left. + \frac{\eta^2}{4}(\boldsymbol{x}_t^\top\boldsymbol{y}_t)^2\left[4(\|\boldsymbol{x}_t\|^2\|\boldsymbol{y}_t\|^2 - 1)^2 - \|\boldsymbol{x}_t\|^2\|\boldsymbol{y}_t\|^2(\|\boldsymbol{x}_t\|^2\|\boldsymbol{y}_t\|^2 - (\boldsymbol{x}_t^\top\boldsymbol{y}_t)^2)\right]\right)^2.
\end{aligned} \tag{126}$$

Furthermore, we define an alignment notation for the variable: $\xi_t := \|\boldsymbol{x}_t\|^2\|\boldsymbol{y}_t\|^2 - (\boldsymbol{x}_t^\top\boldsymbol{y}_t)^2$. By Eq. (126) and Eq. (125), we can derive the following update rule of $\xi$.

$$\xi_{t+1} = \xi_t\left(1 - (\boldsymbol{x}_t^\top\boldsymbol{y}_t)^2\eta\left(\frac{\|\boldsymbol{x}_t\|^2}{2} + \frac{\|\boldsymbol{y}_t\|^2}{2} + \frac{\eta}{4}((2 - \|\boldsymbol{x}_t\|^2\|\boldsymbol{y}_t\|^2)^2 - \|\boldsymbol{x}_t\|^2\|\boldsymbol{y}_t\|^2(\boldsymbol{x}_t^\top\boldsymbol{y}_t)^2)\right)\right)^2 \tag{127}$$

### C.2  PROOF OF THEOREM 3.2

We restate the theorem of the vector case.

**Theorem C.1.** *For a large enough absolute constant $K$, with all the initialization $(\boldsymbol{x}_0, \boldsymbol{y}_0)$ satisfying $\boldsymbol{x}_0 \sim \delta_x Unif(\mathbb{S}^{d-1})$, $\boldsymbol{y}_0 \sim \delta_y Unif(\mathbb{S}^{d-1})$, $\delta_x\delta_y = \frac{1}{2}$, $\delta_x \in (\breve{x} + \frac{1}{80}K^{-2}\eta^{-\frac{1}{2}}, \breve{x} + \frac{1}{8}K^{-2}\eta^{-\frac{1}{2}})$, if*

*step size $\eta < \min\{\frac{K^{-4}}{8000000}, \frac{K^{-2}}{20000+2000(\log(d)-\log(\delta_0))}\}$, and a multiplicative perturbation $y'_t = y_t(1+2K^{-1})$[3] is performed at time $t = t_p$ for some $t_p > \mathcal{O}(-\log(\eta) + \log(d) - \log(\delta_0) + K^3)$, then for any $\epsilon > 0$, with probability $p > 1 - 2\delta_0 - 2\exp\{-\Omega(d)\}$ there exists $T = \mathcal{O}(K^{-2}\kappa^{-\frac{15}{2}} - \log(\epsilon) - \log(\delta_0))$ such that for all $t > T$, $\mathcal{L}(x, y) < \epsilon$ and $\|x_t\|^2 + \|y_t\|^2 \in (\frac{1}{\eta} - \frac{10}{3}\eta, \frac{1}{\eta})$.*

In the following analysis, we still use the operator $\mathcal{O}(\cdot)$ to only hides absolute constants, and $K$ to represent the absolute constant that uniformly upper bounds all the absolute constants of the $\mathcal{O}(\cdot)$ terms. Also $K > 512$. We still use the notation of $\eta$-EoS minimum $(\breve{x}, \breve{y})$ of the scalar case in the vector case.

We first prove some properties of the global minimizers. The following lemma guarantees the alignment of the two vectors $x, y$ at any global minimizer $(x, y)$.

**Lemma 19.** *(Global minimizers) For all global minimizers $(x, y)$ of the optimization problem Eq. (123), we have $x = cy$ for some $c \in \mathbb{R}$, and $\|x\|\|y\| = 1$.*

*Proof.* We directly consider the objective $\left\| I - xy^\top xy^\top \right\|_F^2$.

$$
\begin{aligned}
\left\| I - xy^\top xy^\top \right\|_F^2 &= \mathrm{Tr}((I - xy^\top xy^\top)^\top (I - xy^\top xy^\top)) \\
&= \mathrm{Tr}(I - yx^\top yx^\top - xy^\top xy^\top - (x^\top y)^2 \|x\|^2 yy^\top) \\
&= d - 2(x^\top y)^2 + (x^\top y)^2 \|x\|^2 \|y\|^2 \\
&= d - 1 + ((x^\top y)^2 - 1)^2 + (x^\top y)^2 (\|x\|^2 \|y\|^2 - (x^\top y)^2) \\
&\geq d - 1.
\end{aligned}
$$

The global minimizer takes value $\frac{d-1}{4}$ and the equality holds when

$$
((x^\top y)^2 - 1)^2 = (x^\top y)^2 (\|x\|^2 \|y\|^2 - (x^\top y)^2) = 0.
$$

which is equivalent to $x = cy$ for some $c \in \mathbb{R}$, and $\|x\|\|y\| = 1$. $\qquad \square$

Now we consider the formal proof. To prove the theorem we have four steps: (i) we prove the two vectors $x, y$ will decay geometrically after $T = \mathcal{O}(-\log(\eta) - \log(\delta_0) + \log(d))$ time (Lemma 20, Lemma 21); (ii) we prove the norm of the two vectors $\|x\|, \|y\|$ will satisfy the initialization condition in $T' < 6K^3 + 4K$ time (Lemma 22);(iii) To escape from some sharp minima, we add some deterministic perturbation, and we prove that after the perturbation gradient descent will re-enter the feasible regime, while at the same time keep a constant distance to the manifold of the minimizers; (iv) after entering the feasible region and $\xi$ is small enough, the re-parameterized dynamics of $\|x\|$ and $\|y\|$ can be captured by the scalar dynamics (Lemma 25). Then, we can reduce this vector case to the scalar product case and finish the proof.

Here we present the following key lemmas to prove the convergence at the edge of stability. For simplicity, we denote $\alpha(t) := \eta(\|x_t\|^2 + \|y_t\|^2) - 1$.

**Lemma 20.** *(Alignment) Assume $0 < \alpha(0) < \frac{1}{8}$, $0 < x_0^\top y_0 < \|x_0\|\|y_0\| \leq \frac{1}{2}$, $\eta < 0.01$. Then as long as $-\frac{1}{100} < \alpha(t) < \frac{1}{8}$ holds for $t \in [0, t_1]$ for some $t_1$, then for all $t \in [0, t_1 + 1]$, $0 < (x_t^\top y_t)^2 < \frac{13}{10}$ and $\xi_{t+1} < \xi_t$; Moreover, there exists some $t_0 < \log_2(\frac{7}{20 x_0^\top y_0})$, for all $t \in [t_0, t_1 + 1]$, $\frac{7}{20} < (x_t^\top y_t)^2 < \frac{13}{10}$ and $\xi_{t+1} < 0.7\xi_t$.*

*Proof.* We use induction. For iteration 0, the induction basis holds. Now we consider iteration $t + 1$ when assuming the conclusion holds at iteration $0, 1, 2, ..., t$.

Consider Equation (126). We denote $c_t = (x_t^\top y_t)^2$. And from the induction hypothesis, $\xi_t < \|x_0\|^2\|y_0\|^2 < 1/4$ for all time. Thus $\|x_t\|^2\|y_t\|^2 = c_t + \xi_t < \frac{31}{20}$. Then we have

$$
c_{t+1} < c_t(1 + (1 + \alpha(t))(1 - c_t) + 20\eta^2)^2 < c_t(\frac{1001}{500} + \alpha(t) - (1 + \alpha(t))c_t)^2.
$$

---

[3] $y'_t = y_t(1 + 2K^{-1})$ at time $t_p$ means at this iteration, we multiply $(1 + 2K^{-1})$ to the vector $y$.

Since $0 < c_t < 13/10$ and $-0.01 < \alpha < 1/8$, this function takes maximal value when $c_t = \frac{1001}{1500(1+\alpha(t))}$.

$$c_{t+1} < c_t(\frac{1001}{500} + \alpha(t) - (1 + \alpha(t))c_t)^2 \leq \frac{1001}{1500} \cdot \frac{(\frac{1001}{750} + \alpha(t))^2}{1 + \alpha(t)} < \frac{13}{10}.$$

The lower bound 0 of $c_t$ is straightforward since we have

$$c_{t+1} > c_t(1 + (1 + \alpha(t))(1 - c_t - \xi_t/2))^2 > 0$$

due to $c_t + \xi_t/2 < 31/20$ and $-0.01 < \alpha(t) < 1/8$.

Then we consider the tighter bound and a faster decaying rate after some $t_0$. If $c_0 > 7/20$ and the lower bound holds at $t = 0$, then $t_0 = 0 < \log_2(\frac{7}{20x_0^\top y_0})$. Then we begin the induction from $t_0$. Consider $t \geq t_0$,

$$\xi_t < \xi_0(-1 + (x_0^\top y_0)^2 \eta(\|x_0\|^2 + \|y_0\|^2)/2)^2 < \frac{1}{4}$$

and the lower bound of $c_{t+1}$ becomes (since $7/20 < c_t < 1.3$)

$$c_{t+1} > \min\{c_t(1 + (1 + \alpha(t))(1 - c_t - \xi_t/2))^2, c_t(1 + (1 - c_t - \xi_t/2))^2\}$$
$$> \min\{c_t(1 + \frac{9}{8}(1 - c_t - 1/8))^2, c_t(1 + (1 - c_t - 1/8))^2\}$$
$$> 7/20.$$

The last inequality is because of the monotonic decrement of the last function, which takes minimal value at $c_t = \frac{13}{10}$. This proves the first statement.

Then consider the alignment dynamics by Equation (127). The factor can be bounded as follow:

$$\left(-1 + (x^\top y)^2 \eta \left(\|x\|^2/2 + \|y\|^2/2 + \eta((2 - \|x\|^2\|y\|^2)^2 - \|x\|^2\|y\|^2(x^\top y)^2)/4)\right)\right)^2$$
$$< \max\{1 - \frac{7}{20} \times \frac{1}{2} + 100\eta^2, -1 + \frac{3}{2} \times \frac{9}{8} + 10\eta^2\}^2 < 0.7$$

By induction, we finish the proof.

If $c_0 \leq 7/20$, then we prove that $c_t$ will eventually become larger than $7/20$ after some time $t_0$, and then apply the same induction process above to finish the proof.

Still we consider the lower bound of the dynamics of $c_t$. If we have $c_t < 7/20$,

$$c_{t+1} > c_t(1 + (1 - c_t - \xi_t/2))^2 > c_t(1 + (1 - c_t - 1/8))^2 > 2c_t$$

Then it at most takes $t_0 = \lceil \log_2(\frac{7}{20x_0^\top y_0}) \rceil$ to satisfy the condition. Now we finish the proof. □

We first consider the time $\xi$ takes to become smaller than $\eta^2$.

**Lemma 21.** *(Alignment convergence time) Suppose all the conditions in Theorem C.1 holds. Then with probability $1 - 2\delta_0 - 2\exp\{-\Omega(d)\}$, there exists some time $T_0 = \log_{0.7}(\eta^2) + \log_2(\frac{21d}{20\delta_0^2}) = \mathcal{O}(-\log(\eta) + \log(d) - \log(\delta_0))$ such that $\xi_t < \eta^2$ and $\|x_t\| \in (\check{x} + \frac{1}{200}K^{-2}\eta^{-\frac{1}{2}}, \check{x} + \frac{1}{6}K^{-2}\eta^{-\frac{1}{2}})$ for $t \in [T_0, T_0 + 6K^3 + 4K]$.*

*Proof.* We first prove that the bound of $(x^\top y)^2$ can be reached with probability $1 - 2\delta_0 - 2\exp\{-\Omega(d)\}$, and meanwhile the alignment $\xi$ begins to shrink after $t_0 = \lceil \log_2(\frac{21d}{20\delta_0^2}) \rceil$.

For the initialization, $\delta_x\delta_y = \frac{1}{2}$, and by symmetry we know

$$\Pr[(x_0^\top y_0)^2 < \frac{\delta_0^2}{3d}] = \Pr[x_i^2 < \frac{\delta_0^2}{1.5d}]$$

It is equivalent to consider sampling from $\mathcal{N}(0, \boldsymbol{I})$ and then divide it by its norm. By the initialization condition, we apply the Gaussian concentration bound and Theorem 3.1.1 in Vershynin (2018) and get

$$\Pr[(\boldsymbol{x}_0^\top \boldsymbol{y}_0)^2 < \frac{\delta_0^2}{3d}] < \Pr[\boldsymbol{x}_1^2 < \delta_0^2] + \Pr[\|\boldsymbol{x}\|^2 < 1.5d] < 2\delta_0 + 2\exp\{-\Omega(d)\}$$

Then with probability $p > 1 - 2\delta_0 - 2\exp\{-\Omega(d)\}$, $\frac{\delta_0^2}{d} \leq (\boldsymbol{x}_0^\top \boldsymbol{y}_0)^2 < \frac{1}{2}$. During $t \in [0, T + 6K^3 + 4K]$, we can apply the first argument in Lemma 20 and induction to prove that $(\boldsymbol{x}_t^\top \boldsymbol{y}_t)^2 < \frac{13}{10}$ and $0 < \alpha(t) < \frac{13K^{-2}}{50}$.

For $t = 0$, since $\delta_x \in (\breve{x} + \frac{1}{80}K^{-2}\eta^{-\frac{1}{2}}, \breve{x} + \frac{1}{8}K^{-2}\eta^{-\frac{1}{2}})$, $\frac{1}{40}K^{-2} < \alpha(0) < \frac{1}{4}K^{-2} < \frac{1}{8}$ and $(\boldsymbol{x}_0^\top \boldsymbol{y}_0)^2 < \|\boldsymbol{x}_0\|^2\|\boldsymbol{y}_0\|^2 = \frac{1}{4} < \frac{13}{10}$. Then we suppose for time $t \in [0, t_1 - 1]$ the statement is correct. By the induction hypothesis, we know for $t \in [0, t_1 - 1]$ the condition of Lemma 20 holds. Therefore, by Lemma 20, for all $t \in [0, t_1]$, $(\boldsymbol{x}_t^\top \boldsymbol{y}_t)^2 < \frac{13}{10}$.

With the upper bound of $(\boldsymbol{x}_t^\top \boldsymbol{y}_t)^2$, the one step movement of $\|\boldsymbol{x}\|^2 + \|\boldsymbol{y}\|^2$ can be bounded:

$$\begin{aligned}
&\|\boldsymbol{x}_{t+1}\|^2 + \|\boldsymbol{y}_{t+1}\|^2 - \|\boldsymbol{x}_t\|^2 - \|\boldsymbol{y}_t\|^2 \\
&= (\|\boldsymbol{x}_t\|^2 + \|\boldsymbol{y}_t\|^2)(\eta^2(\boldsymbol{x}_t^\top \boldsymbol{y}_t)^2((\|\boldsymbol{x}_t\|^2\|\boldsymbol{y}_t\|^2 - 1)^2 \\
&\quad + (2 - \frac{3}{4}\|\boldsymbol{x}_t\|^2\|\boldsymbol{y}_t\|^2)(\|\boldsymbol{x}_t\|^2\|\|\boldsymbol{y}_t\|^2 - (\boldsymbol{x}_t^\top \boldsymbol{y}_t)^2))) + 4\eta(\boldsymbol{x}_t^\top \boldsymbol{y}_t)^2(1 - \|\boldsymbol{x}_t\|^2\|\boldsymbol{y}_t\|^2) \\
&< \eta \cdot (1 + \alpha(t)) \cdot \frac{13}{10} \cdot (1 + 1) + 2\eta < 5\eta.
\end{aligned}$$

For $t < T_0 + 6K^3 + 4K$, the total movement of $\|\boldsymbol{x}_t\|^2 + \|\boldsymbol{y}_t\|^2$ is smaller than $5\eta(T + 6K^3 + 4K)$, and the corresponding movement of $\alpha(t)$ is smaller than $5\eta^2(T + 6K^3 + 4K) < \frac{K^{-2}}{200}$ (since $\eta < \min\{\frac{K^{-4}}{8000000}, \frac{K^{-2}}{20000 + 2000(\log(d) - \log(\delta_0))}\}$). Thus by induction, $(\boldsymbol{x}_t^\top \boldsymbol{y}_t)^2 < \frac{13}{10}$ and $0 < \alpha(t) < \frac{K^{-2}}{100} + \frac{K^{-2}}{4} < \frac{13K^{-2}}{50}$. Then by the second argument of Lemma 20, we have $\frac{7}{20} < (\boldsymbol{x}_t^\top \boldsymbol{y}_t)^2 < \frac{13}{10}$ and $\xi_{t+1} < 0.7\xi_t$ for $t > \log_2(\frac{21d}{20\delta_0^2}) = \mathcal{O}(\log(d) - \log(\delta_0))$.

After $t = \lceil\log_2(\frac{21d}{20\delta_0^2})\rceil$, we can calculate the time when $\xi_t < \eta^2$. It needs at most $\log_{0.7}(\eta^2) = -\mathcal{O}(\log(\eta))$ for $\xi_t$ to become smaller than $\eta^2$. Now we know with $T_0 = \log_{0.7}(\eta^2) + \log_2(\frac{21d}{20\delta_0^2}) = \mathcal{O}(-\log(\eta) + \log(d) - \log(\delta_0))$, within $t \in [T_0, T_0 + 6K^3 + 4K]$, $\xi_t < \eta^2$ always holds.

On the other hand, we can also have the lower bound of the $\|\boldsymbol{x}_t\|^2 + \|\boldsymbol{y}_t\|^2$. The one step movement has the lower bound

$$\begin{aligned}
&\|\boldsymbol{x}_{t+1}\|^2 + \|\boldsymbol{y}_{t+1}\|^2 - \|\boldsymbol{x}_t\|^2 - \|\boldsymbol{y}_t\|^2 \\
&= (\|\boldsymbol{x}_t\|^2 + \|\boldsymbol{y}_t\|^2)(\eta^2(\boldsymbol{x}_t^\top \boldsymbol{y}_t)^2((\|\boldsymbol{x}_t\|^2\|\boldsymbol{y}_t\|^2 - 1)^2 \\
&\quad + (2 - \frac{3}{4}\|\boldsymbol{x}_t\|^2\|\boldsymbol{y}_t\|^2)(\|\boldsymbol{x}_t\|^2\|\|\boldsymbol{y}_t\|^2 - (\boldsymbol{x}_t^\top \boldsymbol{y}_t)^2))) + 4\eta(\boldsymbol{x}_t^\top \boldsymbol{y}_t)^2(1 - \|\boldsymbol{x}_t\|^2\|\boldsymbol{y}_t\|^2) \\
&> 0 - 2\eta > -5\eta.
\end{aligned}$$

So for $t < T_0 + 6K^3 + 4K$, the total decrement of $\|\boldsymbol{x}_t\|^2 + \|\boldsymbol{y}_t\|^2$ is larger than $-5\eta(T_0 + 6K^3 + 4K) > -\frac{K^{-2}}{200}\eta^{-\frac{1}{2}}$. Because $\boldsymbol{x}_t^\top \boldsymbol{y}_t \in (\frac{7}{20}, \frac{13}{10})$, $\|\boldsymbol{y}_t\|^2 < \frac{13}{10}/\|\boldsymbol{x}_t\|^2 < \frac{1}{1000}K^{-2}\eta^{-\frac{1}{2}}$. Therefore, $\|\boldsymbol{x}_t\|^2 > \frac{1}{200}K^{-2}\eta^{-\frac{1}{2}}$. $\square$

After we have the alignment guarantee, we can prove that we will enter the feasible regime for the scalar case with high probability. Still, we denote $c_t = (\boldsymbol{x}_t^\top \boldsymbol{y}_t)^2$.

**Lemma 22.** *(Feasible regime guarantee) If $\xi_t < \eta^2$, $\eta < \frac{1}{8000000}K^{-4}$, $-\frac{1}{100} < \alpha(t) < \frac{1}{2}K^{-2}$ for all $t \in [t_1, t_1 + 6K^3 + 4K]$, then there exists some $T < 6K^3 + 4K$ such that for all $t \in [t_1 + T, t_1 + 6K^3 + 4K]$, $|\|\boldsymbol{x}_t\|\|\boldsymbol{y}_t\| - 1| < K^{-1}$.*

*Proof.* To prove the statement, we try to prove a stronger statement:

$$c_t \in (1 - K^{-1}, 1 + K^{-1} - K^{-2}).$$

If this statement holds, because $\xi_t < \eta^2$, the product of the norms are

$$\|\boldsymbol{x}_t\|^2 \|\boldsymbol{y}_t\|^2 \in (1 - K^{-1}, 1 + K^{-1} - K^{-2} + \eta^2)$$

So we have $\|\boldsymbol{x}_t\|\|\boldsymbol{y}_t\| \in (1 - \frac{2}{3}K^{-1}, 1 + \frac{1}{2}K^{-2})$ and the lemma is proved. So in the rest of the proof, we will prove this stronger statement.

First we prove that there exists some time $t < 4K + 1$ that the square of the inner product $c_t$ will be larger than $1 - K^{-1}$.

If $c_t \in (\frac{7}{20}, 1/2)$, then

$$c_{t+1} > c_t(1 + 1 \times (1 - 1/4 - \eta^2 - 1/4))^2 > \frac{7}{20} \times 1.4^2 > 1/2.$$

Thus $c_t$ will provably enter $c_t > 1/2$ in one step. And if $1/2 < c_t < 1 - K^{-1}$, we have

$$c_{t+1} > c_t(1 + (1 - c_t - \eta^2))^2 > c_t(1 + \frac{1}{2}K^{-1}) > c_t + \frac{1}{4}K^{-1},$$

which means it takes at most $4K$ steps for $c_t$ to become larger than $1 - K^{-1}$.

Now we prove if $c_t > 1 - K^{-1}$, then it will take at most $T = 6K^3$ steps s.t. for $t' > t + T$, $c_{t'} \in (1 - K^{-1}, 1 + K^{-1} - K^{-2})$, which finishes the proof. We first prove there exists some $t$, $c_t \in (1 - K^{-1}, 1 + K^{-1} - K^{-2})$; afterwards we prove that once $c_t$ gets in, it will never get out of the region.

If at first $c_t \in (1 - K^{-1}, 1 + K^{-1} - K^{-2})$, the proof is done. Otherwise, we suppose $c_t > 1 + K^{-1} - K^{-2}$. Next, we consider the two step dynamics of $c_t$ and prove it will decay by a constant factor every two steps.

First we pick out the relatively small terms, and find out the main part of the two step dynamics. If $\alpha(t) < \alpha_0 < 1/8$, and $1 + K^{-1} - K^{-2} < c_t < 1.3$, we have:

$$c_{t+1} \le c_t(1 + (1 + \alpha(t))(1 - c_t) + 10\eta^2)^2$$
$$c_{t+1} \ge c_t(1 + (1 + \alpha(t))(1 - c_t) - 2\eta^2)^2$$
$$c_{t+2} \le c_{t+1}(1 + (1 + \alpha(t+1))(1 - c_{t+1}) + 10\eta^2)^2$$
$$< c_t(1 + (1 + \alpha(t))(1 - c_t) + 10\eta^2)^2(1 + (1 + \alpha(t+1))(1 - c_{t+1}) + 10\eta^2)^2$$

Then we upper bound the difference $\alpha(t+1) - \alpha(t)$ in one step. Consider the dynamics of $\|\boldsymbol{x}\|^2 + \|\boldsymbol{y}\|^2$. The difference each step is at most

$$\|\boldsymbol{x}'\|^2 + \|\boldsymbol{y}'\|^2 - \|\boldsymbol{x}\|^2 - \|\boldsymbol{y}\|^2$$
$$= (\|\boldsymbol{x}\|^2 + \|\boldsymbol{y}\|^2)(\eta^2(\boldsymbol{x}^\top \boldsymbol{y})^2((\|\boldsymbol{x}\|^2\|\boldsymbol{y}\|^2 - 1)^2 + (2 - \frac{3}{4}\|\boldsymbol{x}\|^2\|\boldsymbol{y}\|^2)(\|\boldsymbol{x}\|^2\|\boldsymbol{y}\|^2 - (\boldsymbol{x}^\top \boldsymbol{y})^2)))$$
$$+ 4\eta(\boldsymbol{x}^\top \boldsymbol{y})^2(1 - \|\boldsymbol{x}\|^2\|\boldsymbol{y}\|^2)$$
$$< \eta \cdot (1 + \alpha(t)) \cdot \frac{13}{10} \cdot (1 + 1) + 2\eta < 5\eta$$

Then the corresponding update of $\alpha(t)$ will be less than $5\eta^2$. That means we have

$$c_{t+2} < c_t(1 + (1 + \alpha(t))(1 - c_t) + 10\eta^2)^2(1 + (1 + \alpha(t+1))(1 - c_{t+1}) + 10\eta^2)^2$$
$$< c_t(1 + (1 + \alpha(t))(1 - c_t) + 10\eta^2)^2(1 + (1 + \alpha(t))(1 - c_{t+1}) + 15\eta^2)^2$$
$$< c_t(1 + (1 + \alpha(t))(1 - c_t))^2(1 + (1 + \alpha(t))(1 - c_{t+1}))^2 + 100\eta^2$$
$$< c_t(1 + (1 + \alpha(t))(1 - c_t))^2(1 + (1 + \alpha(t))(1 - c_t(1 + (1 + \alpha(t))(1 - c_t) - 2\eta^2)^2))^2 + 100\eta^2$$
$$< c_t(1 + (1 + \alpha(t))(1 - c_t))^2(1 + (1 + \alpha(t))(1 - c_t(1 + (1 + \alpha(t))(1 - c_t))^2))^2 + 200\eta^2$$
$$\tag{128}$$

Similarly, we can have the lower bound of the 2-step dynamics.

$$
\begin{aligned}
c_{t+2} &> c_t(1 + (1+\alpha(t))(1-c_t) - 2\eta^2)^2(1 + (1+\alpha(t+1))(1-c_{t+1}) - 2\eta^2)^2 \\
&> c_t(1 + (1+\alpha(t))(1-c_t) - 2\eta^2)^2(1 + (1+\alpha(t))(1-c_{t+1}) - 7\eta^2)^2 \\
&> c_t(1 + (1+\alpha(t))(1-c_t))^2(1 + (1+\alpha(t))(1-c_{t+1}))^2 - 100\eta^2 \\
&> c_t(1 + (1+\alpha(t))(1-c_t))^2(1 + (1+\alpha(t))(1 - c_t(1 + (1+\alpha(t))(1-c_t) + 10\eta^2)^2))^2 - 100\eta^2 \\
&> c_t(1 + (1+\alpha(t))(1-c_t))^2(1 + (1+\alpha(t))(1 - c_t(1 + (1+\alpha(t))(1-c_t))^2))^2 - 200\eta^2
\end{aligned}
$$
(129)

After bound all small terms to $200\eta^2$, we consider the main part

$$
c_t(1 + (1+\alpha(t))(1-c_t))^2(1 + (1+\alpha(t))(1 - c_t(1 + (1+\alpha(t))(1-c_t))^2))^2
$$

To prove it will decrease, we need to prove the factor

$$
f(\alpha, c) = (1 + (1+\alpha)(1-c))(1 + (1+\alpha)(1 - c(1 + (1+\alpha)(1-c))^2)) < 1
$$

when $\alpha \in (-\frac{1}{100}, \min\{\frac{1}{2}K^{-2}, 1/8\}), 1 + K^{-1} - K^{-2} < c < 1.3$.

We first prove that the function $f(\alpha, c)$ monotonically increases when $\alpha > -\frac{1}{100}$ increases, i.e. $\frac{\partial f(\alpha, c)}{\partial \alpha} > 0$. We directly give the simplified expression of this partial derivative.

$$
\begin{aligned}
\frac{\partial f(\alpha, c)}{\partial \alpha} &= (c-1)(-4 - 2\alpha + c(1+\alpha)[\frac{15}{16} + (2c(1+\alpha) - \frac{17}{4} - 2\alpha)^2]) \\
&= (c-1)(-4 - 2\alpha + (1+\alpha)[\frac{15}{16}c + (2c(1+\alpha) - \frac{17}{4} - 2\alpha)^2 c]) \\
&> (c-1)(-2\alpha - 4 + (1+\alpha)(\frac{15}{16}c + \frac{81}{4}(1 - \frac{1}{2}c)^2 c)) \quad (\text{since } -\frac{1}{100} < \alpha < 1/8) \\
&> (c-1)(-2\alpha - 4 + (1+\alpha) \times 4) \qquad (c \in (1, \frac{13}{10})) \\
&> 0.
\end{aligned}
$$
(130)

Therefore, $f(\alpha, c) < f(\frac{1}{2}K^{-2}, c)$ for all $1 + K^{-1} - K^{-2} < c < 1.3$. Denote $\beta = \frac{1}{2}(K^{-2} - 2K^{-3} + K^{-4})$ and suppose $c = 1 + t\sqrt{\beta}$ for some $t > \sqrt{2}$ (since $c > 1 + K^{-1} - K^{-2}$). Meanwhile, since $t\sqrt{\beta} < 0.3, t < 0.3\beta^{-1/2}$.

Then we plug in $c = 1 + t\sqrt{\beta}$ and expand $f(\beta, c)$.

$$
\begin{aligned}
f(\beta, c) &= f(\beta, 1 + t\sqrt{\beta}) \\
&= 1 + 2t\beta^{3/2} - 2t^3\beta^{3/2} - 3t^2\beta^2 + t^4\beta^2 + 2t\beta^{5/2} - 5t^3\beta^{5/2} - 6t^2\beta^3 + 4t^4\beta^3 - 3t^3\beta^{7/2} \\
&\quad - 3t^2\beta^4 + 6t^4\beta^4 + t^3\beta^{9/2} + 4t^4\beta^5 + t^3\beta^{11/2} + t^4\beta^6 \\
&\leq 1 + (2t - t^3)\beta^{3/2} - t^3\beta^{3/2} - 3t^2\beta^2 + 0.3t^3\beta^{3/2} + 2t\beta^{5/2} - 5t^3\beta^{5/2} - 6t^2\beta^3 + 1.2t^3\beta^{5/2} \\
&\quad - 3t^3\beta^{7/2} - 3t^2\beta^4 + 1.8t^3\beta^{7/2} + 0.3t^2\beta^4 + 0.36t^2\beta^4 + 0.3t^2\beta^5 + 0.09t^2\beta^5 \\
&< 1 - 0.7t^3\beta^{3/2} - 3t^2\beta^2 < 1 - \beta^{3/2} - 6\beta^2
\end{aligned}
$$
(131)

Thus we have

$$
c_{t+2} < c_t(1 - \beta^{3/2} - 6\beta^2)^2 + 200\eta^2 < c_t - \beta^{3/2} < c_t - \frac{1}{10}K^{-3}
$$

Thus it takes at most $2 \times 0.3/(\frac{1}{10}K^{-3}) < 6K^3$ steps for $c_t$ to get into the region $c_t \in (1 - K^{-1}, 1 + K^{-1} - K^{-2})$.

Finally we prove that if $c_s \in (1 - K^{-1}, 1 + K^{-1} - K^{-2})$ for some $s$, then for all $t > s$,

$$
c_t \in (1 - K^{-1}, 1 + K^{-1} - K^{-2}).
$$

We use induction and suppose $c_t$ satisfies the condition above. Note that we have the upper and lower bound of $c_{t+1}$ above:

$$c_{t+1} \leq c_t(1 + (1 + \alpha(t))(1 - c_t) + 10\eta^2)^2 \leq c_t(1 + (1 + \alpha(t))(1 - c_t))^2 + 100\eta^2$$
$$c_{t+1} \geq c_t(1 + (1 + \alpha(t))(1 - c_t) - 2\eta^2)^2 \geq c_t(1 + (1 + \alpha(t))(1 - c_t))^2 - 100\eta^2$$

Meanwhile $c_t(1 + (1 + \alpha(t))(1 - c_t))^2$ is monotonically decreasing with $c_t$ when $-\frac{1}{100} < \alpha(t) < \frac{1}{2}K^{-2}$ and $c_t \in (1 - K^{-1}, 1 + K^{-1} - K^{-2})$. Thus we have:

$$
\begin{aligned}
c_{t+1} &\leq c_t(1 + (1 + \alpha(t))(1 - c_t))^2 + 100\eta^2 \\
&\leq (1 - K^{-1})(1 + K^{-1}(1 + \frac{1}{2}K^{-2}))^2 + 100\eta^2 \\
&= 1 + K^{-1} - K^{-2}(1 + \frac{1}{2}K^{-2})^2 + K^{-3} - K^{-3}(1 + \frac{1}{2}K^{-2})^2 + 100\eta^2 \\
&< 1 + K^{-1} - K^{-2}.
\end{aligned}
$$

$$
\begin{aligned}
c_{t+1} &\geq c_t(1 + (1 + \alpha(t))(1 - c_t))^2 - 100\eta^2 \\
&> (1 + K^{-1} - K^{-2})(1 - (K^{-1} - K^{-2})(1 + \frac{1}{2}K^{-2}))^2 - 100\eta^2 \\
&= 1 - K^{-1} + 2K^{-3} - 3K^{-4} + 4K^{-5} - \frac{15}{4}K^{-6} + \frac{11}{4}K^{-7} - \frac{3}{2}K^{-8} \\
&\quad + \frac{3}{4}K^{-9} - \frac{1}{4}K^{-10} - 100\eta^2 \\
&> 1 - K^{-1}.
\end{aligned}
$$

Therefore, by induction we prove that for all $t > s$, $c_t \in (1 - K^{-1}, 1 + K^{-1} - K^{-2})$. In this way, $|\|\boldsymbol{x}_t\|\|\boldsymbol{y}_t\| - 1| < K^{-1}$ for $t > t_1 + T$. $\qquad\square$

Entering the feasible region is not enough for establishing the equivalence to the scalar case. Furthermore, we need the alignment variable $\xi_t$ to be small enough, such that some $\mathcal{O}(\cdot)$ notation term can contain all the terms with $\xi$ in the $(a, b)$-parameterization dynamics.

However, we need to guarantee that the trajectory does not converge to an unstable point near the minima. We can prove that for the scalar case if the initialization is not exactly on the manifold of minimizers, but it becomes more challenging in higher dimensions. Therefore, we require an additional perturbation to escape from any unstable point. We pick the time $t_p = T_0 + 6K^3 + 4K = \mathcal{O}(-\log(\eta) + \log(d) - \log(\delta_0) + K^3)$ to guarantee that when the perturbation happens, $\boldsymbol{x}$ and $\boldsymbol{y}$ are aligned and $(\boldsymbol{x}^\top \boldsymbol{y})^2$ is not large enough to cause instability. After the perturbation at $t_p$, the gradient descent dynamics prevent the objective from hitting the manifold of minimizers.

The following lemma proves that the properties of the bound of $c_t := (\boldsymbol{x}^\top \boldsymbol{y})^2$, $\alpha = \eta(\|\boldsymbol{x}\|^2 + \|\boldsymbol{y}\|^2) - 1$ and $\xi$ are still valid after the perturbation.

**Lemma 23.** *After the perturbation at time $t_p$, we have the following properties hold in $t \in [t_p, 2t_p + 2]$: (i) After the perturbation $c'_{t_p} \in (1 + K^{-1} - 2K^{-2}, 1 + 3K^{-1} + 2K^{-2})$, and $c_t \in (\frac{7}{20}, \frac{13}{10})$; (ii) $\frac{K^{-2}}{100} < \alpha(t) < \frac{1}{2}K^{-2}$; (iii) $\xi_t < (0.7)^{t - t_p}\eta^2$.*

*Proof.* We prove this property by induction. Firstly, we prove the basis of induction at $t = t_p$.

For (i), before perturbation we have $c_{t_p} \in (1 - K^{-1}, 1 + K^{-1} - K^{-2})$, so after the perturbation we have $c'_{t_p} \in (1 + K^{-1} - 2K^{-2}, 1 + 3K^{-1} + 2K^{-2}) \subset (\frac{7}{20}, \frac{13}{10})$.

For (ii), the initial value of $\alpha(t)$ satisfies that $\frac{1}{40}K^{-2} < \alpha(0) < \frac{1}{4}K^{-2}$. But the total movement of $\alpha(t)$ before perturbation is bounded within $(-5\eta^2 t_p, 5\eta^2 t_p) \subset (-\frac{K^{-2}}{200}, \frac{K^{-2}}{200})$, and the perturbation introduce a movement of $\|\boldsymbol{y}\|^2$ by $\|\boldsymbol{y}_t\|^2 < \frac{13}{10}/\|\boldsymbol{x}_t\|^2 < \frac{1}{1000}K^{-2}\eta^{-\frac{1}{2}}$. Due to the upper bound of $\eta$, $\alpha(t) \in (\frac{K^{-2}}{100}, \frac{K^{-2}}{2})$.

For (iii), since $K > 512$, thus at $t_p$, $\xi_{t_p} < (0.7)^{6K^3}\xi_t < (0.7)^{6K^3}\eta^2$. After the perturbation $\xi'_{t_p} = (1 + 2K^{-1})^2\xi_{t_p} < \eta^2$. So all three statement holds for $t = t_p$.

Then we suppose for $t \in [t_p, t_p + t_1]$ all three statement holds and prove them for $t_p + t_1 + 1$. First by the dynamics of $c_t$ with condition $c_t \in (\frac{7}{20}, \frac{13}{10})$ and $0 < \alpha(t) < \frac{K^{-2}}{2}$, we have $c_{t+1} < c_t(1 + (1 + \frac{K^{-2}}{2})(1 - c_t))^2 < 13/10$ and $c_{t+1} > c_t(1 + (1 - c_t) - 10\eta^2)^2 > \frac{7}{20}$. Statement (i) is proved.

For (ii), since $c_t$ and $\xi_t$ are bounded for $t \le t_p + t_1 + 1$, we can still have the total movement of $\alpha(t)$ lies in $(-5\eta^2 t_p, 5\eta^2 t_p) \subset (-\frac{K^{-2}}{200}, \frac{K^{-2}}{200})$. Combine with the movement of $\boldsymbol{y}$ at perturbation $(< \frac{1}{1000}K^{-2}\eta^{-\frac{1}{2}})$, the total movement is still bounded by $(-\frac{6K^{-2}}{1000}, \frac{6K^{-2}}{1000})$, which proves the bound in (ii).

Finally for (iii), as long as $c_t < \frac{13}{10}$, $\xi_{t+1} < 0.7\xi_t < 0.7^{t_1+1}\eta^2$. Therefore all three statements are proved by induction. $\qquad\square$

After reclaiming all the bounds after the perturbation, we need to prove that the alignment variable $\xi$ will be small enough to approximate the scalar case. The following lemma proves that $b$ stays at a constant level when $\alpha(t)$ is some constant.

**Lemma 24.** *Suppose $\xi_t < \eta^2$, $\eta < \frac{1}{8000000}K^{-4}$, $\frac{K^{-2}}{100} < \alpha(t) < \frac{1}{2}K^{-2}$ for all $t \in [t_1, 2t_1 + 2]$ for some $t_1$. If $(\boldsymbol{x}_{t_1}^\top \boldsymbol{y}_{t_1})^2 \in (1 + K^{-1} - 2K^{-2}, 1 + 3K^{-1} + 2K^{-2})$, then for all $t \in [t_1, 2t_1]$, $(\boldsymbol{x}_{t_1+2k}^\top \boldsymbol{y}_{t_1+2k})^2 \in (1 + \frac{1}{20}K^{-1}, 1 + 3K^{-1} + 2K^{-2})$, $(\boldsymbol{x}_{t_1+2k-1}^\top \boldsymbol{y}_{t_1+2k-1})^2 < 1 - \frac{1}{20}K^{-1}$, $k \in \mathbb{N}$, $k < t_1/2$.*

*Proof.* Denote $c_t := (\boldsymbol{x}_t^\top \boldsymbol{y}_t)^2$. Here we consider the two step dynamics of the inner product (Eq. (128) and Eq. (129)).

$$c_{t+2} < c_t(1 + (1 + \alpha(t))(1 - c_t))^2(1 + (1 + \alpha(t))(1 - c_t(1 + (1 + \alpha(t))(1 - c_t))^2))^2 + 200\eta^2$$
$$c_{t+2} > c_t(1 + (1 + \alpha(t))(1 - c_t))^2(1 + (1 + \alpha(t))(1 - c_t(1 + (1 + \alpha(t))(1 - c_t))^2))^2 - 200\eta^2$$

We first consider the lower bound of the $(\boldsymbol{x}_{t_1+2k}^\top \boldsymbol{y}_{t_1+2k})^2$. We prove the sequence of the $c_{t+2k}$, $k = 0, 1, 2, ...$ by induction, and then use this conclusion to prove the upper bound of $c_{t+2k-1}$. We know that $k = 0$ the statement holds. Then suppose the lower bound holds for $k \le k_0$, and we prove it for $k = k_0 + 1$.

Here we consider the function as we do in Lemma 22:

$$f(\alpha, c) = (1 + (1 + \alpha)(1 - c))(1 + (1 + \alpha)(1 - c(1 + (1 + \alpha)(1 - c))^2)).$$

Denote $\gamma = \frac{1}{100}K^{-2}$. By Eq. (130), we know

$$f(\alpha, c) > f(\gamma, c)$$

Suppose $c_{t_1+2k} = 1 + q\sqrt{\gamma}$, $q \in (\frac{1}{2}, 30 + 40K^{-1})$. Then the expression $f(\gamma, c)$ becomes:

$$
\begin{aligned}
f(\gamma, c) &= f(\gamma, 1 + q\sqrt{\gamma}) \\
&= 1 + 2q\gamma^{3/2} - 2q^3\gamma^{3/2} - 3q^2\gamma^2 + q^4\gamma^2 + 2q\gamma^{5/2} - 5q^3\gamma^{5/2} - 6q^2\gamma^3 + 4q^4\gamma^3 - 3q^3\gamma^{7/2} \\
&\quad - 3q^2\gamma^4 + 6q^4\gamma^4 + q^3\gamma^{9/2} + 4q^4\gamma^5 + q^3\gamma^{11/2} + q^4\gamma^6
\end{aligned}
$$

Notice that when $c > 1$, $f(\gamma, c)$ decrease as $c$ increase. Now we consider the range of $q$: If $q \in (\frac{1}{2}, \frac{2}{3}]$, we have:

$$
\begin{aligned}
f(\gamma, c) &\ge f(\gamma, 1 + \frac{2}{3}\sqrt{\gamma}) \\
&= \frac{20\gamma^{3/2}}{27} - \frac{4\gamma^{5/2}}{27} - \frac{8\gamma^{7/2}}{9} + \frac{8\gamma^{9/2}}{27} + \frac{8\gamma^{11/2}}{27}
\end{aligned}
$$

$$+ \frac{16\gamma^6}{81} + \frac{64\gamma^5}{81} - \frac{4\gamma^4}{27} - \frac{152\gamma^3}{81} - \frac{92\gamma^2}{81} + 1$$

$$> 1 + \frac{10}{27}\gamma^{3/2}. \qquad (\text{since } K > 512, \gamma^{1/2} = \frac{K^{-1}}{10})$$

That means $c_{t_1+2k+2} > ((1 + \frac{10}{27}K^{-2})^2 - 200\eta^2)c_{t_1+2k} > c_{t_1+2k} > (1 + \frac{K^{-1}}{20})$ and the proof is done. Otherwise, we have $q \in (\frac{2}{3}, 30 + 40K^{-1})$, also we have the lower bound of this function:

$$f(\gamma, c) \geq f(\gamma, 1 + \frac{2}{3}\sqrt{\gamma})$$

$$= -127920\gamma^{3/2} - 319920\gamma^{5/2} - 192000\gamma^{7/2} + 64000\gamma^{9/2} + 64000\gamma^{11/2}$$

$$+ 2560000\gamma^6 + 10240000\gamma^5 + 15355200\gamma^4 + 10230400\gamma^3 + 2555200\gamma^2 + 1$$

$$> 1 - 100000\gamma^{3/2}. \qquad (\text{since } K > 512, \gamma^{1/2} = \frac{K^{-1}}{10})$$

That means

$$c_{t_1+2k+2} > ((1 - 100000\gamma^{3/2})^2 - 200\eta^2)c_{t_1+2k}$$

$$> (1 - 200000\gamma^{3/2} - 200\eta^2)(1 + \frac{2}{3}\sqrt{\gamma})$$

$$> (1 - \frac{200000}{26214400}\gamma^{1/2} - 200\eta^2)(1 + \frac{2}{3}\sqrt{\gamma})$$

$$> (1 - \frac{1}{1000}K^{-1} - \frac{1}{4000}K^{-4})(1 + \frac{K^{-1}}{15})$$

$$> (1 + \frac{K^{-1}}{20})$$

Then the lower bound of $c_{t_1+2k}$ is proved.

As for the upper bound, since when $t = t_1 + 2k$, $\boldsymbol{x}, \boldsymbol{y}$ satisfies all the conditions in Lemma 22. If $c_{t_1+2k} < 1 + K^{-1} - K^{-2}$, we have proved that it will never be larger than $1 + K^{-1} - K^{-2}$. Otherwise if $c_{t_1+2k} \in (1 + K^{-1} - K^{-2}, 1 + 3K^{-1} + 2K^{-2})$ We apply the inequality Eq. (131) and know $c_{t_1+2k+2} < c_{t_1+2k}$. In this way, by induction we prove the bound $(\boldsymbol{x}_{t_1+2k}^\top \boldsymbol{y}_{t_1+2k})^2 \in (1 + \frac{1}{20}K^{-1}, 1 + 3K^{-1} + 2K^{-2})$.

As for the upper bound of $(\boldsymbol{x}_{t_1+2k-1}^\top \boldsymbol{y}_{t_1+2k-1})^2$, we directly apply the upper bound of 1-step dynamics (if $c_t > 1 + \frac{1}{20}K^{-1}$):

$$c_{t+1} \leq c_t(1 + (1 + \alpha(t))(1 - c_t) + 10\eta^2)^2$$

$$< c_t(2 - c_t + 10\eta^2)^2$$

$$< (1 + \frac{K^{-1}}{20})(1 - \frac{K^{-1}}{20})^2 + 100\eta^2$$

$$< 1 - \frac{K^{-1}}{20} - \frac{1}{400}K^{-2} + \frac{1}{8000}K^{-3} + 100\eta^2$$

$$< 1 - \frac{1}{20}K^{-1}$$

Therefore the upper bound $(\boldsymbol{x}_{t_1+2k-1}^\top \boldsymbol{y}_{t_1+2k-1})^2 < 1 - \frac{1}{20}K^{-1}$ holds for all $k < t_1/2$.

$\square$

Finally, we denote $a := \sqrt{\|\boldsymbol{x}\|^2 - \|\boldsymbol{y}\|^2 - (\eta^{-2} - 4)^{\frac{1}{2}}}, b := \|\boldsymbol{x}\|\|\boldsymbol{y}\| - 1$. With all the lemmas above, we prove the equivalence between the dynamics of $(a, b)$ and the one step dynamics of in the scalar case (Lemma 1, Lemma 2). For simplicity of notations, when analyzing the 1-step and 2-step dynamics of $(a_t, b_t)$, we use $a, a'$ to denote $a_t, a_{t+1}$ and $b, b'$ to denote $b_t, b_{t+1}$, etc. For simplicity of calculation, we consider the change of variable $\kappa = \sqrt{\eta}$.

**Lemma 25.** *(Equivalence with scalar updates) If $\xi_t < \min\{\eta^2, \eta b_t^4\}$, $\|\boldsymbol{x}_t\| \in (\breve{x} + \frac{1}{200}K^{-2}\eta^{-\frac{1}{2}}, \breve{x} + \frac{1}{4}K^{-2}\eta^{-\frac{1}{2}})$ and $|\|\boldsymbol{x}_t\|\|\boldsymbol{y}_t\| - 1| < K^{-1}$, the following equations hold for some fixed constant $\epsilon = \min\{0.5, \frac{1}{125}K^{-1}\}$:*

$$
\begin{aligned}
a' &= a - 2b^2\kappa^3 + \mathcal{O}(\epsilon b^2\kappa^3) + \mathcal{O}(b^2\kappa^4). \\
b' &= -b - 4ab\kappa - 3b^2 - b^3 + \mathcal{O}(ab^2\kappa) + \mathcal{O}(a^2b\kappa^2) + \mathcal{O}(b^2\kappa^4) + \mathcal{O}(b\kappa^5).
\end{aligned}
\tag{132}
$$

*Proof.* For simplicity, we denote $x = \|\boldsymbol{x}\|$ and $y = \|\boldsymbol{y}\|$. We first prove that:

$$
\begin{aligned}
x' &= x + \kappa^2 xy^2(1 - x^2y^2) + \mathcal{O}(\kappa^4 b^2). \\
y' &= y + \kappa^2 x^2 y(1 - x^2y^2) + \mathcal{O}(\kappa^2 b^2).
\end{aligned}
\tag{133}
$$

We suppose $\boldsymbol{y} = c\boldsymbol{x} + \boldsymbol{\theta}$ for some $\boldsymbol{\theta} \in \mathbb{R}^d$ and $\langle \boldsymbol{\theta}, \boldsymbol{x}\rangle = 0$. In this way, we have $\boldsymbol{x}^\top\boldsymbol{y} = cx^2, y^2 = c^2x^2 + \|\boldsymbol{\theta}\|^2, \xi = x^2\|\boldsymbol{\theta}\|^2$. Since $\xi < \eta b^4$ and $x = \mathcal{O}(\kappa^{-1})$, $\|\boldsymbol{\theta}\| = \mathcal{O}(b^2\kappa^2)$.

Then check the dynamics of $\boldsymbol{x}$ and plug in $\boldsymbol{y} = c\boldsymbol{x} + \boldsymbol{\theta}$.

$$
\boldsymbol{x}' = \boldsymbol{x} + \eta\left((\boldsymbol{x}^\top y)y - \frac{1}{2}(\boldsymbol{x}^\top\boldsymbol{y})x^2y^2\boldsymbol{y} - \frac{1}{2}(\boldsymbol{x}^\top\boldsymbol{y})^2y^2\boldsymbol{x}\right)
$$

$$
(1 + \eta y^2 - \eta x^2y^4)\boldsymbol{x} + (-\eta\|\boldsymbol{\theta}\|^2 + \eta x^2y^2\|\boldsymbol{\theta}\|^2)\boldsymbol{x} + \left(\eta(\boldsymbol{x}^\top\boldsymbol{y}) - \frac{1}{2}\eta(\boldsymbol{x}^\top\boldsymbol{y})x^2y^2\right)\boldsymbol{\theta}.
$$

Since $(\boldsymbol{x}^\top\boldsymbol{y})$ and $xy$ are both bounded as constant, we can directly take the norm of both sides and with triangle inequality we have:

$$
x' = x + \kappa^2 xy^2(1 - x^2y^2) + \mathcal{O}(b^2\kappa^4).
$$

Similarly, we have the dynamics of $y$.

$$
y' = y + \kappa^2 x^2 y(1 - x^2y^2) + \mathcal{O}(b^2\kappa^2).
$$

We now reparameterize the dynamics in $(a, b)$-parameterization.

$$
\begin{aligned}
a' &= \left(a + \left(\kappa^{-4} - 4\right)^{\frac{1}{4}}\right)\left(1 - b^2(1+b)^2(2+b)^2\kappa^4\right)^{\frac{1}{2}} - \left(\kappa^{-4} - 4\right)^{\frac{1}{4}} + \mathcal{O}(b^2\kappa^4). \\
b' &= b - 2b\delta - 3b^2\delta - b^3\delta + 4b^2\kappa^4 + 16b^3\kappa^4 + 25b^4\kappa^4 + 19b^5\kappa^4 + 7b^6\kappa^4 + b^7\kappa^4 + \mathcal{O}(ab^2\kappa^2)
\end{aligned}
$$

$$
\text{where } \delta \triangleq \sqrt{4\eta^2(1+b)^2 + \left(a\eta^{\frac{1}{2}} + (1 - 4\eta^2)^{\frac{1}{4}}\right)^4}.
\tag{134}
$$

And by the bound of $\|\boldsymbol{x}_t\|$ and $|\|\boldsymbol{x}_t\|\|\boldsymbol{y}_t\| - 1|$, we can have $|a| < \epsilon\kappa, |b| < \min\{1, \frac{1}{5}\epsilon\kappa^{-2}\}$, which satisfies the condition in Lemma 1 and Lemma 2. Then we follow the proof of Lemma 1 and Lemma 2 to finish the proof (since the only difference is the two $\mathcal{O}(\cdot)$ notation). $\square$

After we can reduce the dynamics of vector case to the scalar case dynamics, we need to prove that in the scalar case (in all stages), for any $t \geq 0$, $(\frac{b_{t+1}}{b_t})^4 > 0.7$, which means $\xi_t$ shrinks faster than $b_t^4$. In that case, we can conclude that $\xi_t > \eta b_t^4$ always holds along the scalar case trajectory.

**Lemma 26.** *Suppose all conditions in Theorem 3.1 hold. Then for all $t > 0$, $(\frac{b_{t+1}}{b_t})^4 > 0.7$.*

*Proof.* From the proof of Theorem 3.1, we know that for all $t > 0$, $|b_t| < K^{-1}$ and $|a_t| < \kappa^{-1}$. Thus Lemma 2 holds and for all $t \geq 0$ we have

$$
b_{t+1} = -b_t - 4a_tb_t\kappa - 3b_t^2 - b_t^3 + \mathcal{O}(a_tb_t^2\kappa) + \mathcal{O}(a_t^2b_t\kappa^2) + \mathcal{O}(b_t^2\kappa^4) + \mathcal{O}(b_t\kappa^5).
\tag{135}
$$

Since we know all constants hidden by the $\mathcal{O}(\cdot)$ operator are upper bounded by $K$, we have the following lower bound on the multiplicative update of $b$:

$$
\frac{b_{t+1}^4}{b_t^4} \geq \left(1 - 4a_t\kappa - 3b_t - b_t^2 - Ka_tb_t\kappa - Ka_t^2\kappa^2 - Kb_t\kappa^4 - K\kappa^5\right)^4
$$

$$\geq (1 - 2K^{-2} - 3K^{-1} - K^{-2} - K^{-2} - \kappa - \kappa^4 - \kappa^4)^4$$

$$\geq (1 - 10K^{-1})^4 > \left(\frac{502}{512}\right)^4 > 0.7.$$

$\square$

Finally we conclude the proof of Theorem C.1.

*Proof of Theorem C.1.* By Lemma 21, we know with probability $1 - 2\delta_0 - 2\exp\{-\Omega(d)\}$, there exists some time $T_0 = \mathcal{O}(-\log(\eta) - \log(\delta_0) + \log(d))$ that $\xi_t < \eta^2$ for all $t > T_0$. Also for $t \in [T_0, T_0 + 6K^3 + 4K]$, $\|\boldsymbol{x}_t\| \in (\breve{x} + \frac{1}{200}K^{-2}\eta^{-\frac{1}{2}}, \breve{x} + \frac{1}{6}K^{-2}\eta^{-\frac{1}{2}})$. This bound of $\|\boldsymbol{x}\|$ guarantees that for $t \in [T_0, T_0 + 6K^3 + 4K]$, $\alpha(t) \in (0, \frac{K^{-2}}{2})$, which satisfies the condition of Lemma 22.

Then by Lemma 22, we know for some $t^* < T_0 + 6K^3 + 4K < t_p$, $|\|\boldsymbol{x}_t\|\|\boldsymbol{y}_t\| - 1| < K^{-1}$ for $t \in [t^*, t_p]$. After entering and staying in the region, we add the perturbation at $t_p$. By Lemma 23, we have the bounds before perturbation still hold: (i) After the perturbation $c'_{t_p} \in (1 + K^{-1} - 2K^{-2}, 1 + 3K^{-1} + 2K^{-2})$, and $c_t \in (\frac{7}{20}, \frac{13}{10})$; (ii) $\frac{K^{-2}}{100} < \alpha(t) < \frac{1}{2}K^{-2}$; (iii) $\xi_t < (0.7)^{t - t_p}\eta^2$. Therefore, the condition of Lemma 22 and Lemma 24 are both met. So after another $6K^3 + 4K$ steps, we have $|b_t| = |\|\boldsymbol{x}_t\|\|\boldsymbol{y}_t\| - 1| < K^{-1}$ and $|(\boldsymbol{x}_t^\top \boldsymbol{y}_t)^2 - 1| > 1 - \frac{K^{-1}}{20}$. The second expression can lead to $|b_t| > \frac{K^{-1}}{20}/(1 + \|\boldsymbol{x}_t\|\|\boldsymbol{y}_t\|) > \frac{K^{-1}}{41}$. This means $\xi_t < (0.7)^{4K}\eta^2 < (0.7)^{2000}\eta \cdot \frac{K^{-4}}{8000000} < \eta|b_t|^4$.

By Lemma 25, we know the dynamics of the norm of the vectors $(\|\boldsymbol{x}\|, \|\boldsymbol{y}\|)$ can be captured by the scalar case (including the initialization condition and the one step update rules). And by Lemma 26, we know the alignment will be kept and the dynamics of the vectors will always be true.

Finally, we apply Theorem 3.1 and finish the proof. $\square$

