# OpenReview forum: "Understanding Edge-of-Stability Training Dynamics with a Minimalist Example"
_ICLR.cc/2023/Conference — ICLR 2023 poster_

### Official Review · Reviewer_DD1S · 2022-10-19

**Confidence:** 4
**Correctness:** 4
**Technical Novelty And Significance:** 3
**Empirical Novelty And Significance:** 3
**Recommendation:** 8

**Clarity, Quality, Novelty And Reproducibility:**

The paper is overall well written and clear, though I have two small remarks, see below. The mathematics seem correct. I am not extremely familiar with this line of work, so cannot assert the novelty of this work with certainty, but the related works section is detailed and appears complete.

Small remarks:
 - I found the summary of the results in page 2 a bit confusing, it almost seems that the theorem and its corollary are the same statement. If I understand correctly the sets  $\mathbb S$ (in the theorem) and  $\mathbb S$ (in the corollary) are different. In the theorem  $\mathbb S$ depends on the learning rate, while in the corollary  $\mathbb S$ applies to whole range? I think it would be much clearer if you would write $\mathbb{S}_\lambda$ to emphasize the dependence on $\lambda$.
- I think there is a typo in Equation 6 on page 5, the second line should start with $b_{t+1} = b_t + [\dots]$ instead of $b_{t+1} = b + [\dots]$.

**Strength And Weaknesses:**

The phenomenon of 'edge of stability' is the subject of much interest and while there are a number of results that assume convergence at a $\eta$-stable global minima, it is important to also understand how the network converges to such points.

Despite its simplicity, the model exhibit similar behavior to the ones observed in practice. The reason behind the simplifications are quite well explained and motivated. The authors have done a great job explaining the ideas behind the proofs, both visually and mathematically.

**Summary Of The Paper:**

The papers studies the gradient descent dynamics on the loss $\frac{1}{2}(1-x^2y^2)^2$ as a model for deep neural networks (it can be viewed as a depth 4 scalar network with a specific initialization).

They show two (related) results: that for small enough learning rates $\eta$, there is a region of initializations $\mathbb{S}_\eta$ where gradient descent converges to a global minimum whose largest eigenvalue of the Hessian is close to $2 / \eta$. Secondly, there is a range of learning rates and a region $\mathbb{S}$ such that gradient descent starting in $\mathbb{S}$ with any learning rate in the range will converge to a solution with almost minimal stability (i.e. a largest eigenvalue close to $2 / \eta$).

The authors then show that the same does not apply to shallow networks, suggesting that this phenomenon is characteristic to deeper networks.

Finally to motivate the requirement to start within a specific region the authors show empirically (as well as motivating theoretically) that the same loss can lead to chaotic behavior in the trajectory of the gradient. This might explain the difficulty in proving any global convergence.

**Summary Of The Review:**

The paper shows the appearance of Edge of Stability dynamics in a simplified model of deep neural networks. The results are interesting and well explained.

---

> ### Author Response · Authors · 2022-11-17
> **Response to Reviewer DD1S**
>
> Dear reviewer, thank you very much for your recognition and constructive comments on our work.
>
> We have updated the manuscript and addressed the two remarks. Moreover, we have included more experiments connected to the real-world model (in **Section 6** and Appendix **Section A.6**) and stochastic gradient descent (in Appendix **Section A.7**). We hope you will find the additional empirical findings interesting.
>
> We are looking forward to your further comments, and please let us know if there is anything else we can clarify on.

---

### Official Review · Reviewer_sSz2 · 2022-10-23

**Confidence:** 2
**Clarity, Quality, Novelty And Reproducibility:** .
**Correctness:** 4
**Technical Novelty And Significance:** 2
**Empirical Novelty And Significance:** 2
**Recommendation:** 5

**Strength And Weaknesses:**

Strength. I believe this paper has some contribution towards  understanding the EOS phenomenon, especially by pointing out the difference between degree 2 construction and the connection between chaos theory. The analysis seems non-trivial.

Weakness. The paper works on some toy case, as pointed out by the paper, even the toy case seems to be complicated and the paper provides theoretical guarantee only over local region. The major unsatisfactory point for me is that it is hard to generalize the finding of this paper to general regime.

**Summary Of The Paper:**

The paper studies the the phenomenon of "Edge-of-Statility" (EOS) training dynamics and analyses the gradient dynamics on a simple example (the example is simple but the analysis is quite complicated).

The EOS of training dynamics, initially observed by [Cohen et al. 2021], refers the phenomenon that when running gradient descent for deep neural networks with learning rate $\eta$, despite the fact that the sharpness (maximum eigenvalue of the Hessian) is often larger than stability threshold $2/\eta$, the loss converges in the long run, and the sharpness at the end is around $2/\eta$.

The paper examines the function of $f(x, y, z, w) = \frac{1}{2}(1- xyzw)^2$ and proves the EOS occurs in a local region. The paper also highlights the importance of considering degree 4 function versus degree 2 (or degree 3), and argues that even for this simple example, global analysis is hard since it corresponds to some problems in chaos theory.



**Summary Of The Review:**

The paper provides a simple example and analyze the GD dynamics on it, however it is unclear how does it contribute to the broad understanding of EOS.

---

> ### Author Response · Authors · 2022-11-17
> **Response to Reviewer sSz2**
>
> Dear reviewer, thank you very much for your recognition and constructive comments on our work.
>
> For the detailed comments in the review, our responses are as follows:
>
> >  The paper works on some toy case, as pointed out by the paper, even the toy case seems to be complicated and the paper provides theoretical guarantee only over local region.
>
> We want to highlight that even though we consider a simple setting, proving EoS without any assumption is not easy. Even in the local region, most of the other works either require additional assumptions that may not always hold or only provide asymptotic results. Therefore, we believe that our result has its distinct contribution to the line of research for EoS.
>
> On the region of theoretical guarantee, we would like to highlight that currently our theorem allows the initialization to be in a constant size region. Further extending the region is very technically challenging given the bifurcation structure discussed in **Section 5**. Thus, proving a global convergence result would be hard, and we view this as an interesting future direction.
>
> > The major unsatisfactory point for me is that it is hard to generalize the finding of this paper to general regime.
>
> Thank you for bringing up this point. In the updated manuscript, we have included some additional experiments (see **Section 6** in main text and **Section A.6** in appendix) on training a over-parameterized deep fully-connected network on a binary subset of CIFAR-10. Despite the simple setting, we could show that our minimalist model shares some essential observations with more general settings with real word data. We observed that toward the final converging phase, the trajectory of the over-parameterized model mainly lies on a 2-dimensional subspace spanned by an "oscillating direction" and a "movement direction". The trajectory can also be very well characterized by a parabola as in our theoretical result on the scalar example. With this connection to real-world networks and dataset, we believe our understanding and rigorous analysis of the scalar model can bring useful insight for understanding EoS on general over-parameterized models.
>
> Thank you again for your valuable feedbacks. We hope the response can address some of the questions and concerns.
>
> We are eager to hear about your further comments. Please let us know if there is anything else we can clarify on.

---

### Official Review · Reviewer_8B9h · 2022-10-25

**Confidence:** 3
**Correctness:** 4
**Technical Novelty And Significance:** 3
**Empirical Novelty And Significance:** 3
**Recommendation:** 8

**Clarity, Quality, Novelty And Reproducibility:**

The paper is reasonably clearly written. I didn't check the proofs in the appendix.

I find the technical contributions of the paper new and interesting, but I'm not sure about their significance in the present form. Rigorous results are proved only for a toy model and only confirm a known phenomenon. On the other hand, the discussion in section 5 of global sharpness pattern as resulting from "de-bifurcation" seems conceptually novel and interesting, but is presented only on a phenomenological level. It is also not clear to which extent it is relevant for real world problems.

**Strength And Weaknesses:**

EoS is a popular topic, and there are several concurrent submissions explaining this effect from somewhat different points of view (e.g. https://openreview.net/forum?id=nhKHA59gXz, https://openreview.net/forum?id=R2M14I9LEwW). Compared to these other works, the present submission has a rather narrow scope limited to one or two toy models. Accordingly, there are no empirical results involving real world data. I also have the impression that, relative to the other submissions, the exposition is a little bogged down by technical details and conceptually not very illuminating. Though the model is a toy one, its analysis involves cumbersome computations and does not convey easy intuition applicable in more general settings.

On the other hand, the paper does discuss some interesting points not found elsewhere, particularly the effect of the model degree on EoS and a fractal structure of converged sharpness.

**Summary Of The Paper:**

The paper studies the "Edge-of-Stability" effect on a toy model of deep linear neural network with single neuron per layer (with a further extension to another toy model of rank-1 matrix factorization). The authors demonstrate that under suitable conditions this model exhibits "sharpness adaptivity". Then, the paper shows that, due to the absence of high order terms, this adaptivity need not hold for shallow networks, in agreement with empirical observations. Finally, the paper discusses the fractal structure of the global convergence pattern and draws parallels with chaotic nonlinear systems such as the well-known logistic map.

**Summary Of The Review:**

An interesting paper that clarifies some aspects of the EoS effect, but has several weaknesses, especially in comparison to existing alternative explanations (only toy models; no experiments with real world data; somewhat too technical exposition; no rigorous results confirming the global fractal picture - probably the most interesting part of the paper)

---

> ### Author Response · Authors · 2022-11-17
> **Response to Reviewer 8B9h**
>
> Dear reviewer, thank you very much for your recognition and constructive comments on our work.
>
> For the detailed comments in the review, our responses are as follows:
>
> > Rigorous results are proved only for a toy model and only confirm a known phenomenon.
>
> We acknowledge the restriction for this work that the analysis is only done for a very simple model. However, we would like to respectfully disagree with the claim that we are only "confirming a known phenomenon on a toy model" since the existence of such "toy model" is not trivial to discover. In fact, we try to understand EoS from the first principle by discovering the minimalist model where EoS happens (including sharpness concentration and adaptivity). This is not covered in any existing literature.
>
> > There are several concurrent submissions explaining this effect from somewhat different points of view (e.g. https://openreview.net/forum?id=nhKHA59gXz, https://openreview.net/forum?id=R2M14I9LEwW). Compared to these other works, the present submission has a rather narrow scope limited to one or two toy models.
>
> The two other submissions mentioned by the reviewer have very interesting results. However, we would like to highlight that our main contribution over others is that we provide a self-contained proof for the EoS phenomenon *without any assumptions* in our setting, while other works need additional assumptions that may not always hold (such as [1]) or only have asymptotic result that does not provide explicit order of convergence rate/initialization region/step size (such as [2]).
>
> > Accordingly, there are no empirical results involving real-world data.
>
> Thank you for bringing up this point. In the updated manuscript, we have included some additional experiments (see **Section 6** in main text and **section A.6** in appendix) on training a over-parameterized deep fully-connected network on a binary subset of CIFAR-10. We observed that toward the final converging phase, the trajectory of the overparameterized model mainly lies on a 2-dimensional subspace spanned by an "oscillating direction" and a "movement direction". The trajectory itself can also be very well characterized by a parabola as in our theoretical result on the scalar example. With this connection to real-world networks and dataset, we believe our understanding and rigorous analysis of the scalar model can shed light on understanding EoS on general over-parameterized models.
>
> > On the other hand, the discussion in section 5 of global sharpness pattern as resulting from "de-bifurcation" seems conceptually novel and interesting, but is presented only on a phenomenological level.
>
> Thank you for finding the "de-bifurcation" phenomenon interesting. In general, dynamical systems exhibiting bifurcation is very difficult to analyze. Even though it is not proved rigorously, we provide an intuitive explanation based on our dynamics approximation and such intuition could give a qualitatively similar trajectory (see **Fig. 7c**). Rigorously proving such a bifurcation structure would an interesting future direction, but we believe it is beyond the scope of this paper which focuses on the EoS phenomenon.
>
> Thank you again for your valuable feedbacks. We hope the response can address some of the questions and concerns.
>
> We are eager to hear about your further comments. Please let us know if there is anything else we can clarify on.
>
> ---
> **References**
>
> [1]: Self-Stabilization: The Implicit Bias of Gradient Descent at the Edge of Stability. https://openreview.net/forum?id=nhKHA59gXz
>
> [2]: A second order regression model shows edge of stability behavior. https://openreview.net/forum?id=R2M14I9LEwW

---

> > ### Comment · Reviewer_8B9h · 2022-11-23
> > **Thank you**
> >
> > Thank you for your comments. Your arguments are convincing, and the revision has improved the paper in several aspects, so I'm increasing my score.

---

### Official Review · Reviewer_wrPK · 2022-10-28

**Confidence:** 4
**Correctness:** 4
**Technical Novelty And Significance:** 3
**Empirical Novelty And Significance:** 2
**Recommendation:** 8

**Clarity, Quality, Novelty And Reproducibility:**

The work is clear and of high quality. There is a technical novelty, and the experiments and calculations seem reproducible.







**Strength And Weaknesses:**

The paper is well-written and precise. The flow of the paper gives a good intuition for different re-parameterizations, convergence properties of the system, and a rough proof sketch outlining different training phases. Since the paper is supposed to model a complex phenomenon in neural networks, one desirable outcome is to provide an intuition about the edge of stability phenomenon. I believe it does so well. The paper also highlights the theoretical result's limitation and the roadblock, thus providing a balanced view of the contribution. It compares itself to the previous attempts at modeling the problem, highlighting their limitations.

I believe the paper would benefit from some more experiments for completeness. For instance, try to minimize loss (11) and demonstrate edge-of-stability. It would also be good to see some experiments to check if the hardness of showing global convergence is an analysis issue or a genuine artifact. To do this, the authors could explicitly run synthetic experiments on their function and vary the size of the ball for initializing their models. Repeating this for several different step sizes and dimensional problems would be insightful.

In my understanding, training with SGD may or may not demonstrate the same sharpening depending on the noise level. I am curious if the authors can extend some of their simpler results to incorporate stochasticity, perhaps just simple additive noise. It would be good to discuss how the noise scale qualitatively changes the phenomenon. Intuitively the noise might also smoothen the boundary between initialization that do and do not converge. At least some experiments with SGD can shed light on its behavior.

**Summary Of The Paper:**

This paper describes a simple non-convex function in four dimensions (a four-layer scalar network):

1. which empirically demonstrates the *"edge-of-stability"* phenomenon while training with gradient descent,
2. is qualitatively more similar to deeper neural networks than previously proposed model functions in terms of *"sharpening"*,
3. and is analytically simple enough to show a convergence theoretically to the sharpest global minimum attainable for a given step size.

The paper provides a local convergence guarantee (which they call sharpness concentration), i.e., for any learning rate, provided the initialization is in a ball around the sharpest global minima (attainable for that learning rate), the optimizer will converge to that minima. And it discusses the roadblock in extending it to a global convergence guarantee (i.e, convergence to the edge of stability for any initialization):

> the boundary separating converging and diverging initializations (Fig. 7a) exhibits complicated fractal structures.

To underline the limitations of previous works trying to model the phenomenon, the paper discusses the difference between two, three, and four-layered scalar networks. It shows, empirically and analytically, that the phenomenon gets more pronounced, i.e., there is less of a sharpness gap w.r.t. sharpest attainable minima, when depth increases.  The theoretical results are also extended to the vector case, i.e., rank-1 factorization of an isotropic matrix.

**Summary Of The Review:**

I believe the proposed model captures the essence of the edge-of-stability phenomenon while training with gradient descent. The paper is well-written and highlights why even their simple model has useful insights about different aspects of training.  I recommend accepting the paper.

---

> ### Author Response · Authors · 2022-11-17
> **Response to Reviewer wrPK**
>
> Dear reviewer, thank you very much for your recognition and constructive comments on our work.
>
> For the detailed comments in the review, our responses are as follows:
>
> > I believe the paper would benefit from some more experiments for completeness. For instance, try to minimize loss (11) and demonstrate edge-of-stability
>
> Thanks for pointing this out. We have included the experiment results for the vector example in the Appendix **Section A.5** of the revised paper.
>
> > It would also be good to see some experiments to check if the hardness of showing global convergence is an analysis issue or a genuine artifact.
>
> We believe it is intrinsically difficult for global analysis as shown by the fractal structure in **Fig.7a** (which is the zoomed-in version of the lower right part of Fig.2b where we included all initialization that converges in the positive quadrant).
>
> Besides the complicated boundary of converging and diverging initialization shown in Fig.7a, the global trajectory shown in Fig.7b also suggest the potential difficulty of proving global convergence. Such complicated boundaries can also be observed under different step size (in Appendix **Section A.2.1, Fig. 10**), which suggests that it is a quite general phenomenon.
>
> > In my understanding, training with SGD may or may not demonstrate the same sharpening depending on the noise level. I am curious if the authors can extend some of their simpler results to incorporate stochasticity, perhaps just simple additive noise.
>
> Thank you for bringing up this idea. We believe it is a very interesting and important direction to generalize the theoretical understandings to the regime of SGD. In the updated manuscript, we have added a new section (Appendix A.7) to present some primitive experiments that simulate stochasticity by adding slight perturbations to the label or gradient (e.g. for each step, minimizing $(xyzw-1+\delta)^2$ for some $\delta\sim \mathcal{N}(0, \sigma^2)$ with small $\sigma$). We also did mini-batch SGD experiments on the over-parameterized real-world models, which exhibits different phenomenon compared to the noise-injected GD for the scalar models.
>
> We note that the nature of the noise in the minibatch SGD is probably different from label/gradient noise as the over-parameterized model can eventually memorize all data and converge to a fixed point. However our scalar model can only memorize one data point and hence cannot simulate the case with minibatch SGD. We think this is a very interesting future direction to work on.
>
> Thank you again for your valuable feedbacks. We hope the response can address some of the questions and concerns.
>
> We are eager to hear about your further comments. Please let us know if there is anything else we can clarify on.

---

> > ### Comment · Reviewer_wrPK · 2022-12-04
> > **Thanks**
> >
> > Thanks for your reply and the new experiments.

---

### Official Review · Reviewer_jBkM · 2022-11-02

**Confidence:** 4
**Correctness:** 4
**Technical Novelty And Significance:** 4
**Empirical Novelty And Significance:** 2
**Recommendation:** 8

**Clarity, Quality, Novelty And Reproducibility:**

The paper is fairly clear and of good quality. The novelty value is high since there are not many analyses of optimization in this regime; in particular, many prior works focus on more complex neural network settings with many additional assumptions, whereas this work focuses on a simple example in which it is possible to establish everything from first principles.

Typos:
- Pg. 1, “training trajectory often oscillate” -> “training trajectory often oscillates”
- Pg. 4 “Despite we are able” is improper grammar
- Pg. 6, the paragraph beginning with “Note that”: the last sentence is a run-on sentence

**Strength And Weaknesses:**

The strength of the work is that it rigorously establishes the EoS phenomenon with a very careful analysis. Despite only considering a simple objective function, the dynamics are quite intricate and it is technically challenging to establish the result. As it is important to understand non-convex dynamics, especially those capturing the behavior of neural network training, this work makes a valuable contribution to the literature.

However, due to the initialization w = y and z = x, the authors really end up analyzing a 2-variable function. In fact, up to rescaling the step size, it is easy to see that the dynamics they analyze are equivalent to GD on the 2-variable function (x, y) -> (x^2 y^2 - 1)^2. Hence I believe that the introduction of the 4-variable function is highly misleading and I ask that the authors reformulate the discussion in terms of the 2-variable objective that they actually analyze.

Also, it is difficult to claim that this example is minimalistic considering that the analysis and dynamics are so complicated, but I believe it is out of scope of the present work to greatly simplify the analysis.


**Summary Of The Paper:**

This work is motivated by the “edge of stability” phenomenon, observed by Cohen et al.: when training neural networks with step size η, the largest eigenvalue of the Hessian (dubbed the “sharpness”) along the training trajectory often stabilizes near 2/η, which is the stability threshold in convex optimization. The paper carefully analyzes the GD dynamics on the loss function (w, x, y, z) -> (wxyz - 1)^2 (actually because the initialization is taken to be w = y and z = x, it is really a 2-variable analysis, see comment below) and shows that for a region of initializations and step sizes, the limiting sharpness of the GD trajectory is 2/η + O(η).

**Summary Of The Review:**

I have issues with the way that the results are presented; namely, I believe that the paper should not mention the four-variable objective if it is not used in the analysis. Overall, I think that the contribution is solid, the problem setting (EoS) is important, and the work is technically difficult, and hence it merits acceptance.

---

> ### Author Response · Authors · 2022-11-17
> **Response to Reviewer jBkM**
>
> Dear reviewer, thank you very much for your recognition and constructive comments on our work. We have updated the paper to address the typos.
>
> For the detailed comments in the review, our responses are as follows:
>
> > In fact, up to rescaling the step size, it is easy to see that the dynamics they analyze are equivalent to GD on the 2-variable function $(x, y) \mapsto (x^2 y^2 - 1)^2$.
>
> Thank you for pointing this out. We agree that introducing the theoretical analysis in the context of a 4-layer network is a bit misleading given that the analysis is fundamentally 2D. In the updated version, we have explicitly point out this equivalence when we first introduce the model. We also changed the reference of the "4-scalar model" to "degree-4 model" to avoid confusion.
>
> By the end of **Section 4** and Appendix **Section A.4** (with more detail), we discussed some empirical results on how the training of general scalar networks (initialized without equal entries) reduces to our regime with duplicating entries values. Hence the coupling of entries is likely to be a consequence of GD dynamics on linearly activated scalar networks in a more general setting instead of an artifact we have to oppose on a model for EoS to exist. Thus we believe it is still helpful for keeping the general narrative of "degree-4" instead of reducing to the quadratic case of $x^2y^2$.
>
> > Also, it is difficult to claim that this example is minimalistic considering that the analysis and dynamics are so complicated
>
> We agree that our proof is quite complicated. Here we use the term "minimalist" to reflect that our scalar network model is structurally minimal for sharpness concentration and adaptivity to happen. It is a function on two variables (and since edge of stability needs an oscillating direction and a movement direction, having one variable should not be possible), and simpler functions such as $(xy-1)^2$ do not have the same properties.
>
> Thank you again for your valuable feedbacks. We hope the response can address some of the concerns.
>
> We are eager to hear about your further comments on the presentation issue. Please let us know if there is anything else we can clarify on.

---

### Author Response · Authors · 2022-11-17
**Revised Manuscript Uploaded.**

Dear Reviewers, thank you so much for the constructive and insightful advice. Following your suggestions, we have updated a revision of our paper.

In this update, we have made the following changes to the paper's content:

* Added **Section 6** to include experiments on real-world dataset and models (training over-parameterized fully connected network on a binary subset of CIFAR-10). Discussed the similarity between the converging trajectory for the real-world model and the degree-4 scalar model.

* Added Appendix **Section A.6** for detailed explanation and additional results about the real-world experiments.

* Moved **Section 4.2** (degree-3 example with mixed behavior around different EoS minima) to Appendix **Section A.2.3**.

* Added Appendix **Section A.5** for experiment results on rank-1 approximation of isotropic matrix (Theorem 3.2).

* Added Appendix **Section A.7** to include experiment results and discussion on noisy GD of the scalar model and mini-batch SGD on over-parameterized fully-connected networks trained on a CIFAR-10 subset.

* Added **Figure 10** in Appendix Section A.2.1 for presenting Figure 2b under different step sizes.

We have also made the following minor changes to enhance the presentation of the paper

* Added discussion on comparing our paper and existing literature.

* Changed "4-scalar model" to "degree-4 model" to avoid confusion between the general 4-layer scalar network and the coupling model $\mathcal{L}(x,y)=\frac14(1-x^2y^2)^2$.

* Changed "$\mathbb{S}$" in Theorem 1.1 (Informal) to $\mathbb{S}_\eta$ to explicitly reflect its dependency on learning rate $\eta$.

* Fixed several typos and grammar issues.

We hope that the modifications can strengthen the state of our submission. Please let us know if there are any additional points that we can clarify/modify!

---

### Decision · Program_Chairs · 2023-01-20

**Decision:**

Accept: poster

**Justification For Why Not Higher Score:**

Interesting paper that gives some additional evidence about the EOS phenomenon but the result is still restricted to a toy example, so this does not seem to be a paper that should be presented as a spotlight. This is reflected in the review of Reviewer sSz2 who was slightly more negative than the other reviewers.

**Justification For Why Not Lower Score:**

Paper brings some new insights, all reviewers agree it should be accepted.

**Metareview: Summary, Strengths And Weaknesses:**

The paper studies the "Edge-of-Statility" (EOS)  phenomenon for a simple example consisting of a deep linear neural network with a single neuron per layer.

The model analyzed by the authors is very simple and some of the reviewers were initialized semi enthusiastic about the results. However, during the discussion with the authors, the reviewers agreed that this is a difficult problem to analyze and that the paper does make some novel contributions and does provide somewhat novel insights into the optimization dynamics.

Some additional experiments were also added (post reviews) by the authors, providing additional evidence supporting the claims made in the paper.

Overall, some reviewers (notably Reviewer 8B9h who was more negative before the rebuttal) ended up raising their scores and I, therefore, recommend acceptance.

**Note From Pc:**

if the above contains the word "oral" or "spotlight" please see: "oral" presentation means -> notable-top-5% and "spotlight" means -> notable-top-25%. As stated in our emails, we are disassociating presentation type from AC recommendations

**Summary Of Ac-Reviewer Meeting:**

N/A